# Private Estimation with Public Data[*]

**Alex Bie**
University of Waterloo
yabie@uwaterloo.ca

**Gautam Kamath**
University of Waterloo
g@csail.mit.edu

**Vikrant Singhal**
University of Waterloo
vikrant.singhal@uwaterloo.ca

## Abstract

We initiate the study of differentially private (DP) estimation with access to a small amount of public data. For private estimation of $d$-dimensional Gaussians, we assume that the public data comes from a Gaussian that may have vanishing similarity in total variation distance with the underlying Gaussian of the private data. We show that under the constraints of pure or concentrated DP, $d + 1$ public data samples are sufficient to remove any dependence on the range parameters of the private data distribution from the private sample complexity, which is known to be otherwise necessary without public data. For separated Gaussian mixtures, we assume that the underlying public and private distributions are the same, and we consider two settings: (1) when given a dimension-independent amount of public data, the private sample complexity can be improved polynomially in terms of the number of mixture components, and any dependence on the range parameters of the distribution can be removed in the approximate DP case; (2) when given an amount of public data linear in the dimension, the private sample complexity can be made independent of range parameters even under concentrated DP, and additional improvements can be made to the overall sample complexity.

## 1 Introduction

Differential privacy (DP) [DMNS06] guarantees the privacy of *every* point in a dataset. This is a strong requirement, and often gives rise to qualitatively new requirements when performing private data analysis. For instance, for the problem of private mean estimation (under pure or concentrated DP), the analyst must specify a guess for the unknown parameter, and needs more data to get an accurate estimate depending on how good their guess is. This cost can be prohibitive in cases where the data domain may be unfamiliar.

Fortunately, in many cases, it is natural to assume that there exists an additional *public* dataset. This public dataset may vary in both size and quality. For example, one could imagine that a fraction of users opt out of privacy considerations, giving a small set of public in-distribution data. Alternatively, it is common to pretrain models on large amounts of public data from the web, which may be orders of magnitude larger than the private data but significantly out of distribution. In a variety of such settings, this public data can yield dramatic theoretical and empirical improvements to utility in private data analysis (see discussion in Section 1.3). We seek to answer the following question:

*How can one take advantage of public data for private estimation?*

We initiate the study of differentially private statistical estimation with a supplementary public dataset. In particular, our goal is to understand when a small amount of public data can significantly reduce the cost of private estimation.

---

[*]Authors are listed in alphabetical order.

36th Conference on Neural Information Processing Systems (NeurIPS 2022).

## 1.1 Problem Formulation

We investigate private estimation with public data through the canonical problems of estimating Gaussians and mixtures of Gaussians. We focus on parameter estimation from i.i.d. samples drawn from public and private data distributions.

We wish to design estimation algorithms that take as input both public data and private data, and are private with respect to private data. Formally, we say a randomized algorithm $\mathcal{A} : \mathcal{X}^m \times \mathcal{X}^n \to \mathcal{Y}$ taking as input public data $\widetilde{X} \in \mathcal{X}^m$ and private data $X \in \mathcal{X}^n$ is *private with respect to private data* if for any setting of the public data $\widetilde{X}$, $\mathcal{A}(\widetilde{X}, \cdot) : \mathcal{X}^n \to \mathcal{Y}$ is differentially private.[2]

**Gaussians.** Let $\mu, \widetilde{\mu} \in \mathbb{R}^d$ and positive definite $\Sigma, \widetilde{\Sigma} \in \mathbb{R}^{d \times d}$ be unknown, with $d_{\mathrm{TV}}(\mathcal{N}(\mu, \Sigma), \mathcal{N}(\widetilde{\mu}, \widetilde{\Sigma})) \leq \gamma$ for some known $0 \leq \gamma < 1$. Then for $0 < \alpha, \beta < 1$, given public samples from $\mathcal{N}(\widetilde{\mu}, \widetilde{\Sigma})$ and private samples from $\mathcal{N}(\mu, \Sigma)$, we would like to output $\hat{\mu}, \widehat{\Sigma}$ such that with probability $\geq 1 - \beta$, $d_{\mathrm{TV}}(\mathcal{N}(\mu, \Sigma), \mathcal{N}(\hat{\mu}, \widehat{\Sigma})) \leq \alpha$.

**Mixtures of Gaussians.** We represent a Gaussian mixture $D$ by a sequence of $k$ tuples $((\mu_1, \Sigma_1, w_1), \ldots, (\mu_k, \Sigma_k, w_k))$, each representing the mean, covariance matrix, and mixing weight of a component. Then for $0 < \alpha, \beta < 1$, given public and private samples drawn from $D$, we would like to output a Gaussian mixture $\widehat{D} = \{(\widehat{\mu}_1, \widehat{\Sigma}_1, \widehat{w}_1), \ldots, (\widehat{\mu}_k, \widehat{\Sigma}_k, \widehat{w}_k))\}$ such that with probability $\geq 1 - \beta$, for each $i \in [k]$, there exists a distinct $j \in [k]$, such that $d_{\mathrm{TV}}(\mathcal{N}(\mu_i, \Sigma_i), \mathcal{N}(\widehat{\mu}_j, \widehat{\Sigma}_j)) \leq \alpha$ and $|w_i - \widehat{w}_j| \leq \alpha/k$. This implies $d_{\mathrm{TV}}(D, \widehat{D}) \leq O(\alpha)$, and we refer to this stronger notion of component-wise estimation as $(\alpha, \beta)$-learning (Definition A.10).

## 1.2 Results

**Gaussians.** In the case of pure and concentrated DP, exactly $d + 1$ public samples are sufficient to remove any dependence on "range parameters" in the private sample complexity, even when the underlying Gaussian distribution of the public data bears *almost no similarity* to that of the private data. We start with following result for the scenario when the two distributions are the same.

**Theorem 1.1.** *For all $\alpha, \beta, \varepsilon > 0$ and $d \geq 1$ there exists a computationally efficient $\frac{\varepsilon^2}{2}$-zCDP (and a computationally inefficient $\varepsilon$-DP) algorithm $\mathcal{M}$ that takes $d + 1$ public samples and $n$ private samples from a Gaussian $D$ over $\mathbb{R}^d$, which is private with respect to private data, such that if $n = O\left(\frac{d^2}{\alpha^2} + \frac{d^2}{\alpha\varepsilon}\right) \cdot \mathrm{polylog}\left(d, \frac{1}{\beta}, \frac{1}{\alpha}\right)$, then $\mathcal{M}$ estimates the parameters of $D$ to within TV error $\alpha$ with probability $\geq 1 - \beta$.*

Without public data, the analyst would be required to specify a bound $R$ on the $\ell_2$ norm of the mean and a bound $K$ on the condition number of the covariance, and the private sample complexity would scale logarithmically in terms of these parameters (which we denote as *range parameters*). Thus, this cost could be arbitrarily large, or even infinite, given poor *a priori* knowledge of the problem. Note that these requirements are not due to suboptimality of current private algorithms. Packing lower bounds for estimation under pure and concentrated DP show that dependence on range parameters is an inherent cost of privacy. However, this dependence can be entirely eliminated at the price of $d + 1$ public data points, which is less public data than the non-private $\Theta(\frac{d^2}{\alpha^2})$ sample complexity of the problem. For estimating Gaussians with identity or known covariance (that is, mean estimation) it turns out that just *one* public sample suffices to remove all dependence on range parameters.

The next natural question to ask is: what if our public data does not come from the same distribution as our private data? Indeed, in practical settings, there may be significant domain shift between data that is publicly available and the private data of interest. Our next result shows that we can relax this assumption: it suffices that our public data comes from another Gaussian that is at most $\gamma < 1$ away in TV distance from the underlying private data distribution.

**Theorem 1.2.** *For all $\alpha, \beta, \varepsilon > 0$, $d \geq 1$, and $0 \leq \gamma < 1$, there exists a computationally efficient $\frac{\varepsilon^2}{2}$-zCDP (and a computationally inefficient $\varepsilon$-DP) algorithm $\mathcal{M}$ that takes $d + 1$ public samples from*

---

[2]Previous work exploring private algorithms with additional public data, such as [BNS16a; BCMNUW20], adopt this privacy requirement for their algorithms.

a Gaussian $\widetilde{D}$ over $\mathbb{R}^d$ and $n$ private samples from a Gaussian $D$ over $\mathbb{R}^d$ with $\mathrm{d}_{\mathrm{TV}}(D, \widetilde{D}) \leq \gamma$, which is private with respect to private data, such that if $n = O\left(\frac{d^2}{\alpha^2} + \frac{d^2}{\alpha\varepsilon}\right) \cdot \mathrm{polylog}\left(d, \frac{1}{\beta}, \frac{1}{\alpha}, \frac{1}{1-\gamma}\right)$, then $\mathcal{M}$ estimates the parameters of $D$ to within TV error $\alpha$ with probability $\geq 1 - \beta$.

The upper bound on the TV gap $\gamma$ results in the sample complexity picking up an extra $\mathrm{polylog}(\frac{1}{1-\gamma})$ factor. For instance, even when the public and private distributions have total variation distance $1 - 1/d^{100}$ (that is, they are *almost entirely dissimilar*), we incur only an extra $\mathrm{poly}\log d$ factor in the sample complexity. In particular, note that the public dataset may be simultaneously quite small and significantly out of distribution with respect to the private data.

**Mixtures of Gaussians.** We focus on estimation of mixtures of $k$ *separated* Gaussians in $d$ dimensions. Specifically, we assume that, given mixture components $\mathcal{N}(\mu_i, \Sigma_i)$ with respective mixing weights $w_i$ for $i \in [k]$ (with $w_i \geq w_{\min}$), then for each $i \neq j$,

$$\|\mu_i - \mu_j\|_2 \geq \widetilde{\Omega}\left(\sqrt{k} + \frac{1}{\sqrt{w_i}} + \frac{1}{\sqrt{w_j}}\right) \cdot \max\left\{\sqrt{\|\Sigma_i\|_2}, \sqrt{\|\Sigma_j\|_2}\right\}. \tag{1}$$

Private estimation of such mixtures of Gaussians (without public data) was previously studied by [KSSU19]. Their algorithms proceed in two stages: first, a clustering step to isolate the samples from each component, and an estimation step, which estimates the parameters of each individual Gaussian using the isolated samples from each component. Our algorithms follow a similar structure, but employ the public data at appropriate points in the procedure to enable substantial savings.

In all the results that follow, $\widetilde{O}$ hides the $\mathrm{polylog}$ factors of $d, k$, the inverse of error parameters $\frac{1}{\alpha}, \frac{1}{\beta}$, and the inverse of privacy parameters $\frac{1}{\varepsilon}, \frac{1}{\delta}, \frac{1}{\rho}$. We provide two flavours of results for learning mixtures of Gaussians. First, we investigate the setting when we have a very small, dimension-independent amount of public data.

**Theorem 1.3.** *For all $\alpha, \beta, \varepsilon, \delta > 0$, there exists an $(\varepsilon, \delta)$-DP algorithm $\mathcal{M}$ that takes $n$ private samples and $m$ public samples from a Gaussian mixture $D$, satisfying (1), which is private with respect to private data, such that if $m = O\left(\frac{\log(k/\beta)}{w_{\min}}\right)$ and $n = \widetilde{O}\left(\frac{d^2}{w_{\min}\alpha^2} + \frac{d^2}{w_{\min}\alpha\varepsilon} + \frac{d^{1.5}k^{2.5}}{\varepsilon}\right)$, then $\mathcal{M}$ $(\alpha, \beta)$-learns $D$.*

**Theorem 1.4.** *For all $\alpha, \beta, \rho > 0$, there exists a $\rho$-zCDP algorithm $\mathcal{M}$ that takes $n$ private samples and $m$ public samples from a Gaussian mixture $D$ satisfying (1), which is private with respect to private data, such that if $\forall i, \|\mu_i\|_2 \leq R$ and $\mathbb{I} \preceq \Sigma_i \preceq K\mathbb{I}$, and $m = O\left(\frac{\log(k/\beta)}{w_{\min}}\right)$ and $n = \widetilde{O}\left(\frac{d^2}{w_{\min}\alpha^2} + \frac{d^2}{w_{\min}\alpha\sqrt{\rho}} + \frac{d^{1.5}k^{2.5}}{\sqrt{\rho}}\right) \cdot \mathrm{polylog}(K, R)$, then $\mathcal{M}$ $(\alpha, \beta)$-learns $D$.*

Our results differ from the prior work of [KSSU19] in the following ways: (1) we have results for both approximate DP and zCDP, as opposed to just approximate DP in [KSSU19]; (2) the private sample complexity due to the clustering step does not involve any dependence on the range parameters; and (3) the sample complexity due to the clustering step is now $O(d^{1.5}k^{2.5})$, as opposed to $O(d^{1.5}k^{9.06})$, which is a significant improvement, and shows up only due to applications of private PCA under better sensitivity guarantees. As far as the private sample complexity in the component estimation process is concerned, in the case of approximate DP, we have expressions independent of distribution parameters owing to the recent Gaussian learner by [AL21].

Finally, we investigate Gaussian mixture estimation with an amount of public data which is linear in the dimension.

**Theorem 1.5.** *For all $\alpha, \beta, \varepsilon, \delta > 0$, there exists an $(\varepsilon, \delta)$-DP algorithm $\mathcal{M}$ that takes $n$ private samples and $m$ public samples from a Gaussian mixture $D$ satisfying (1), which is private with respect to private data, such that if $m = O\left(\frac{d\log(k/\beta)}{w_{\min}}\right)$ and $n = \widetilde{O}\left(\frac{d^2}{w_{\min}\alpha^2} + \frac{d^2}{w_{\min}\alpha\varepsilon}\right)$, then $\mathcal{M}$ $(\alpha, \beta)$-learns $D$.*

**Theorem 1.6.** *For all $\alpha, \beta, \rho > 0$, there exists a $\rho$-zCDP algorithm $\mathcal{M}$ that takes $n$ private samples and $m$ public samples from a Gaussian mixture $D$ satisfying (1), which is private with respect to private data, such that if $m = O\left(\frac{d\log(k/\beta)}{w_{\min}}\right)$ and $n = \widetilde{O}\left(\frac{d^2}{w_{\min}\alpha^2} + \frac{d^2}{w_{\min}\alpha\sqrt{\rho}}\right)$, then $\mathcal{M}$ $(\alpha, \beta)$-learns $D$.*

Using $\widetilde{O}(d/w_{\min})$ public samples establishes all the advantages that using $\widetilde{O}(1/w_{\min})$ public samples gives, but additionally improves the private sample complexity even further. As we will describe in the technical overview, the clustering process is done entirely using the public data, and ends up removing the term due to private PCA from the private sample complexity in both approximate DP and zCDP regimes. Additionally, in the zCDP case, we do not have any dependence on distribution range parameters even in the component estimation process because we can use results from Theorem 1.1 this time.

## 1.3 Related Work

A large part of our work focuses on using public data to obviate the dependence on range parameters in private estimation. Understanding these dependences without public data has been a topic of significant study. [KV18] investigated private estimation of univariate Gaussians, showing that logarithmic dependences on the range parameters were both necessary and sufficient for pure differential privacy, but by using stability-based histograms [KKMN09; BNS16b], they could be removed under approximate differential privacy. Similar results have been shown in the multivariate setting: while logarithmic dependences are necessary and sufficient under pure or concentrated DP [KLSU19], they can be removed under approximate DP [AAK21; KMSSU21; TCKMS21; AL21; KMV21; LKO21]. More broadly, a common theme is that the dependences which are intrinsic to pure or concentrated DP can be eliminated by relaxing to approximate DP. Our results demonstrate that instead of relaxing the privacy definition for all the data, one can achieve the same goal by employing a much smaller amount of public data.

Our investigation fits more broadly into a line of work employing public data for private data analysis. Works, both theoretical and empirical, investigate the role of public data in private query release, synthetic data generation, and prediction [JE13; BNS16a; ABM19; NB20; BCMNUW20; BMN20; LVSUW21]. For distribution-free classification with public and private data, it was shown in [ABM19] that, roughly speaking, if a hypothesis class is privately learnable with a small amount of public data ($o(1/\alpha)$), it is privately learnable with no public data. For density estimation, we show that the family of unbounded Gaussians is learnable with a small amount of public data ($\alpha$-independent), despite the fact that it is not learnable without public data.

Additionally, a number of empirical works study the efficacy of public data in private machine learning, including methods such as public pre-training [ACGMMTZ16; PCSTE19; TB21; LWAFF21; YZCL21; LTLH22; Yu+22], using public data to compute useful statistics about the private gradients [ZWB21; YZCL21; KRRT21; AGMRSSTT21], or using unlabeled public data to train a student model [PAEGT17; PSMRTE18; BTGT18].

Another setting with mixed privacy guarantees is when different users may require local versus central differential privacy [AKZHL17]. Mean estimation has also been studied in this setting [ADK19].

## 1.4 Organization

The remainder of the paper is organized as follows. In Section 2, we discuss our results for estimating multivariate Gaussians. After that in Section 3, we give a technical overview of our results for estimating mixtures of Gaussians. We present all the relevant notations and preliminaries about differential privacy, Gaussian mixtures, and techniques like PCA, along with a few useful concentration inequalities in Appendix A. In Appendix B, we provide further technical details that are missing from Section 2. We provide all our technical results about mixtures of Gaussians from Section 3, including algorithms, theorems, and proofs, in Appendix C. Finally, in Appendix D we present some proof-of-concept numerical simulations demonstrating the effectivenss of public data for private estimation.

## 2 Estimating Gaussians

We discuss our results for estimating multivariate Gaussians with access to a limited amount of public data. First, as a warm-up, we consider the case of estimating the mean of an identity (or known) covariance Gaussian, and give a $\rho$-zCDP private algorithm that uses 1 public sample. We then generalize the high-level approach taken there to arrive at a private algorithm that uses $(d+1)$ public samples for general Gaussians. Furthermore, we show that we can relax the condition that the

public and the private data come from the same distribution – the public data can come from another Gaussian that is at most $\gamma$ far away in TV distance.

## 2.1 Warmup: Identity Covariance Gaussians with $1$ Public Sample

In this setting, we are given a single public sample $\widetilde{X}$, along with private samples $X_1, \ldots, X_n$, where $\widetilde{X}$ and the $X_j$ are drawn from a $d$-dimensional Gaussian $\mathcal{N}(\mu, \mathbb{I})$ independently. Our main observation is that a single public sample is sufficient to get a "good enough" coarse estimate of $\mu$, allowing us to apply existing private algorithms for a finer estimate. This is captured in the following claim, which is a standard fact regarding norm concentration of Gaussian vectors (Lemma A.3).

**Claim 2.1.** *Suppose we get a sample $\widetilde{X} \sim \mathcal{N}(\mu, \mathbb{I})$. Then with probability at least $1 - \beta$, $\|\mu - \widetilde{X}\|_2 \leq \sqrt{d + 2\sqrt{d \log(1/\beta)} + 2 \log(1/\beta)}$.*

We restate the guarantees of an existing folklore $\rho$-zCDP "clip-and-noise" algorithm via the Gaussian mechanism for estimating identity covariance Gaussians.

**Lemma 2.2** (zCDP Private Gaussian Mean Estimation). *For all $\alpha, \beta, \rho > 0$, there exists a $\rho$-zCDP algorithm that takes $n = \widetilde{O}\left(\frac{d + \log(1/\beta)}{\alpha^2} + \frac{\sqrt{d}(R + \sqrt{d}) \log(1/\beta)}{\alpha \sqrt{\rho}}\right)$ private samples from $\mathcal{N}(\mu, \mathbb{I})$ over $\mathbb{R}^d$, where $\|\mu\|_2 \leq R$, and outputs an estimate $\widehat{\mu} \in \mathbb{R}^d$, such that with probability at least $1 - \beta$, $\mathrm{d}_{\mathrm{TV}}(\mathcal{N}(\mu, \mathbb{I}), \mathcal{N}(\widehat{\mu}, \mathbb{I})) \leq \alpha$.*

The above "clip-and-noise" mechanism simply clips all the points in the dataset to within radius $\lambda = R + \sqrt{d + 2\sqrt{d \log(n/\beta)} + 2 \log(n/\beta)}$ around the origin, and outputs the noisy empirical mean of the clipped points via the Gaussian mechanism.

Claim 2.1, combined with the result above, yields a 1-public-sample, private mean estimation algorithm via the following three-step process: (1) form shifted private dataset $Y_1, \ldots, Y_n$ by taking each $Y_j := X_j - \widetilde{X}$; (2) run "clip-and-noise" on $Y_1, \ldots, Y_n$, setting $R = \sqrt{d + 2\sqrt{d \log(2/\beta)} + 2 \log(2/\beta)} = O(\sqrt{d \log(1/\beta)})$ and target failure probability $\beta/2$, to obtain $\widehat{\mu}_Y$; and (3) output $\widehat{\mu} := \widehat{\mu}_Y + \widetilde{X}$.

By a union bound, with probability $\geq 1 - \beta$, the distribution of $Y_1, \ldots, Y_n$ satisfies tight boundedness independent of the range parameters, and furthermore, succeeds in outputting a good estimate $\widehat{\mu}_Y$ for $\mu_Y := \mathbb{E} Y_1$. The last step undoes the shift to yield a good estimate $\widehat{\mu}$ for the original $\mu$. Privacy is preserved with respect to private data, since we are doing entry-wise pre-processing and post-processing with public data. For full version of this argument (which includes the $\varepsilon$-DP case), see Theorem B.2.

This yields a private sample complexity that no longer depends on the range bound $R$, allowing us to use a fixed sample size to estimate the family of all identity covariance Gaussians over $\mathbb{R}^d$.

**Theorem 2.3** (zCDP Gaussian Mean Estimation with Public Data). *For all $\alpha, \beta, \rho > 0$, there exists a $\rho$-zCDP algorithm that is private with respect to the private samples, takes $1$ public sample and $n = \widetilde{O}\left(\frac{d + \log(1/\beta)}{\alpha^2} + \frac{d \log^{1.5}(1/\beta)}{\alpha \sqrt{\rho}}\right)$ private samples from $\mathcal{N}(\mu, \mathbb{I})$ over $\mathbb{R}^d$, and outputs an estimate $\widehat{\mu} \in \mathbb{R}^d$, such that with probability at least $1 - \beta$, $\mathrm{d}_{\mathrm{TV}}(\mathcal{N}(\mu, \mathbb{I}), \mathcal{N}(\widehat{\mu}, \mathbb{I})) \leq \alpha$.*

## 2.2 General Gaussians with $d + 1$ Public Samples

In this section, we discuss our main result for privately estimating Gaussians with public data.

**Theorem 2.4** (Private Gaussian Estimation with Public Data). *For all $\alpha, \beta, \varepsilon, \rho > 0$, there exist $\rho$-zCDP and $\varepsilon$-DP algorithms that take $d + 1$ public samples $\widetilde{X}_1, \ldots, \widetilde{X}_{d+1}$ and $n$ private samples $X_1, \ldots, X_n$ from unknown, $d$-dimensional Gaussians $\mathcal{N}(\widetilde{\mu}, \widetilde{\Sigma})$ and $\mathcal{N}(\mu, \Sigma)$, respectively, with $0 \leq \mathrm{d}_{\mathrm{TV}}(\mathcal{N}(\mu, \Sigma), \mathcal{N}(\widetilde{\mu}, \widetilde{\Sigma})) \leq \gamma < 1$, are private with respect to the private samples, and return $\widehat{\mu}_X \in \mathbb{R}^d$ and $\widehat{\Sigma}_X \in \mathbb{R}^{d \times d}$, such that $\mathrm{d}_{\mathrm{TV}}(\mathcal{N}(\widehat{\mu}_X, \widehat{\Sigma}_X), \mathcal{N}(\mu, \Sigma)) \leq \alpha$ with probability at least $1 - \beta$, as long as the following bounds on $n$ hold.*

    *1. For $\rho$-zCDP (with computational efficiency) via [KLSU19] and $d + 1$ public samples,*

$$n = O\left(\frac{d^2 + \log\left(\frac{1}{\beta}\right)}{\alpha^2} + \frac{d^2 \cdot \operatorname{polylog}\left(\frac{d}{\alpha\beta\rho}\right) + d\log\left(\frac{d}{\alpha\beta\rho(1-\gamma)}\right)}{\alpha\sqrt{\rho}} + \frac{d^{1.5} \cdot \operatorname{polylog}\left(\frac{d}{\beta\rho(1-\gamma)}\right)}{\sqrt{\rho}}\right).$$

2. *For $\varepsilon$-DP (without computational efficiency) via [BKSW19] and $d+1$ public samples,*

$$n = O\left(\frac{d^2 + \log\left(\frac{1}{\beta}\right)}{\alpha^2} + \frac{d^2\log\left(\frac{d}{\alpha\beta(1-\gamma)}\right)}{\alpha\varepsilon}\right).$$

For simplicity, the rest of this section examines the $\gamma = 0$ case, that is, when the public data and the private data distributions are the same. Later, we address what modifications are needed for the $\gamma > 0$ case.

**Public Data Preconditioning.** We extend the same high-level idea from the identity covariance case. We use public data to do coarse estimation, and then use that coarse estimate to transform the private data, reducing to the bounded case that can be solved using existing private algorithms (Lemma A.22). Specifically, we use the $d+1$ public samples for "public data preconditioning".

---

**Algorithm 1:** Public Data Preconditioner $\operatorname{PubPreconditioner}_\beta(\widetilde{X})$

**Input:** Public samples $\widetilde{X} = (\widetilde{X}_1, \ldots, \widetilde{X}_{d+1})$. Failure probability $\beta > 0$.
**Output:** $\widehat{\mu} \in \mathbb{R}^d$, $\widehat{\Sigma} \in \mathbb{R}^{d \times d}$, $L \in \mathbb{R}$, $U \in \mathbb{R}$.

// Compute the empirical mean and covariance of $\widetilde{X}$.

$$\widehat{\mu} \leftarrow \frac{1}{d+1}\sum_{i=1}^{d+1}\widetilde{X}_i \quad \text{and} \quad \widehat{\Sigma} \leftarrow \frac{1}{d}\sum_{i=1}^{d+1}(\widetilde{X}_i - \widehat{\mu})(\widetilde{X}_i - \widehat{\mu})^T$$

// Compute L and U.

$$L \leftarrow \frac{d}{4d + 4\sqrt{2d\log\frac{3}{\beta}} + 2\log\frac{3}{\beta}} \quad \text{and} \quad U \leftarrow \frac{9d^2}{\beta^2}$$

**Return** $(\widehat{\mu}, \widehat{\Sigma}, L, U)$.

---

**Public-Private Gaussian Estimator.** We obtain a public-private Gaussian estimator via the following process. (1) We use the preconditioning parameters output by Algorithm 1 to recenter, then rescale our private data $X_1, \ldots, X_n$. Specifically, the transformed private data, denoted by $Y_1, \ldots, Y_n$, are given by $Y_j := \frac{1}{\sqrt{L}}\widetilde{\Sigma}^{-1/2}(X_j - \widehat{\mu})$. (2) The $Y_1, \ldots, Y_n$ are then fed as input to an existing $\operatorname{DPGaussianEstimator}$ (from Lemma A.22) for bounded Gaussians. It outputs estimates $\widehat{\mu}_Y$ and $\widehat{\Sigma}_Y$. (3) For our final estimates, we output $\widehat{\Sigma}_X := L\widehat{\Sigma}^{1/2}\Sigma_Y\widehat{\Sigma}^{1/2}$ and $\widehat{\mu}_X := \sqrt{L}\widehat{\Sigma}^{1/2}\widehat{\mu}_Y + \widehat{\mu}$. See Algorithm 2 in Appendix B for an exact description.

To establish correctness, we must ensure that, with high probability over the sampling of public data, the parameters of the true distribution underlying $Y_1, \ldots, Y_n$ indeed satisfy tight range bounds required by $\operatorname{DPGaussianEstimator}$ to provide the desired success guarantee with parameter-free private sample complexity. In other words, the mean of the transformed Gaussian must lie in a known $\operatorname{poly}(d, 1/\beta)$ ball, and the condition number of its covariance must be $\operatorname{poly}(d, 1/\beta)$.

**Lemma 2.5** (Public Data Preconditioning). *For all $\beta > 0$, there exists an algorithm that takes $d+1$ independent samples $\widetilde{X}_1, \ldots, \widetilde{X}_{d+1}$ from a Gaussian $\mathcal{N}(\mu, \Sigma)$ over $\mathbb{R}^d$, and outputs $\widehat{\mu} \in \mathbb{R}^d$, $\widehat{\Sigma} \in \mathbb{R}^{d \times d}$, $L \in \mathbb{R}$, and $U \in \mathbb{R}$, such that if $\Sigma_Y = \frac{1}{L}\widehat{\Sigma}^{-1/2}\Sigma\widehat{\Sigma}^{-1/2}$ and $\mu_Y = \frac{1}{\sqrt{L}}\widehat{\Sigma}^{-1/2}(\mu - \widehat{\mu})$, then with probability at least $1 - \beta$ over the sampling of the data, (1) $\mathbb{I} \preceq \Sigma_Y \preceq \frac{U}{L}\mathbb{I}$, and (2) $\|\mu_Y\| \leq \sqrt{\frac{U}{L}} \cdot \sqrt{5\log(3/\beta)}$, where $\frac{U}{L} = O\left(\frac{d^2\log(1/\beta)}{\beta^2}\right)$.*

*Proof.* We prove the theorem by proving the utility of Algorithm 1. Note that the quantities $L, U$, as defined in Algorithm 1, satisfy $U/L = O(d^2 \log(1/\beta)/\beta^2)$. To prove part (1), where we wish to bound the eigenvalues of $\Sigma_Y$, by the the properties of Loewner ordering, and since $\widehat{\Sigma}, \Sigma$ are symmetric, we have

$$\mathbb{I} \preceq \frac{1}{L}\widehat{\Sigma}^{-1/2}\Sigma\widehat{\Sigma}^{-1/2} \preceq \frac{U}{L}\mathbb{I} \iff L\widehat{\Sigma} \preceq \Sigma \preceq U\widehat{\Sigma} \iff L\Sigma^{-1/2}\widehat{\Sigma}\Sigma^{-1/2} \preceq \mathbb{I} \preceq U\Sigma^{-1/2}\widehat{\Sigma}\Sigma^{-1/2}.$$

Therefore, it is sufficient to prove the final inequality. Let $\widehat{\Sigma}_Z := \Sigma^{-1/2}\widehat{\Sigma}\Sigma^{-1/2}$. Then

$$\widehat{\Sigma}_Z = \frac{1}{d}\sum_{i=1}^{d+1}(\Sigma^{-1/2}(\widetilde{X}_i - \widehat{\mu}))(\Sigma^{-1/2}(\widetilde{X}_i - \widehat{\mu}))^T = \frac{1}{d}\sum_{i=1}^{d+1}(Z_i - \widehat{\mu}_Z)(Z_i - \widehat{\mu}_Z)^T$$

where for each $i \in [d+1]$, $Z_i := \Sigma^{-1/2}(X_i - \mu) \sim N(0, \mathbb{I})$ independently and $\widehat{\mu}_Z := \Sigma^{-1/2}(\widehat{\mu} - \mu) = \frac{1}{d+1}\sum_{i=1}^{d+1} Z_i$. Therefore, it suffices to show $L\widehat{\Sigma}_Z \preceq \mathbb{I}$ and $\mathbb{I} \preceq U\widehat{\Sigma}_Z$. For $i \in [d]$, let $W_i \sim \mathcal{N}(0, \mathbb{I})$ independently. Then Cochran's theorem (see Fact B.4) says that $\widehat{\Sigma}_Z$ is *identically distributed* to $\frac{1}{d}\sum_{i=1}^{d} W_i W_i^T$. From here, we can apply the bounds from Fact B.3 by noting that $\sigma_d(\widehat{\Sigma}_Z) \sim \frac{1}{d}\sigma_d([W_1, \ldots, W_d])^2$ and $\sigma_1(\widehat{\Sigma}_Z) \sim \frac{1}{d}\sigma_1([W_1, \ldots, W_d])^2$.

With probability $\geq 1 - \beta/3$, $\frac{1}{d}\sigma_d([W_1, \ldots, W_d])^2 > \frac{(\beta/3)^2}{d^2} \implies U\widehat{\Sigma}_Z \succeq \mathbb{I}$.

With probability $\geq 1 - \beta/3$, $\frac{1}{d}\sigma_1([W_1, \ldots, W_d])^2 < \frac{\left(2\sqrt{d} + \sqrt{2\log(3/\beta)}\right)^2}{d} \implies L\widehat{\Sigma}_Z \preceq \mathbb{I}$.

Taking the union bound, (1) holds with probability at least $1 - \frac{2\beta}{3}$. Next, we prove the bound on $\mu_Y$ in (2). We have

$$\mu_Y = \frac{1}{\sqrt{L}}\widehat{\Sigma}^{-1/2}(\mu - \widehat{\mu}) = \frac{1}{\sqrt{L}}\widehat{\Sigma}^{-1/2}\left(\mu - \frac{1}{d+1}\sum_{i=1}^{d+1}(\Sigma^{1/2}Z_i + \mu)\right) = -\frac{1}{\sqrt{L}}\widehat{\Sigma}^{-1/2}\Sigma^{1/2}\widehat{\mu}_Z.$$

Since $\widehat{\mu}_Z$ is identically distributed to $\frac{1}{\sqrt{d+1}}Z_1$, by concentration of the norm for multivariate Gaussians (Lemma A.3), we have that with probability $\geq 1 - \beta/3$,

$$\|\widehat{\mu}_Z\|_2 \leq \sqrt{\frac{d + 2\sqrt{d\log(3/\beta)} + 2\log(3/\beta)}{d+1}}$$

From (1), we know that $\left\|\frac{1}{L}\widehat{\Sigma}^{-1/2}\Sigma\widehat{\Sigma}^{-1/2}\right\|_2 \leq \frac{U}{L}$. Hence, $\left\|-\frac{1}{\sqrt{L}}\widehat{\Sigma}^{-1/2}\Sigma^{1/2}\right\|_2 \leq \sqrt{U/L}$. This implies that

$$\|\mu_Y\|_2 \leq \left\|-\frac{1}{\sqrt{L}}\widehat{\Sigma}^{-1/2}\Sigma^{1/2}\right\|_2 \cdot \|\widehat{\mu}_Z\|_2 \leq \sqrt{\frac{U}{L}} \cdot \sqrt{\frac{d + 2\sqrt{d\log(3/\beta)} + 2\log(3/\beta)}{d+1}}$$

which implies (2). Applying the union bound again, we have the claim. $\qquad\square$

Hence the process described above is indeed a $(d+1)$-public-sample private algorithm satisfying privacy with respect to private data $X_1, \ldots, X_n$ and the required accuracy guarantees. This follows from the guarantees of public data preconditioning (as stated in the aforementioned Lemma 2.5) combined with the guarantees of existing DPGaussianEstimator (stated in Lemma A.24). See Theorem B.5 for full details. Again, we note that our sample complexity no longer depends on the *a priori* bounds on the mean and the covariance of the unknown private data distribution.

**Different Public and Private Distributions.** Thus far, we have only considered the case where the public and the private data come from the same distribution. In the more general case where the public data $\widetilde{X}_1, \ldots, \widetilde{X}_{d+1}$ comes from a Gaussian $\widetilde{D}$ within TV distance $\gamma < 1$ of the private data distribution $D$, it can be shown that Algorithm 1 (public data preconditioning) with a slight modification will work similarly to map our unbounded problem onto a bounded one. In particular, outputting $U_\gamma = 4U/(1-\gamma)^4$ and $L_\gamma = (1-\gamma)^4 L/4$ instead of $U$ and $L$ leads to covariance and norm bounds $K = O(d^2 \log(1/\beta)/\beta^2(1-\gamma)^8)$ and $R = O(d\log(1/\beta)/\beta(1-\gamma)^{4.5})$ on the

underlying parameters of the transformed private data. This result is the consequence of a technical Lemma that translates the TV bound $\gamma$ into a bound on Gaussian parameters. Full details are in Appendix B.2.

With modified Algorithm 1, we can perform the same process of: (1) transforming private data with our public data preconditioner; (2) running an existing private algorithm requiring bounded range parameters; and (3) undoing the transformation. The guarantees of such a process are stated in Theorem 2.4. We pick up an extra $\mathrm{polylog}(1/(1-\gamma))$ factor in our sample complexity.

# 3 Estimating Gaussian Mixtures

The general process of estimating the parameters of Gaussian mixtures involves separating the mixture components by using techniques like PCA, followed by their isolation, and finally by individual component estimation. In other words, the components are first projected onto a low-dimensional subspace of $\mathbb{R}^d$, which separates the large mixture components from the rest, given that the separation in the original space is enough. The subspace is computed with respect to the available data, that is, it is chosen to be the top $k$ subspace of the data matrix, which preserves the original distances between the individual components, but shrinks down each component so that the intra-component distances are small but inter-component distances are large still. This allows us to separate a group of components from the rest. Repeating this process on each group, we can narrow down to a single Gaussian in each group, and estimate its parameters.

This has been the high-level idea in a lot of prior work, both non-private [VW02; AM05] and private [KSSU19]. In our setting, we utilise the public data available in different settings at various points to establish these tasks, while having a low cost in the sample complexity of the private data. Given a lower bound on the mixing weights ($w_{\min}$), we consider two settings in terms of the availability of the public data: (1) when there is very little available – $\widetilde{O}(1/w_{\min})$; and (2) when there is a lot more available – $\widetilde{O}(d/w_{\min})$. The key challenge, as indicated, is accurate clustering of data. Due to space constraints, the complete technical details for this section are deferred to Appendix C.

## 3.1 Public Data Sample Complexity: $\widetilde{O}(1/w_{\min})$

**Clustering.** This is the more challenging setting of the two because there is not enough public data for techniques such as PCA to give accurate results. We therefore have to rely on the private data for these tasks, and use their private counterparts to get the desired results. The entire procedure is described in detail in Algorithm 6, and its corresponding result in Theorem C.21.

1. Superclustering to reduce sensitivity for Private PCA: The main goal, as in most private estimation tasks, is to reduce the amount of noise added for privacy in order to get better accuracy. This boils down to limiting the sensitivity of the target empirical function of the data. For this, we would like to make sure that the range of the data (to which, we scale the noise) is very tight – in our context, it means that we would like to have a ball that contains a group of mixture components, which are very close to one another (a "supercluster"), having close to the smallest possible radius. We can limit the data to within this ball, and the sensitivity for the next step would be reduced. To get such a ball, we use the public data, and come up with a "superclustering" algorithm (Algorithm 3, Theorem C.14), which does exactly that. This algorithm essentially relies on the concentration properties of high-dimensional Gaussians – it finds the radius of the largest Gaussian and a point (as the centre) within that Gaussian, and additively grows the ball until it stops finding more points. This gives a ball whose radius is an $O(k)$ approximation to the radius of the largest Gaussian. We use this ball to isolate the same components in the private data, and perform private PCA to separate the components within. Getting this tight ball is crucial to reduce the cost of private PCA in terms of the sample complexity compared to the prior work of [KSSU19].

2. Private PCA: We use the well-known private PCA algorithm (Algorithm 4, Corollary C.18) that has been previously analysed in [DTTZ14; KSSU19] for this.

3. Partitioning in low dimensions: The next task is to partition the components in the low-dimensional subspace in a way that there is no overlap. For this, we use an algorithm (Algorithm 5, Theorem C.20) that is similar to the "terrific ball" algorithm of [KSSU19],

and relies upon the accuracy guarantees of the private PCA step. It uses both public and private data, and looks for a terrific ball within them. If there is one, indicating that there is more than one component in that subset of the datasets, it partitions the datasets (and the components), and queues the partitions for further work. Otherwise, it just adds that subset of the private dataset to the set of discovered clusters.

4. The whole process is repeated on all the remaining partitions.

Here, we provide some intuition about the clustering process. To explain the implication of step (1), we state an informal version of Theorem C.14 only for the first application of Algorithm 3 when it is being run initially on the whole public dataset (of size $m$).

**Theorem 3.1.** *There exists an algorithm, which if given $m \geq O\left(\frac{\ln(k/\beta)}{w_{\min}}\right)$ samples from a Gaussian mixture $\mathcal{D}$, it outputs $c \in \mathbb{R}^d$ and $R \in \mathbb{R}$, such that with probability $\geq 1 - \beta$: (1) $B_R(c)$ is non-empty and contains either $\geq 1 - \beta/k$ probability mass of a Gaussian component or $< \beta/k$; and (2) $R$ is an $O(k)$ multiplicative approximation of the radius of the largest Gaussian within $B_R(c)$.*

The above result essentially says that a supercluster is captured by this algorithm in a tight ball. Because with high probability, the points from high-dimensional Gaussians are highly concentrated, $B_R(c)$ would capture all points from both public and private datasets only corresponding to the captured components. Note that this process is performed entirely using the public dataset.

In step (2), where we perform private PCA, the radius of the ball we're limiting the points to ($B_R(c)$) is small, that is, $R$ is an $O(k)$ approximation of the radius of the largest mixture component contained within it. Therefore, while computing the noisy empirical covariance, we add a relatively small amount of noise that preserves the signal of the points from that Gaussian. This ensures that projection matrix we get preserves the inter-Gaussian distances appropriately, but significantly reduces the intra-Gaussian distances. In other words, it preserves the distance between the mean of that largest Gaussian from the means of the rest of Gaussians within $B_R(c)$, while shrinking down all the Gaussians. This helps to isolate at least one component in step (3).

Step (3) relies on the following observation about Gaussians, along with the result from step (2). In the case when there are at least two components within $B_R(c)$, step (2) ensures that in the low-dimensional subspace, there is at least one Gaussian that is far from all the others. Thus, there must be a ball that contains a large number of points, such that expanding it by a constant factor would not give us any new points (because of the available empty space), and the complement of that expanded ball would contain a large number of points. On the other hand, if there is only one component within $B_R(c)$, then projecting it to a low-dimensional subspace would never give us such a ball. Using this observation, Algorithm 5 uses both public and private datasets to determine whether there are one or more components in $B_R(c)$, and in the first case, returns a ball that partitions the datasets in a non-trivial way that no component has points in both subsets (a "clean partition"), or in the second case, returns $\perp$. The number of public samples is enough, such that in the first case, querying the projected public dataset for the existence of such a ball would give us one, but not enough to correctly reject in the second case, which is why we use the much larger private dataset to confirm whether the positive answer we get by querying the public dataset is correct or not.

In the end, the idea of the clustering algorithm is that in each iteration, it would either isolate both public and private points from a component or partition the datasets further in a clean way. In either case, we move one step closer towards obtaining $k$ clean subsets of the two datasets.

**Component Estimation.** The final task is to estimate the parameters of the $k$ isolated components (Algorithm 7, Theorem C.22). In the $(\varepsilon, \delta)$-DP setting (Theorem C.23), we simply apply the private Gaussian learner of [AL21] to each component, which enables us to learn with no dependence in the sample complexity on the range parameters of the Gaussian itself (among other available choices, were the $(\varepsilon, \delta)$-DP learners from [KMSSU21; KMV21; TCKMS21], but this one had the joint best sample complexity with [TCKMS21]). In the $\rho$-zCDP setting (Theorem C.24), we use the Gaussian learner from [KLSU19], instead. Note that in this setting, the private sample complexity does depend on the range parameters of the Gaussians, themselves.

## 3.2 Public Data Sample Complexity: $\widetilde{O}(d/w_{\min})$

**Clustering.** In the second setting, clustering is easier in terms of the tasks that need to be done privately – it can be done entirely by using the public dataset itself (Algorithm 9, Theorem C.26). This time, the number of public samples is enough to be able to accurately do PCA on the public data, hence, perform clustering using just the public data itself. This improves the private data sample complexity by removing the term due to private PCA altogether.

1. No supercustering involved: The first change in the clustering algorithm is eliminating the superclustering step.

2. PCA: We essentially get the projection matrix for the top $k$ subspace of the data using non-private PCA, and work within that subspace using the public data to further partition the two datasets (and the components).

3. Partitioning in low dimensions: The algorithm for partitioning in the low-dimensional subspace (Algorithm 8, Theorem C.25) is similar to the private partitioner used in the previous case, except that there is no need for privacy in this case.

4. The whole process is repeated on all the remaining partitions.

**Component Estimation.** Once the components are separated, as before, we individually estimate them using the partitions of the private dataset privately (Algorithm 10, Theorem C.27). In the $(\varepsilon, \delta)$-DP setting (Theorem C.28), we again use the learner from [AL21], which ensures parameter-free private data sample complexity in this process. In the $\rho$-zCDP setting though (Theorem C.29), we use our own zCDP learner from this text (Theorem 1.1), which uses the public data, as well, to ensure parameter-free private data sample complexity for this process.

## 4 Conclusion

We investigate the task of private estimation with access to a small amount of public data. Availability of some public data is a reasonable assumption in practice, therefore, we demonstrate ways to take advantage of it. We show that significant improvements can be made in terms of the private sample complexity for the tasks of estimating multivariate Gaussians and Gaussian mixtures, given a small amount of public data. For the first problem, we show that even in the case where the public data distribution differs from the private data distribution significantly, we can still estimate the private data distribution using a small number of public samples, and improve the private sample complexity by removing any dependence on the range parameters of the private data distribution, which is not possible without any public data. For the problem of estimating Gaussian mixtures under privacy constraints, our new algorithms take advantage of public data to obtain sample complexity bounds, which are significantly better than those of the best known private algorithms for this problem.

**Limitations and Open Problems.** One question in the context of Gaussian estimation is whether we could obtain computationally efficient, pure DP algorithms that use public data, and match the best known private sample complexity for the problem. Another question is whether we could polynomially improve the private sample complexity, given a much larger, but $o(d^2)$, number of public samples. A natural question to ask is whether our results for Gaussian estimation could be extended to mean and covariance estimation of heavy-tailed distributions. For estimation of Gaussian mixtures, our work focuses on the case where the public data and the private data distributions are the same, so studying this problem for the case where they are different would be an interesting challenge. Another related question is understanding other instances of clustering problems, where the results of the existing private algorithms could be improved using some additional public data.

## Acknowledgments and Disclosure of Funding

AB, GK, and VS are supported by an NSERC Discovery Grant, a University of Waterloo Startup Grant, an unrestricted gift from Google, and an unrestricted gift from Apple. AB is supported by a Vector Scholarship in Artificial Intelligence.

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
