# A  Preliminaries

## A.1  Notations

We define a few notations here to be used throughout the paper.

- Private data are denoted by $X = (X_1, \ldots, X_n)$, with each $X_j \in \mathbb{R}^d$. Public data is denoted by $\widetilde{X} = (\widetilde{X}_1, \ldots, \widetilde{X}_m)$, with each $\widetilde{X}_i \in \mathbb{R}^d$.
- We denote a ball of radius $r > 0$ centred around a point $c \in \mathbb{R}^d$ by $B_r(c)$.
- For a set $S$, we denote its power set by $\mathcal{P}(S)$.

## A.2  Useful Concentration Inequalities

We first state a multiplicative Chernoff bound.

**Lemma A.1** (Multiplicative Chernoff). *Let $X_1, \ldots, X_m$ be independent Bernoulli random variables, and let $X$ be their sum. If $p = \mathbb{E}(X_i)$, then for $0 \leq \delta_1 \leq 1$ and $\delta_2 \geq 0$,*

$$\mathbb{P}(X \leq (1 - \delta_1)pm) \leq e^{-\frac{\delta_1^2 pm}{2}}$$

*and*

$$\mathbb{P}(X \leq (1 + \delta_2)pm) \leq e^{-\frac{\delta_2^2 pm}{2 + \delta_2}}.$$

Next, we state Bernstein's inequality.

**Lemma A.2** (Bernstein's Inequality). *Let $X_1, \ldots, X_m$ be independent Bernoulli random variables. Let $p = \mathbb{E}(X_i)$. Then for $m \geq \frac{5p}{2\varepsilon^2} \ln(2/\beta)$ and $\varepsilon \leq p/4$,*

$$\mathbb{P}\left(\left|\frac{1}{m}\sum X_i - p\right| \geq \varepsilon\right) \leq 2e^{-\varepsilon^2 m/2(p+\varepsilon)} \leq \beta.$$

Now, we state the Hanson-Wright inequality about quadratic forms.

**Lemma A.3** (Hanson-Wright inequality [HW71]). *Let $X \sim \mathcal{N}(0, \mathbb{I}_{d \times d})$ and let $A$ be a $d \times d$ matrix. Then for all $t > 0$, the following two bounds hold:*

$$\mathbb{P}\left(X^T A X - \mathrm{tr}(A) \geq 2\|A\|_F \sqrt{t} + 2\|A\|_2 t\right) \leq \exp(-t);$$

$$\mathbb{P}\left(X^T A X - \mathrm{tr}(A) \leq -2\|A\|_F \sqrt{t}\right) \leq \exp(-t).$$

We mention an inequality that bounds the tails of a one-dimensional Gaussian $\mathcal{N}(\mu, \sigma^2)$.

**Lemma A.4** (1-D Gaussian Concentration). *Let $Z \sim \mathcal{N}(\mu, \sigma^2)$. Then,*

$$\mathbb{P}(|Z - \mu| \leq t\sigma) \leq 2e^{-\frac{t^2}{2}}.$$

Next, we state a concentration inequality for $0$-mean Laplace random variables.

**Lemma A.5** (Laplace Concentration). *Let $Z \sim \mathrm{Lap}(t)$. Then $\mathbb{P}(|Z| > t \cdot \ln(1/\beta)) \leq \beta$.*

We now mention an anti-concentration inequality for weighted $\chi^2$ distributions from [ZZ18]. We adjust the constants appropriately in the following theorem as per the specifications in the aforementioned article.

**Lemma A.6** (Theorem 6 from [ZZ18]). *Let $Z \sim \mathcal{N}(0, 1)$, $a > 0$, and $Y = aZ$. Then for $\tau \geq a$, we have the following.*

$$\mathbb{P}(Y \geq a + \tau) \geq 0.06 e^{-\frac{3\tau}{2a}}$$

The following are standard concentration results for the empirical mean and covariance of a set of Gaussian vectors (see, e.g., [DKKLMS16]).

**Lemma A.7.** *Let $X_1, \ldots, X_n$ be i.i.d. samples from $\mathcal{N}(0, \mathbb{I}_{d \times d})$. Then we have that*

$$\mathbb{P}\left(\left\|\frac{1}{n}\sum_{i \in [n]} X_i\right\|_2 \geq t\right) \leq 4\exp(c_1 d - c_2 n t^2);$$

$$\mathbb{P}\left(\left\|\frac{1}{n}\sum_{i \in [n]} X_i X_i^T - I\right\|_2 \geq t\right) \leq 4\exp(c_3 d - c_4 n \min(t, t^2)),$$

*where $c_1, c_2, c_3, c_4 > 0$ are some absolute constants.*

### A.3 Gaussian Mixtures

Here, we provide preliminaries for the problem of parameter estimation of mixtures of Gaussians. Let $\mathsf{Sym}_d^+$ denote set of all $d \times d$, symmetric, and positive semidefinite matrices. Let $\mathcal{G}(d) = \{\mathcal{N}(\mu, \Sigma) : \mu \in \mathbb{R}^d, \Sigma \in \mathsf{Sym}_d^+\}$ be the family of $d$-dimensional Gaussians. We can now define the class $\mathcal{G}(d, k)$ of mixtures of Gaussians as follows.

**Definition A.8** (Gaussian Mixtures)**.** The class of *Gaussian $k$-mixtures in $\mathbb{R}^d$* is

$$\mathcal{G}(d, k) := \left\{\sum_{i=1}^k w_i G_i : G_1, \ldots, G_k \in \mathcal{G}(d), w_1, \ldots, w_k > 0, \sum_{i=1}^k w_i = 1\right\}.$$

We can specify a Gaussian mixture by a set of $k$ tuples as: $\{(\mu_1, \Sigma_1, w_1), \ldots, (\mu_k, \Sigma_k, w_k)\}$, where each tuple represents the mean, covariance matrix, and mixing weight of one of its components. Additionally, for each $i$, we refer to $\sigma_i^2 = \|\Sigma_i\|_2$ as the maximum directional variance of component $i$.

We assume that the mixing weight of each component is lower bounded by $w_{\min}$. We impose a separation condition for mixtures of Gaussians to be able to learn them.

**Definition A.9** (Separated Mixtures)**.** For $s > 0$, a Gaussian mixtures $\mathcal{D} \in \mathcal{G}(d, k)$ is *$s$-separated* if

$$\forall 1 \leq i < j \leq k, \quad \|\mu_i - \mu_j\|_2 \geq \left(s + \frac{10}{\sqrt{w_i}} + \frac{10}{\sqrt{w_j}}\right) \cdot \max\{\sigma_i, \sigma_j\}.$$

We denote the family of separated Gaussian mixtures by $\mathcal{G}(d, k, s)$.

Now, we define what it means to "learn" a Gaussian mixture in our setting.

**Definition A.10** ($(\alpha, \beta)$-Learning)**.** Let $\mathcal{D} \in \mathcal{G}(d, k)$ be parameterized by $\{(\mu_1, \Sigma_1, w_1), \ldots, (\mu_k, \Sigma_k, w_k)\}$. We say that an algorithm $(\alpha, \beta)$-*learns* $\mathcal{D}$, if on being given sample-access to $\mathcal{D}$, it outputs with probablity at least $1 - \beta$ a distribution $\widehat{\mathcal{D}} \in \mathcal{G}(d, k)$ parameterised by $\{(\widehat{\mu}_1, \widehat{\Sigma}_1, \widehat{w}_1), \ldots, (\widehat{\mu}_k, \widehat{\Sigma}_k, \widehat{w}_k)\}$, such that there exists a permutation $\pi : [k] \to [k]$, for which the following conditions hold.

1. For all $1 \leq i \leq k$, $\mathrm{d}_{\mathrm{TV}}(\mathcal{N}(\mu_i, \Sigma_i), \mathcal{N}(\widehat{\mu}_{\pi(i)}, \widehat{\Sigma}_{\pi(i)})) \leq O(\alpha)$.

2. For all $1 \leq i \leq k$, $\left|w_i - \widehat{w}_{\pi(i)}\right| \leq O\left(\frac{\alpha}{k}\right)$.

Note that the above two conditions together imply that $\mathrm{d}_{\mathrm{TV}}(\mathcal{D}, \widehat{\mathcal{D}}) \leq \alpha$.

Finally, we define the "median radius" of a Gaussian.

**Definition A.11.** Let $G := \mathcal{N}(\mu, \Sigma)$ be a $d$-dimensional Gaussian, and $R > 0$. Then $R$ is called the median radius of $G$, if the $G$-measure of $B_R(\mu)$ is exactly $1/2$.

The following is an anti-concentration lemma from [AK01] about Gaussians. It lower bounds the distance of a sample from a Gaussian from any arbitrary point in space.

**Lemma A.12** (Lemma 6 from [AK01])**.** *Let $G := \mathcal{N}(\mu, \Sigma)$ be a Gaussian in $\mathbb{R}^d$ with median radius $R$, and let $\sigma^2 = \|\Sigma\|$. Suppose $z \in \mathbb{R}^d$ is an arbitrary point, and $x \sim G$. Then for all $t \geq 1$, with probability at least $1 - 2e^{-t}$,*

$$\|x - z\|^2 \geq (\max\{R - t\sigma, 0\})^2 + \|z - \mu\|^2 - 2\sqrt{2t}\sigma\|z - \mu\|.$$

## A.4 Robustness of PCA to Noise

Principal Component Analysis (PCA) is one of the main tools in learning mixtures of Gaussians. It is common to project the entire data onto the top-$k$ principal directions (a subspace, which would approximately contain all the means of the $k$ components). Ideally, PCA should eliminate directions that are not so useful, whilst maintaining the distances among the Gaussian components. This would allow us to cluster the data in low dimensions easily based on the most useful directions, while not having to worry about the rest of the directions that do not give us useful information for this process. For the purpose of doing PCA under differential privacy, we would like to have results for PCA when there is noise involved in the process of obtaining DP guarantees.

Let $X \in \mathbb{R}^{n \times d}$ be the dataset obtained from the mixture of Gaussians in question. Suppose $A \in \mathbb{R}^{n \times d}$, such that for each $i$, $A_i$ is the true mean of the Gaussian from which $X_i$ has been sampled. Let $n_j$ denote the number of points in $X$ belonging to component $j$. The following lemma gives guarantees for PCA when we have a noisy approximation to the top-$k$ subspace of the dataset.

**Lemma A.13** (Lemma 3.1 from [KSSU19]). *Let $X \in \mathbb{R}^{n \times d}$ be a collection of $n$ datapoints from $k$ clusters each centered at $\mu_1, \mu_2, ..., \mu_k$. Let $A \in \mathbb{R}^{n \times d}$ be the corresponding matrix of (unknown) centers (for each $j$ we place the center $\mu_{c(j)}$ with $c(j)$ denoting the clustering point $X_j$ belongs to). Let $\Pi_{V_k} \in \mathbb{R}^{d \times d}$ denote the $k$-PCA projection of $X$'s rows. Let $\Pi_U \in \mathbb{R}^{d \times d}$ be a projection such that for some bound $B \geq 0$ it holds that $\|X^T X - (X\Pi_U)^T (X\Pi_U)\|_2 \leq \|X^T X - (X\Pi_{V_k})^T (X\Pi_{V_k})\|_2 + B$. Denote $\bar{\mu}_i$ as the empirical mean of all points in cluster $i$ and denote $\hat{\mu}_i$ as the projection of the empirical mean $\hat{\mu}_i = \Pi_U \bar{\mu}_i$. Then*

$$\|\bar{\mu}_i - \hat{\mu}_i\|_2 \leq \tfrac{1}{\sqrt{n_i}}\|X - A\|_2 + \sqrt{\tfrac{B}{n_i}}$$

The next lemma bounds the singular values of a matrix $X$ that is sampled from a mixture of Gaussians, but centred around $A$.

**Lemma A.14** (Lemma 3.2 from [KSSU19]). *Let $X \in \mathbb{R}^{n \times d}$ be a sample from $\mathcal{D} \in \mathcal{G}(d, k)$, and let $A \in \mathbb{R}^{n \times d}$ be the matrix where each row $i$ is the (unknown) mean of the Gaussian from which $X_i$ was sampled. For each $i$, let $\sigma_i^2$ denote the maximum directional variance of component $i$, and $w_i$ denote its mixing weight. Define $\sigma^2 = \max_i\{\sigma_i^2\}$ and $w_{\min} = \min_i\{w_i\}$. If*

$$n \geq \frac{1}{w_{\min}} \left( \xi_1 d + \xi_2 \log\left(\frac{2k}{\beta}\right) \right),$$

*where $\xi_1, \xi_2$ are universal constants, then with probability at least $1 - \beta$,*

$$\frac{\sqrt{nw_{\min}}\sigma}{4} \leq \|X - A\|_2 \leq 4\sqrt{n \sum_{i=1}^{k} w_i \sigma_i^2}.$$

## A.5 Privacy Preliminaries

We start with different definitions of differential privacy.

**Definition A.15** (Differential Privacy (DP) [DMNS06]). A randomized algorithm $M : \mathcal{X}^n \to \mathcal{Y}$ satisfies $(\varepsilon, \delta)$-differential privacy ($(\varepsilon, \delta)$-DP) if for every pair of neighboring datasets $X, X' \in \mathcal{X}^n$ (i.e., datasets that differ in exactly one entry),

$$\forall Y \subseteq \mathcal{Y} \quad \mathbb{P}\left(M(X) \in Y\right) \leq e^\varepsilon \cdot \mathbb{P}\left(M(X') \in Y\right) + \delta.$$

When $\delta = 0$, we say that $M$ satisfies $\varepsilon$-differential privacy or pure differential privacy.

**Definition A.16** (Concentrated Differential Privacy (zCDP) [BS16]). A randomized algorithm $M : \mathcal{X}^n \to \mathcal{Y}$ satisfies $\rho$-zCDP if for every pair of neighboring datasets $X, X' \in \mathcal{X}^n$,

$$\forall \alpha \in (1, \infty) \quad D_\alpha\left(M(X)\|M(X')\right) \leq \rho\alpha,$$

where $D_\alpha\left(M(X)\|M(X')\right)$ is the $\alpha$-Rényi divergence between $M(X)$ and $M(X')$.[3]

---

[3] Given two probability distributions $P, Q$ over $\Omega$, $D_\alpha(P\|Q) = \frac{1}{\alpha-1} \log\left(\sum_x P(x)^\alpha Q(x)^{1-\alpha}\right)$.

Note that $(\varepsilon, 0)$-DP implies $\frac{\varepsilon^2}{2}$-zCDP, which implies $(\varepsilon\sqrt{\log(1/\delta)}, \delta)$-DP for every $\delta > 0$ [BS16]. Now, we define the notion of "public-private" algorithms that take public data samples and private data samples as input, and guarantee differential privacy with respect to the private data.

**Definition A.17** (Public-Private Algorithms). *Let $\widetilde{\mathcal{X}}$ be the domain of public data and $\mathcal{X}$ be the domain of private data. A randomized algorithm $M : \mathcal{X}^n \times \widetilde{\mathcal{X}}^m \to \mathcal{Y}$ taking in public and private data satisfies $(\varepsilon, \delta)$-DP (or $\rho$-zCDP) with respect to the private data if for any public dataset $\widetilde{X} \in \widetilde{\mathcal{X}}^m$, the resulting randomized algorithm $M(\widetilde{X}, \cdot) : \mathcal{X}^n \to \mathcal{Y}$ is $(\varepsilon, \delta)$-DP (or $\rho$-zCDP, respectively).*

These definitions of DP are closed under post-processing, and can be composed with graceful degradation of the privacy parameters.

**Lemma A.18** (Post-Processing [DMNS06; BS16]). *If $M : \mathcal{X}^n \to \mathcal{Y}$ is $(\varepsilon, \delta)$-DP, and $P : \mathcal{Y} \to \mathcal{Z}$ is any randomized function, then the algorithm $P \circ M$ is $(\varepsilon, \delta)$-DP. Similarly if $M$ is $\rho$-zCDP then the algorithm $P \circ M$ is $\rho$-zCDP.*

**Lemma A.19** (Composition of DP [DMNS06; DRV10; BS16]). *If $M$ is an adaptive composition of differentially private algorithms $M_1, \ldots, M_T$, then the following all hold:*

1. *If $M_1, \ldots, M_T$ are $(\varepsilon_1, \delta_1), \ldots, (\varepsilon_T, \delta_T)$-DP then $M$ is $(\varepsilon, \delta)$-DP for*
$$\varepsilon = \sum_t \varepsilon_t \quad and \quad \delta = \sum_t \delta_t.$$

2. *If $M_1, \ldots, M_T$ are $(\varepsilon_0, \delta_1), \ldots, (\varepsilon_0, \delta_T)$-DP for some $\varepsilon_0 \leq 1$, then for every $\delta_0 > 0$, $M$ is $(\varepsilon, \delta)$-DP for*
$$\varepsilon = \varepsilon_0 \sqrt{6T \log(1/\delta_0)} \quad and \quad \delta = \delta_0 + \sum_t \delta_t$$

3. *If $M_1, \ldots, M_T$ are $\rho_1, \ldots, \rho_T$-zCDP then $M$ is $\rho$-zCDP for $\rho = \sum_t \rho_t$.*

### A.5.1 Known Differentially Private Mechanisms

We state standard results on achieving differential privacy via noise addition proportional to the sensitivity [DMNS06].

**Definition A.20** (Sensitivity). *Let $f : \mathcal{X}^n \to \mathbb{R}^d$ be a function, its $\ell_1$-sensitivity and $\ell_2$-sensitivity are*
$$\Delta_{f,1} = \max_{X \sim X' \in \mathcal{X}^n} \|f(X) - f(X')\|_1 \quad and \quad \Delta_{f,2} = \max_{X \sim X' \in \mathcal{X}^n} \|f(X) - f(X')\|_2,$$

respectively. Here, $X \sim X'$ denotes that $X$ and $X'$ are neighboring datasets (i.e., those that differ in exactly one entry).

For functions with bounded $\ell_1$-sensitivity, we can achieve $\varepsilon$-DP by adding noise from a Laplace distribution proportional to $\ell_1$-sensitivity. For functions taking values in $\mathbb{R}^d$ for large $d$ it is more useful to add noise from a Gaussian distribution proportional to the $\ell_2$-sensitivity, to get $(\varepsilon, \delta)$-DP and $\rho$-zCDP.

**Lemma A.21** (Laplace Mechanism). *Let $f : \mathcal{X}^n \to \mathbb{R}^d$ be a function with $\ell_1$-sensitivity $\Delta_{f,1}$. Then the Laplace mechanism*
$$M(X) = f(X) + \mathrm{Lap}\left(\frac{\Delta_{f,1}}{\varepsilon}\right)^{\otimes d}$$

*satisfies $\varepsilon$-DP.*

**Lemma A.22** (Gaussian Mechanism). *Let $f : \mathcal{X}^n \to \mathbb{R}^d$ be a function with $\ell_2$-sensitivity $\Delta_{f,2}$. Then the Gaussian mechanism*
$$M(X) = f(X) + \mathcal{N}\left(0, \left(\frac{\Delta_{f,2}\sqrt{2\ln(2/\delta)}}{\varepsilon}\right)^2 \cdot \mathbb{I}_{d \times d}\right)$$

*satisfies $(\varepsilon, \delta)$-DP. Similarly, the Gaussian mechanism*
$$M_f(X) = f(X) + \mathcal{N}\left(0, \left(\frac{\Delta_{f,2}}{\sqrt{2\rho}}\right)^2 \cdot \mathbb{I}_{d \times d}\right)$$

*satisfies $\rho$-zCDP.*

Next, we mention a very basic pure DP algorithm that computes the cardinality of a dataset as a simple application of Lemma A.21.

**Lemma A.23** (PCount). *Let $X = (X_1, \ldots, X_n)$ be a set of points from some data universe $\chi$. Then for all $\varepsilon > 0$ and $0 < \beta < 1$, there exists an $\varepsilon$-DP mechanism (PCount $: \chi^* \to \mathbb{R}$) that on input $X$ outputs $n'$, such that with probability at least $1 - \beta$, $|n - n'| \leq \frac{\ln(1/\beta)}{\varepsilon}$.*

*Proof.* The algorithm is just the following. Step 1: sample $\gamma \sim \mathrm{Lap}\left(\frac{1}{\varepsilon}\right)$. Step 2: output $|X| + \gamma$.

Since the sensitivity of $|X|$ is 1, by Lemma A.21, PCount is $\varepsilon$-DP.

Next, by the guarantees of Lemma A.5, $|\gamma| \leq \frac{\ln(1/\beta)}{\varepsilon}$ with probability at least $1 - \beta$. $\qquad\square$

Now, we mention three results from prior work for learning high-dimensional Gaussians under approximate DP, $z$CDP, and pure DP constraints, respectively.

**Lemma A.24** (DPGaussianEstimator). *Let $\Sigma \in \mathbb{R}^{d \times d}$ be symmetric and positive-definite, and $\mu \in \mathbb{R}^d$. For all $0 < \alpha, \beta < 1$, if given $n_{GE}$ independent samples from $\mathcal{N}(\mu, \Sigma)$, then the following algorithms output a symmetric and positive-definite $\widehat{\Sigma} \in \mathbb{R}^{d \times d}$, and a vector $\widehat{\mu} \in \mathbb{R}^d$, such that with probability at least $1 - \beta$, $\mathrm{d}_{\mathrm{TV}}(\mathcal{N}(\mu, \Sigma), \mathcal{N}(\widehat{\mu}, \widehat{\Sigma})) \leq \alpha$.*

1. *For the $(\varepsilon, \delta)$-DP Gaussian learner from [AL21],*

$$n_{GE} = O\left(\frac{d^2 + \log\left(\frac{1}{\beta}\right)}{\alpha^2} + \frac{\left(d^2\sqrt{\log\left(\frac{1}{\delta}\right)} + d\log\left(\frac{1}{\delta}\right)\right) \cdot \mathrm{polylog}\left(d, \frac{1}{\alpha}, \frac{1}{\beta}, \frac{1}{\varepsilon}, \log\left(\frac{1}{\delta}\right)\right)}{\alpha\varepsilon}\right).$$

2. *For the $\rho$-zCDP Gaussian learner from [KLSU19], for $\mathbb{I} \preceq \Sigma \preceq K\mathbb{I}$ and $\|\mu\| \leq R$ (where $K \geq 1$ and $R > 0$),*

$$n_{GE} = O\left(\frac{d^2 + \log\left(\frac{1}{\beta}\right)}{\alpha^2} + \frac{d^2 \cdot \mathrm{polylog}\left(\frac{d}{\alpha\beta\rho}\right) + d\log\left(\frac{d\log(R)}{\alpha\beta\rho}\right)}{\alpha\sqrt{\rho}}\right.$$

$$\left. + \frac{d^{1.5}\sqrt{\log(K)} \cdot \mathrm{polylog}\left(\frac{d\log(K)}{\rho\beta}\right) + \sqrt{d\log\left(\frac{Rd}{\beta}\right)}}{\sqrt{\rho}}\right).$$

3. *For the (computationally inefficient) $\varepsilon$-DP Gaussian learner from [BKSW19], for $\mathbb{I} \preceq \Sigma \preceq K\mathbb{I}$ and $\|\mu\| \leq R$ (where $K \geq 1$ and $R > 0$),*

$$n_{GE} = O\left(\frac{d^2 + \log\left(\frac{1}{\beta}\right)}{\alpha^2} + \frac{d\log\left(\frac{dR}{\alpha}\right) + d^2\log\left(\frac{dK}{\alpha}\right) + \log\left(\frac{1}{\beta}\right)}{\alpha\varepsilon}\right).$$

## B  Estimating Gaussians

Here we present missing details from Section 2.

### B.1  Same Public and Private Distributions

#### B.1.1  Unknown Mean, Identity Covariance with 1 Public Sample

In this setting, we are given a single public sample $\widetilde{X}$, along with private samples $X_1, \ldots, X_n$, where the $\widetilde{X}$ and the $X_j$ are drawn from a $d$-dimensional Gaussian $\mathcal{N}(\mu, \mathbb{I})$ independently. Recall our observation from Section 2: a single public sample is sufficient to get a "good enough" coarse estimate of $\mu$, allowing us to apply existing private algorithms for a finer estimate. This was captured in Claim 2.1.

We restate existing pure and concentrated DP algorithms for Gaussian mean estimation below.

**Lemma B.1** (Known Private Mean Estimators for Identity Covariance Gaussians). *For all $\alpha, \beta, \varepsilon, \rho > 0$, there exist $\varepsilon$-DP and $\rho$-zCDP algorithms that take $n > 0$ samples from $\mathcal{N}(\mu, \mathbb{I})$ over $\mathbb{R}^d$, where $\|\mu\|_2 \leq R$, and output estimate $\widehat{\mu} \in \mathbb{R}^d$, such that with probability at least $1 - \beta$, $\mathrm{d}_{\mathrm{TV}}(\mathcal{N}(\mu, \mathbb{I}), \mathcal{N}(\widehat{\mu}, \mathbb{I})) \leq \alpha$, as long as the following bounds on $n$ hold.*

1. *For the $\rho$-zCDP "clip-and-noise" algorithm via the Gaussian Mechanism,*

$$n = \widetilde{O}\left(\frac{d + \log(1/\beta)}{\alpha^2} + \frac{\sqrt{d}(R + \sqrt{d})\log(1/\beta)}{\alpha\sqrt{\rho}}\right).$$

2. *For the (inefficient) $\varepsilon$-DP algorithm from [BKSW19],*

$$n = O\left(\frac{d + \log(1/\beta)}{\alpha^2} + \frac{d\log(dR/\alpha) + \log(1/\beta)}{\alpha\varepsilon}\right).$$

3. *For the $\varepsilon$-DP algorithm from [HKM22],*

$$n = \widetilde{O}\left(\frac{d + \log(1/\beta)}{\alpha^2} + \frac{d + \log(1/\beta)}{\alpha^2\varepsilon} + \frac{d\log R + \min\{d, \log R\} \cdot \log(1/\beta)}{\varepsilon}\right).$$

In the above, the "clip-and-noise" mechanism simply clips all the points in the dataset to within radius $\lambda = R + \sqrt{d + 2\sqrt{d\log(n/\beta)} + 2\log(n/\beta)}$ around the origin, and outputs the noisy empirical mean of the clipped points via the Gaussian mechanism. Note that the $\ell_2$ sensitivity of the empirical mean is now $\frac{2\lambda}{n}$, therefore, the magnitude of the noise vector added to the empirical mean is at most $\widetilde{O}\left(\frac{\sqrt{d}\lambda}{\sqrt{\rho}n}\right)$ with high probability, which is less than $\alpha$ when $n \geq \widetilde{O}\left(\frac{\sqrt{d}\lambda}{\sqrt{\rho}\alpha}\right)$. This is $\rho$-zCDP from Lemma A.22.

Also, in part (3) of the above lemma, the third term of the sample complexity arises when the algorithm tries to find a coarse estimate of the mean, that is, when it attempts to get an estimate that lies within $O(\sqrt{d})$ of the true mean (Theorem 5.1 of [HKM22]). If we already have such a coarse estimate with us (which we will), then that term would disappear, and we would simply have to apply Theorem 6.1 from [HKM22] to get the first term.

Claim 2.1 combined with the results above, yields 1-public-sample, private mean estimation algorithms.

**Theorem B.2** (Public-Private Gaussian Mean Estimation). *For all $\alpha, \beta, \varepsilon, \rho > 0$, there exist $\varepsilon$-DP and $\rho$-zCDP algorithms that take $n > 0$ private samples $X = (X_1, \ldots, X_n)$ and 1 public sample $\widetilde{X}$ from $\mathcal{N}(\mu, \mathbb{I})$ over $\mathbb{R}^d$, are private with respect to the private samples $X$, and output estimate $\widehat{\mu} \in \mathbb{R}^d$, such that with probability at least $1 - \beta$, $\mathrm{d}_{\mathrm{TV}}(\mathcal{N}(\mu, \mathbb{I}), \mathcal{N}(\widehat{\mu}, \mathbb{I})) \leq \alpha$, as long as the following bounds on $n$ hold.*

1. *For the $\rho$-zCDP "clip-and-noise" algorithm via the Gaussian Mechanism and 1 public sample,*

$$n = \widetilde{O}\left(\frac{d + \log\left(\frac{1}{\beta}\right)}{\alpha^2} + \frac{d\log^{1.5}\left(\frac{1}{\beta}\right)}{\alpha\sqrt{\rho}}\right).$$

2. *For the (inefficient) $\varepsilon$-DP algorithm via [BKSW19] and 1 public sample,*

$$n = O\left(\frac{d + \log\left(\frac{1}{\beta}\right)}{\alpha^2} + \frac{d\log\left(\frac{d\log(1/\beta)}{\alpha}\right) + \log\left(\frac{1}{\beta}\right)}{\alpha\varepsilon}\right).$$

3. *For the (efficient) $\varepsilon$-DP algorithm via [HKM22] and 1 public sample,*

$$n = \widetilde{O}\left(\frac{d + \log\left(\frac{1}{\beta}\right)}{\alpha^2} + \frac{d + \log\left(\frac{1}{\beta}\right)}{\alpha^2\varepsilon}\right).$$

*Proof.* For concreteness, we look at (1), however an analogous argument suffices for (2) and (3). Say we draw our one public sample $\widetilde{X}$, and form a shifted private dataset $Y_1, \ldots, Y_n$, each $Y_j = X_j - \widetilde{X}$. Define $\lambda = \sqrt{d + 2\sqrt{d\log(2/\beta)} + 2\log(2/\beta)}$ and $\mu_Y := \mathbb{E}(Y_1) = \mu - \widetilde{X}$. By Claim 2.1, we have that with probability $\geq 1 - \beta/2$ over the sampling of $\widetilde{X}$, $\|\mu_Y\|_2 \leq \lambda$. Hence, we set $R = \lambda$ and target failure probability $\beta/2$, and run the zCDP mean estimation algorithm from Lemma B.1 on $Y_1, \ldots, Y_n$. Suppose the output of the algorithm is $\widehat{\mu}_Y$. Then we return $\widehat{\mu} = \widehat{\mu}_Y + \widetilde{X}$.

By union bound, with probability $\geq 1 - \beta$, we have that our private data distribution satisfies the range requirements for our private algorithm's guarantees to hold *and* that our private algorithm succeeds. In this case, we have $\|\mu - \widehat{\mu}\|_2 = \|(\mu - \widetilde{X}) - (\widehat{\mu} - \widetilde{X})\|_2 = \|\mu_Y - \widehat{\mu}_Y\|_2$, so $\mathrm{d_{TV}}(\mathcal{N}(\widehat{\mu}, \mathbb{I}), \mathcal{N}(\mu, \mathbb{I})) = \mathrm{d_{TV}}(\mathcal{N}(\widehat{\mu}_Y, \mathbb{I}), \mathcal{N}(\mu_Y, \mathbb{I})) \leq \alpha$. Plugging in $R = \lambda$ and target failure probability $\beta/2$ into the algorithm's sample complexity in Lemma B.1 gives us the desired result. As mentioned earlier, if we are to use part (3) of Lemma B.1, then the third term of the sample complexity would disappear because the public sample already gives us an $O(\sqrt{d})$ estimate of $\mu$.

Notice that $\widehat{\mu}_Y$ is $\rho$-zCDP with respect to $Y_1, \ldots, Y_n$, which implies that $\widehat{\mu}$ is $\rho$-zCDP *with respect to private data* $X_1, \ldots, X_n$. To see why, note that for any fixed $\widetilde{X}$, the private algorithm from Lemma B.1 is robust to arbitrary change in $Y_j$, therefore, to any change in $X_j$ because each $X_j$ maps to exactly one $Y_j$. By the post-processing guarantee of zCDP (Lemma A.18), $\widehat{\mu}$ is also private with respect to $X$. $\qquad\square$

We remark that the above 1-public-sample private mean estimators can also be used to estimate the mean of a Gaussian with arbitrary known covariance, by reducing to the identity covariance case via rescaling with the known covariance.

### B.1.2 Unknown Mean and Covariance with $d + 1$ Public Samples

In this setting, we are given $d + 1$ public samples $\widetilde{X}_1, \ldots, \widetilde{X}_{d+1}$ and $n$ private samples $X_1, \ldots, X_n$, where all the $\widetilde{X}_i$ and $X_j$ are drawn from an unknown, $d$-dimensional Gaussian $\mathcal{N}(\mu, \Sigma)$ independently. Public data is used to transform private samples via "public data preconditioning" (Algorithm 1, as discussed in Section 2), which reduces the estimation problem to the bounded case which can be solved using the existing private algorithms.

Algorithm 2 gives the precise description of how we can use Algorithm 1 and existing private Gaussian estimators to obtain a public-private Gaussian estimator.

---

**Algorithm 2:** Public-Private Gaussian Estimator $\mathrm{PubDPGaussianEstimator}_{\alpha, \beta, \mathrm{PrivParams}}(\widetilde{X}, X)$

**Input:** Public samples $\widetilde{X} = (\widetilde{X}_1, \ldots, \widetilde{X}_{d+1})$. Private samples $X = (X_1, \ldots, X_n)$. Error tolerance $\alpha > 0$, failure probability $\beta > 0$. Privacy parameters $\mathrm{PrivParams} \subset \mathbb{R}$.
**Output:** $\widehat{\mu}_X \in \mathbb{R}^d, \widehat{\Sigma}_X \in \mathbb{R}^{d \times d}$

// Precondition the private data using the public dataset.
$(\widehat{\mu}, \widehat{\Sigma}, L, U) \leftarrow \mathrm{PubPreconditioner}_{\frac{\beta}{2}}(\widetilde{X})$.
**For** $j \in [n]$
$\quad$ Set $Y_j \leftarrow \frac{1}{\sqrt{L}} \widehat{\Sigma}^{-1/2}(X_j - \widehat{\mu})$.

// Set range parameters for private applications.

$$R = \sqrt{U/L} \cdot \sqrt{\log(6/\beta)} \quad \text{and} \quad K = U/L$$

// Estimate Gaussian using private data and private algorithm.
Set $Y \leftarrow (Y_1, \ldots, Y_n)$ and $(\widehat{\mu}_Y, \widehat{\Sigma}_Y) \leftarrow \mathrm{DPGaussianEstimator}_{\alpha, \frac{\beta}{2}, \mathrm{PrivParams}, R, K}(Y)$.
Set

$$\widehat{\mu}_X \leftarrow \sqrt{L} \widehat{\Sigma}^{1/2} \widehat{\mu}_Y + \widehat{\mu} \quad \text{and} \quad \widehat{\Sigma}_X \leftarrow L \widehat{\Sigma}^{1/2} \widehat{\Sigma}_Y \widehat{\Sigma}^{1/2}$$

**Return** $\widehat{\mu}_X, \widehat{\Sigma}_X$.

---

The proof of the correctness of Algorithm 2 relies on proof of correctness for public data pre-conditioning, which was given in Section 2. The following facts were used for the proof in that section.

**Fact B.3** (Singular Values of Gaussian Matrices [SST06]). *Let $Z \in \mathbb{R}^{d \times d}$ be a matrix with each $Z_{ij} \sim N(0,1)$ independently. Denote by $\sigma_d(Z)$ the smallest singular value of Z, and by $\sigma_1(Z)$ its largest singular value. Then we have the following.*

*1.* $\mathbb{P}\left(|\sigma_d(Z)| \leq \beta/\sqrt{d}\right) \leq \beta$

*2.* $\mathbb{P}\left(|\sigma_1(Z)| \geq 2\sqrt{d} + \sqrt{2\log(1/\beta)}\right) \leq \beta$

**Fact B.4** (Properties of Wishart Distribution[4]). *Let $Z_1, \ldots, Z_{m+1}$, $W_1, \ldots, W_m$ be chosen i.i.d. from $\mathcal{N}(0, \mathbb{I})$ over $\mathbb{R}^d$. Suppose $\widehat{\mu} = \frac{1}{m+1} \sum_{i=1}^{m+1} Z_i$. Then*

$$\frac{1}{m} \sum_{i=1}^{m+1} (Z_i - \widehat{\mu})(Z_i - \widehat{\mu})^T \sim \frac{1}{m} \sum_{i=1}^{m} W_i W_i^T \sim \frac{1}{m} \mathcal{W}_d(m, \mathbb{I}).$$

*In other words, the two quantities are identically distributed according to the scaled, d-dimensional Wishart distribution with $m$ degrees of freedom.*

Below is the same-distribution version of Theorem 2.4, which employs Lemma 2.5 from Section 2.

**Theorem B.5** (Public-Private Gaussian Estimation). *For all $\alpha, \beta, \varepsilon, \rho > 0$, there exist $\varepsilon$-DP and $\rho$-zCDP algorithms that take $d + 1$ public samples $\widetilde{X} = (\widetilde{X}_1, \ldots, \widetilde{X}_{d+1})$ and $n$ private samples $X = (X_1, \ldots, X_n)$ from an unknown, d-dimensional Gaussian $\mathcal{N}(\mu, \Sigma)$, and are private with respect to the private samples, and return $\widehat{\mu}_X \in \mathbb{R}^d$ and $\widehat{\Sigma}_X \in \mathbb{R}^{d \times d}$, such that $d_{\mathrm{TV}}(\mathcal{N}(\widehat{\mu}_X, \widehat{\Sigma}_X), \mathcal{N}(\mu, \Sigma)) \leq \alpha$ with probability at least $1 - \beta$, as long as the following bounds on $n$ hold.*

*1. For $\rho$-zCDP (with computational efficiency) via [KLSU19] and $d + 1$ public samples,*

$$n = O\left(\frac{d^2 + \log\left(\frac{1}{\beta}\right)}{\alpha^2} + \frac{d^2 \cdot \mathrm{polylog}\left(\frac{d}{\alpha\beta\rho}\right)}{\alpha\sqrt{\rho}}\right).$$

*2. For $\varepsilon$-DP (without computational efficiency) via [BKSW19] and $d + 1$ public samples,*

$$n = O\left(\frac{d^2 + \log\left(\frac{1}{\beta}\right)}{\alpha^2} + \frac{d^2 \log\left(\frac{d}{\alpha\beta}\right)}{\alpha\varepsilon}\right).$$

*Proof.* We prove the privacy and the utility guarantees for Algorithm 2. In this proof, we will work in the $\rho$-zCDP regime (where, PrivParams $= \{\rho\}$) using the Gaussian estimator of [KLSU19] (the second algorithm in Lemma A.24), and the analogous guarantees in the $\varepsilon$-DP case (where PrivParams $= \{\varepsilon\}$) would follow by the same argument (but by using the third algorithm from Lemma A.24, instead).

We start by proving the utility guarantees first. Applying our public data preconditioner (Algorithm 1 with public data $\widetilde{X}$ as input) on $X$ with target failure probability $\beta/2$ yields $Y_1, \ldots, Y_n$, such that for each $j \in [n]$, $Y_j \sim \mathcal{N}(\mu_Y, \Sigma_Y)$, where $\mu_Y$ and $\Sigma_Y$ are quantites as defined in Lemma 2.5. Then with probability $\geq 1 - \beta/2$ over the sampling of public data, we have $\mathbb{I} \preceq \Sigma_Y \preceq \frac{U}{L}\mathbb{I}$ and $\|\mu_Y\|_2 \leq \sqrt{U/L} \cdot \sqrt{5\log(6/\beta)}$ (Lemma 2.5). Hence, we can set $K = U/L = O(d^2 \log(1/\beta)/\beta^2)$, and $R = \sqrt{U/L} \cdot \sqrt{5\log(6/\beta)} = O(d\log(1/\beta)/\beta)$, and run the $\rho$-zCDP Gaussian estimator on $Y_1, \ldots, Y_n$ with target failure probability $\beta/2$. We obtain our private sample complexity, which is now independent of the range parameters of the underlying distribution, by plugging in these values into the private Gaussian estimator's sample complexity.

---

[4]See Theorem 6 from `https://www.stat.pitt.edu/sungkyu/course/2221Fall13/lec2.pdf`.

Under these parameter settings and sample complexity, by Lemma A.24 and the union bound, we have that with probability $\geq 1 - \beta$, the algorithm succeeds in outputting $\widehat{\mu}_Y$ and $\widehat{\Sigma}_Y$, such that $d_{TV}(\mathcal{N}(\widehat{\mu}_Y, \widehat{\Sigma}_Y), \mathcal{N}(\mu_Y, \Sigma_Y)) \leq \alpha$. We output the estimates $\widehat{\Sigma}_X := L\widehat{\Sigma}^{1/2}\widehat{\Sigma}_Y\widehat{\Sigma}^{1/2}$ and $\widehat{\mu}_X := \sqrt{L}\widehat{\Sigma}^{1/2}\widehat{\mu}_Y + \widehat{\mu}$. Denoting $A := \frac{1}{L}\widehat{\Sigma}^{-1}$, by the properties of the Mahalanobis norm, $\|\cdot\|_\Sigma$:

$$\|\widehat{\Sigma}_X - \Sigma\|_\Sigma = \|A^{1/2}\widehat{\Sigma}_X A^{1/2} - A^{1/2}\Sigma A^{1/2}\|_{A^{1/2}\Sigma A^{1/2}} = \|\widehat{\Sigma}_Y - \Sigma_Y\|_{\Sigma_Y}$$

$$\|\widehat{\mu}_X - \mu\|_\Sigma = \|A^{1/2}\widehat{\mu}_X - A^{1/2}\mu\|_{A^{1/2}\Sigma A^{1/2}}$$
$$= \|(\widehat{\mu}_Y + A^{1/2}\widehat{\mu}) - (\mu_Y + A^{1/2}\widehat{\mu})\|_{\Sigma_Y}$$
$$= \|\widehat{\mu}_Y - \mu_Y\|_{\Sigma_Y}$$

which implies that $d_{TV}(\mathcal{N}(\widehat{\mu}_X, \widehat{\Sigma}_X), \mathcal{N}(\mu, \Sigma)) = d_{TV}(\mathcal{N}(\widehat{\mu}_Y, \widehat{\Sigma}_Y), \mathcal{N}(\mu_Y, \Sigma_Y)) \leq \alpha$.

To argue about privacy, note that releasing $\widehat{\mu}_X$ and $\widehat{\Sigma}_X$ is $\rho$-zCDP with respect to $Y$, since it is post-processing (Lemma A.18) of the output $(\widehat{\mu}_Y, \widehat{\Sigma}_Y)$ of a $\rho$-zCDP algorithm using public information. To argue about the $\rho$-zCDP of our algorithm *with respect to the private dataset $X$*, note that for any fixed public dataset $\widetilde{X}$, the application of DPGaussianEstimator from Lemma A.24 is robust to changing one sample $Y_j$ arbitrarily. Since each $X_j$ maps to exactly one $Y_j$, DPGaussianEstimator is robust against changing any $X_j$ arbitrarily, as well. This gives us the final privacy guarantee with respect to $X$. $\qquad\square$

## B.2 Different Public Data and Private Data Distributions

In this section, we give results for the scenario, where the public data does not come from the same distribution as that of the private data. Specifically, we show that in the case, where our public data comes from another Gaussian within TV distance $\gamma$ of our private data distribution, a slight modification of Algorithm 1 (public data preconditioning) will also work to reduce our unbounded private estimation problem to a bounded one.

### B.2.1 Technical Lemmata

We start by stating results about different distance metrics for distributions. Let $d_H(\cdot, \cdot)$ denote the Hellinger distance between two distributions, and let $d_{TV}(\cdot, \cdot)$ denote their total variation (TV) distance. We start with a known fact about the relation between $d_H(\cdot, \cdot)^2$ and $d_{TV}(\cdot, \cdot)$.

**Lemma B.6** (Hellinger Distance vs. TV Distance). *Let $P, Q$ be distributions over $R^d$. Then $d_H(P, Q)^2 \leq d_{TV}(P, Q)$.*

Next, we state the expression for Hellinger distance between two univariate Gaussians.

**Lemma B.7** (Hellinger Distance for Gaussians). *Let $G_1 \equiv \mathcal{N}(\mu_1, \sigma_1^2), G_2 \equiv \mathcal{N}(\mu_2, \sigma_2^2)$ be Gaussians over $\mathbb{R}$. Then*

$$d_H(G_1, G_2)^2 = 1 - \sqrt{\frac{2\sigma_1\sigma_2}{\sigma_1^2 + \sigma_2^2}} \cdot e^{-\frac{(\mu_1 - \mu_2)^2}{4(\sigma_1^2 + \sigma_2^2)}}.$$

Now, we lower-bound the Hellinger distance between two univariate Gaussians.

**Lemma B.8** (Hellinger Distance Lower Bounds). *Let $G_1 \equiv \mathcal{N}(\mu_1, \sigma_1^2), G_2 \equiv \mathcal{N}(\mu_2, \sigma_2^2)$ be Gaussians over $\mathbb{R}$. Suppose $\sigma_{\max} = \max\{\sigma_1, \sigma_2\}$. Then $d_H(G_1, G_2)^2 \geq d_H(\mathcal{N}(\mu_1, \sigma_{\max}^2), \mathcal{N}(\mu_2, \sigma_{\max}^2))^2$ and $d_H(G_1, G_2)^2 \geq d_H(\mathcal{N}(0, \sigma_1^2), \mathcal{N}(0, \sigma_2^2))^2$.*

*Proof.* Using Lemma B.7, we have the following.

$$d_H(G_1, G_2)^2 = 1 - \sqrt{\frac{2\sigma_1\sigma_2}{\sigma_1^2 + \sigma_2^2}} \cdot e^{-\frac{(\mu_1 - \mu_2)^2}{4(\sigma_1^2 + \sigma_2^2)}}$$
$$\geq 1 - e^{-\frac{(\mu_1 - \mu_2)^2}{4(\sigma_1^2 + \sigma_2^2)}}$$
$$\geq 1 - e^{-\frac{(\mu_1 - \mu_2)^2}{8\sigma_{\max}^2}}$$

$$= d_H(\mathcal{N}(\mu_1, \sigma_{\max}^2), \mathcal{N}(\mu_2, \sigma_{\max}^2))^2$$

In the above, the second line follows from AM-GM inequality, that is, $\frac{\sigma_1^2 + \sigma_2^2}{2} \geq \sigma_1 \sigma_2$. This proves the first part. Now, we prove the second part.

$$d_H(G_1, G_2)^2 = 1 - \sqrt{\frac{2\sigma_1\sigma_2}{\sigma_1^2 + \sigma_2^2}} \cdot e^{-\frac{(\mu_1 - \mu_2)^2}{4(\sigma_1^2 + \sigma_2^2)}}$$

$$\geq 1 - \sqrt{\frac{2\sigma_1\sigma_2}{\sigma_1^2 + \sigma_2^2}}$$

$$= d_H(\mathcal{N}(0, \sigma_1^2), \mathcal{N}(0, \sigma_2^2))^2$$

In the above, the second line follows from the fact that $e^{-x} \leq 1$ for $x \geq 0$. This completes our proof. $\qquad\square$

The next lemma describes the relation between the parameters of two univariate Gaussians when their TV distance is upper-bounded.

**Lemma B.9** (TV Distance and Gaussian Parameters). *Let $G_1 \equiv \mathcal{N}(\mu_1, \sigma_1^2), G_2 \equiv \mathcal{N}(\mu_2, \sigma_2^2)$ be Gaussians over $\mathbb{R}$. Suppose $\sigma_{\max} = \max\{\sigma_1, \sigma_2\}$ and $\sigma_{\min} = \min\{\sigma_1, \sigma_2\}$. If $d_{TV}(G_1, G_2) \leq \alpha$, then*

$$\frac{(\mu_2 - \mu_1)^2}{\sigma_{\max}^2} \leq \frac{8\alpha}{1 - \alpha} \quad and \quad \frac{\sigma_{\max}}{\sigma_{\min}} \leq \frac{2}{(1 - \alpha)^2}.$$

*Proof.* We have the following from Lemmata B.6 and B.8.

$$\alpha \geq d_{TV}(G_1, G_2)$$

$$\geq d_H(\mathcal{N}(\mu_1, \sigma_{\max}^2), \mathcal{N}(\mu_2, \sigma_{\max}^2))^2$$

$$= 1 - e^{-\frac{(\mu_1 - \mu_2)^2}{8\sigma_{\max}^2}}$$

$$\iff e^{-\frac{(\mu_1 - \mu_2)^2}{8\sigma_{\max}^2}} \geq 1 - \alpha$$

$$\iff \frac{(\mu_2 - \mu_1)^2}{\sigma_{\max}^2} \leq 8 \ln\left(\frac{1}{1 - \alpha}\right)$$

$$\leq \frac{8\alpha}{1 - \alpha}$$

The last line holds because $\frac{1}{1-\alpha} = 1 + \frac{\alpha}{1-\alpha}$, and $x \geq \ln(1 + x)$ for all $x \in \mathbb{R}$.

For the second part, we apply Lemmata B.6 and B.8 again to get the following.

$$\alpha \geq d_{TV}(G_1, G_2)$$

$$\geq d_H(\mathcal{N}(0, \sigma_1^2), \mathcal{N}(0, \sigma_2^2))^2$$

$$= 1 - \sqrt{\frac{2\sigma_1\sigma_2}{\sigma_1^2 + \sigma_2^2}}$$

$$\iff \frac{1}{(1 - \alpha)^2} \geq \frac{\sigma_1^2 + \sigma_2^2}{2\sigma_1\sigma_2}$$

$$\iff \frac{2}{(1 - \alpha)^2} \geq \frac{\sigma_1}{\sigma_2} + \frac{\sigma_2}{\sigma_1}$$

$$= \frac{\sigma_{\max}}{\sigma_{\min}} + \frac{\sigma_{\min}}{\sigma_{\max}}$$

$$\implies \frac{2}{(1 - \alpha)^2} \geq \frac{\sigma_{\max}}{\sigma_{\min}}$$

$\qquad\square$

Finally, we state a multivariate analogue of Lemma B.9.

**Lemma B.10.** *Suppose* $d_{\mathrm{TV}}(\mathcal{N}(\mu, \Sigma), \mathcal{N}(\widetilde{\mu}, \widetilde{\Sigma})) \leq \gamma$ *for some* $\gamma < 1$. *Then*

1. $\frac{(1-\gamma)^4}{4} \widetilde{\Sigma} \preceq \Sigma \preceq \frac{4}{(1-\gamma)^4} \widetilde{\Sigma}$

2. $(\mu - \widetilde{\mu})(\mu - \widetilde{\mu})^T \preceq \frac{8\gamma}{1-\gamma}(\Sigma + \widetilde{\Sigma})$

*Proof.* Let $G_1 \equiv \mathcal{N}(\mu, \Sigma)$ and $G_2 \equiv \mathcal{N}(\widetilde{\mu}, \widetilde{\Sigma})$. For any unit vector $v \in \mathbb{R}^d$, denote by $v^T G_i$ the distribution over $\mathbb{R}$ obtained by sampling $x \sim G_i$ and outputting $v^T x$. A data-processing inequality for the TV distance gives us that, for any unit vector $v \in \mathbb{R}^d$,

$$d_{\mathrm{TV}}(\mathcal{N}(v^T \mu, v^T \Sigma v), \mathcal{N}(v^T \widetilde{\mu}, v^T \widetilde{\Sigma} v)) = d_{\mathrm{TV}}(v^T G_1, v^T G_2) \leq d_{\mathrm{TV}}(G_1, G_2)$$

where the first equality comes from the fact that the projection of a Gaussian is also Gaussian, with the above parameters.

By the second univariate bound in Lemma B.9, we have that for every unit vector $v \in \mathbb{R}^d$,

$$\frac{(1-\gamma)^4}{4} \leq \frac{v^T \Sigma v}{v^T \widetilde{\Sigma} v} \leq \frac{4}{(1-\gamma)^4}.$$

Rearranging the above gives us (1).

For (2), we have that for every unit vector $v \in \mathbb{R}^d$,

$$\frac{v^T (\widetilde{\mu} - \mu)(\widetilde{\mu} - \mu)^T v}{v^T (\Sigma + \widetilde{\Sigma}) v} = \frac{(v^T \widetilde{\mu} - v^T \mu)^2}{v^T \Sigma v + v^T \widetilde{\Sigma} v} \leq \frac{(v^T \widetilde{\mu} - v^T \mu)^2}{\max\{v^T \Sigma v, v^T \widetilde{\Sigma} v\}} \leq \frac{8\gamma}{1-\gamma}$$

where the last inequality comes from applying the first univariate bound in Lemma B.9. Rearranging the above gives us (2). $\qquad \square$

### B.2.2 Unknown Mean and Covariance with $d + 1$ Public Samples

Let $L, U$ be quantities, as defined in Algorithm 1, and let $L_\gamma = \frac{(1-\gamma)^4}{4} \cdot L$ and $U_\gamma = \frac{4}{(1-\gamma)^4} \cdot U$ for $0 \leq \gamma < 1$. The following is an analogue of Lemma 2.5 for the case, where we apply Algorithm 1 (public data preconditioning) with public data that comes from a Gaussian that is at most $\gamma$-far in TV distance from the private data distribution.

**Lemma B.11** ($\gamma$-Far Public Data Preconditioning)**.** *For all $\beta > 0$, there exists an algorithm that takes $d + 1$ independent samples $\widetilde{X} = (\widetilde{X}_1, \ldots, \widetilde{X}_{d+1})$ from a Gaussian $\mathcal{N}(\widetilde{\mu}, \widetilde{\Sigma})$ over $\mathbb{R}^d$, and outputs $\widehat{\mu} \in \mathbb{R}^d$, $\widehat{\Sigma} \in \mathbb{R}^{d \times d}$, $L_\gamma \in \mathbb{R}$, and $U_\gamma \in \mathbb{R}$, such that for a Gaussian $\mathcal{N}(\mu, \Sigma)$ over $\mathbb{R}^d$ with $0 \leq d_{\mathrm{TV}}(\mathcal{N}(\mu, \Sigma), \mathcal{N}(\widetilde{\mu}, \widetilde{\Sigma})) \leq \gamma < 1$, if $\Sigma_Y = \frac{1}{L_\gamma} \widehat{\Sigma}^{-1/2} \Sigma \widehat{\Sigma}^{-1/2}$ and $\mu_Y = \frac{1}{\sqrt{L_\gamma}} \widehat{\Sigma}^{-1/2}(\mu - \widehat{\mu})$, then with probability at least $1 - \beta$ over the sampling of the data,*

1. $\mathbb{I} \preceq \Sigma_Y \preceq \frac{U_\gamma}{L_\gamma} \mathbb{I}$

2. $\|\mu_Y\|_2 \leq \sqrt{\frac{U_\gamma}{L_\gamma}} \cdot \left( \sqrt{\frac{10\gamma}{1-\gamma}} + \sqrt{5 \log(3/\beta)} \right)$

*where* $\frac{U_\gamma}{L_\gamma} = O\left( \frac{d^2 \log(1/\beta)}{\beta^2 (1-\gamma)^8} \right)$.

*Proof.* We prove the lemma by proving the utility of a modified version of Algorithm 1, which returns $L_\gamma$ and $U_\gamma$, instead of $L$ and $U$, respectively. The result follows from tracing through the proof of Lemma 2.5, and applying Lemma B.10 as necessary. We highlight the differences.

We start with (1). By the same chain of equivalences in the proof of Lemma 2.5, it suffices to show that $L_\gamma \Sigma^{-1/2} \widehat{\Sigma} \Sigma^{-1/2} \preceq \mathbb{I} \preceq U_\gamma \Sigma^{-1/2} \widehat{\Sigma} \Sigma^{-1/2}$. We have the following.

$$\Sigma^{-1/2} \widehat{\Sigma} \Sigma^{-1/2} = \Sigma^{-1/2} \left( \frac{1}{d} \sum_{i=1}^{d+1} ((\widetilde{\Sigma}^{1/2} Z_i + \widetilde{\mu}) - \widehat{\mu})((\widetilde{\Sigma}^{1/2} Z_i + \widetilde{\mu}) - \widehat{\mu})^T \right) \Sigma^{-1/2}$$

$$= \Sigma^{-1/2} \left( \frac{1}{d} \sum_{i=1}^{d+1} (\widetilde{\Sigma}^{1/2}(Z_i - \widehat{\mu}_Z))(\widetilde{\Sigma}^{1/2}(Z_i - \widehat{\mu}_Z))^T \right) \Sigma^{-1/2}$$

$$= \Sigma^{-1/2} \widetilde{\Sigma}^{1/2} \widehat{\Sigma}_Z \widetilde{\Sigma}^{1/2} \Sigma^{-1/2}$$

In the above, $Z_i$, $\widehat{\mu}_Z$, and $\widehat{\Sigma}_Z$ are quantities, as defined in the proof of Lemma 2.5. From the same proof, we know that with probability $\geq 1 - \beta/3$, we have $U\widehat{\Sigma}_Z \succeq \mathbb{I}$, which implies that

$$U\Sigma^{-1/2}\widetilde{\Sigma}^{1/2}\widehat{\Sigma}_Z\widetilde{\Sigma}^{1/2}\Sigma^{-1/2} \succeq \Sigma^{-1/2}\widetilde{\Sigma}\Sigma^{-1/2} \succeq \frac{(1-\gamma)^4}{4} \cdot \mathbb{I}$$

where the last inequality follows from (1) in Lemma B.10. Recalling that we set $U_\gamma = \frac{4}{(1-\gamma)^4} \cdot U$, and rearranging gives us that $U_\gamma \Sigma^{-1/2}\widehat{\Sigma}\Sigma^{-1/2} \succeq \mathbb{I}$, as desired. Similarly, with probability $\geq 1 - \beta/3$, $L\widehat{\Sigma}_Z \preceq \mathbb{I}$, which implies that

$$L\Sigma^{-1/2}\widetilde{\Sigma}^{1/2}\widehat{\Sigma}_Z\widetilde{\Sigma}^{1/2}\Sigma^{-1/2} \preceq \Sigma^{-1/2}\widetilde{\Sigma}\Sigma^{-1/2} \preceq \frac{4}{(1-\gamma)^4} \cdot \mathbb{I}$$

which again, after rearranging, allows us to conclude $L_\gamma \Sigma^{-1/2}\widehat{\Sigma}\Sigma^{-1/2} \preceq \mathbb{I}$.

It remains to verify that (2) holds. Write

$$\mu_Y = \frac{1}{\sqrt{L_\gamma}}\widehat{\Sigma}^{-1/2}(\mu-\widehat{\mu}) = \frac{1}{\sqrt{L_\gamma}}\widehat{\Sigma}^{-1/2}(\mu-\widetilde{\mu}-\widetilde{\Sigma}^{1/2}\widehat{\mu}_Z) = \frac{1}{\sqrt{L_\gamma}}\widehat{\Sigma}^{-1/2}(\mu-\widetilde{\mu}) - \frac{1}{\sqrt{L_\gamma}}\widehat{\Sigma}^{-1/2}\widetilde{\Sigma}^{1/2}\widehat{\mu}_Z.$$

We bound the two terms separately. Note that the second term appears in the proof of Lemma 2.5. By the same argument as in that proof, we claim that with probability $\geq 1 - \frac{\beta}{3}$, $\|\widehat{\mu}_Z\|_2 \leq \sqrt{\frac{d+2\sqrt{d\log(3/\beta)}+2\log(3/\beta)}{d+1}}$. We already know that $\left\|\widehat{\Sigma}^{-1/2}\widetilde{\Sigma}^{1/2}\right\|_2 \leq \sqrt{U}$. Taking the union bound, with probability $\geq 1 - \beta$,

$$\left\| -\frac{1}{\sqrt{L_\gamma}}\widehat{\Sigma}^{-1/2}\widetilde{\Sigma}^{1/2}\widehat{\mu}_Z \right\|_2 \leq \sqrt{\frac{U}{L_\gamma}} \cdot \sqrt{\frac{d+2\sqrt{d\log(3/\beta)}+2\log(3/\beta)}{d+1}}.$$

Now, we argue that the first term is also bounded. First, we apply Lemma B.10 to get

$$(\mu - \widetilde{\mu})(\mu - \widetilde{\mu})^T \preceq \frac{8\gamma}{1-\gamma}(\Sigma + \widetilde{\Sigma}) \preceq \frac{8\gamma}{1-\gamma}\left(\frac{4}{(1-\gamma)^4}\widetilde{\Sigma} + \widetilde{\Sigma}\right) \preceq \frac{40\gamma}{(1-\gamma)^5}\widetilde{\Sigma}.$$

Note that $U\widehat{\Sigma}_Z \succeq \mathbb{I} \implies \widetilde{\Sigma} \preceq U\widehat{\Sigma}$. Plugging this in above, and rearranging gives

$$\widehat{\Sigma}^{-1/2}(\mu-\widetilde{\mu})(\mu-\widetilde{\mu})^T\widehat{\Sigma}^{-1/2} \preceq \frac{4}{(1-\gamma)^4} \cdot U \cdot \frac{10\gamma}{1-\gamma} \cdot \mathbb{I} = U_\gamma \cdot \frac{10\gamma}{1-\gamma} \cdot \mathbb{I}.$$

Thus, we have

$$\left\| \frac{1}{\sqrt{L_\gamma}}\widehat{\Sigma}^{-1/2}(\mu-\widetilde{\mu}) \right\|_2 \leq \frac{1}{\sqrt{L_\gamma}} \cdot \sqrt{\sigma_1(\widehat{\Sigma}^{-1/2}(\mu-\widetilde{\mu})(\mu-\widetilde{\mu})^T\widehat{\Sigma}^{-1/2})} \leq \sqrt{\frac{U_\gamma}{L_\gamma}} \cdot \sqrt{\frac{10\gamma}{1-\gamma}}.$$

Combining the two terms gives us

$$\|\mu_Y\|_2 \leq \sqrt{\frac{U_\gamma}{L_\gamma}} \cdot \left( \sqrt{\frac{10\gamma}{1-\gamma}} + \sqrt{\frac{d+2\sqrt{d\log(3/\beta)}+2\log(3/\beta)}{d+1}} \right)$$

which completes the proof. $\qquad\square$

Lemma B.11, combined with guarantees of the Gaussian estimators from Lemma A.24, allows us to conclude the following analogue to Theorem B.5 (this stated in the main body as Theorem 2.4).

**Theorem B.12** ($\gamma$-Far Public-Private Gaussian Estimation). *For all $\alpha, \beta, \varepsilon, \rho > 0$, there exist $\varepsilon$-DP and $\rho$-zCDP algorithms that take $d + 1$ public samples $\widetilde{X} = (\widetilde{X}_1, \ldots, \widetilde{X}_{d+1})$ and $n$ private samples $X = (X_1, \ldots, X_n)$ from unknown, d-dimensional Gaussians $\mathcal{N}(\widetilde{\mu}, \widetilde{\Sigma})$ and $\mathcal{N}(\mu, \Sigma)$, respectively, such that $0 \leq \mathrm{d_{TV}}(\mathcal{N}(\mu, \Sigma), \mathcal{N}(\widetilde{\mu}, \widetilde{\Sigma})) \leq \gamma < 1$, and are private with respect to the private samples, and return $\widehat{\mu}_X \in \mathbb{R}^d$ and $\widehat{\Sigma}_X \in \mathbb{R}^{d \times d}$, such that $\mathrm{d_{TV}}(\mathcal{N}(\widehat{\mu}_X, \widehat{\Sigma}_X), \mathcal{N}(\mu, \Sigma)) \leq \alpha$ with probability at least $1 - \beta$, as long as the following bounds on $n$ hold.*

1. *For $\rho$-zCDP (with computational efficiency) via [KLSU19] and $d + 1$ public samples,*

$$n = O\left( \frac{d^2 + \log\left(\frac{1}{\beta}\right)}{\alpha^2} + \frac{d^2 \cdot \mathrm{polylog}\left(\frac{d}{\alpha\beta\rho}\right) + d\log\left(\frac{d}{\alpha\beta\rho(1-\gamma)}\right)}{\alpha\sqrt{\rho}} \right.$$
$$\left. + \frac{d^{1.5} \cdot \mathrm{polylog}\left(\frac{d}{\beta\rho(1-\gamma)}\right)}{\sqrt{\rho}} \right).$$

2. *For $\varepsilon$-DP (without computational efficiency) via [BKSW19] and $d + 1$ public samples,*

$$n = O\left( \frac{d^2 + \log\left(\frac{1}{\beta}\right)}{\alpha^2} + \frac{d^2 \log\left(\frac{d}{\alpha\beta(1-\gamma)}\right)}{\alpha\varepsilon} \right).$$

*Proof.* The theorem follows from the privacy and the utility guarantees of a modified version of Algorithm 2, which uses the modified version of Algorithm 1 as outlined in Lemma B.11, and parameters $L_\gamma$ and $U_\gamma$, instead. The proof remains the same as that of Theorem B.5. $\qquad\square$

## C  Estimating Gaussian Mixtures

In this section, we state our algorithms to learn mixtures of well-separated Gaussians under differential privacy, when we have trace amounts of public data available to us. We then provide theoretical guarantees for privacy of our algorithms with respect to the private data samples, along with its utility. We analyse two cases here: when we have $\widetilde{O}(1/w_{\min})$ public samples available to us, and when we have $\widetilde{O}(d/w_{\min})$ public samples available.

The general scheme of our algorithms is similar to that of the non-private algorithm for learning Mixtures of Gaussians by [AM05], and the private algorithm (with no public data available) by [KSSU19]. Many algorithms for learning mixtures of Gaussians follow a specific outline – use PCA to project the data onto a low-dimensional subspace, which would separate the largest Gaussian from the rest; partition the dataset again, if there is more than one Gaussian present, otherwise isolate the lone Gaussian; estimate the parameters of that Gaussian; repeat the process on the remaining points. Our algorithms are also spectral algorithms that rely upon techniques, like PCA, but use their private counterparts at various stages.

The difference in the aforementioned cases, where we have $O(1/w_{\min})$ and $O(d/w_{\min})$ public samples available, is that in the former, we have very few public samples, but they are enough to be able to isolate a group of nearby clusters (called, a "supercluster"), which we could then use for an application of private PCA. In the latter case, we wouldn't need to apply private PCA at all because the number of public samples is enough to be able to do non-private PCA accurately.

**Assumption.**  We make an assumption about the shape of the covariances of all Gaussians, which essentially says that the Gaussians are not too flat or degenerate. Let $N$ be the total number of points sampled from $D \in \mathcal{G}(d, k, s)$. We formalise this as follows.

$$\forall i \in [k], \quad \|\Sigma_i\|_F \sqrt{\log(Nk/\beta)} \leq \frac{1}{8}\mathrm{tr}(\Sigma_i) \quad \text{and} \quad \|\Sigma_i\|_2 \log^2(Nk/\beta) \leq \frac{1}{8}\mathrm{tr}(\Sigma_i) \qquad (2)$$

Note that this implies that $d \geq 8\log^2(Nk/\beta)$ because $\mathrm{tr}(\Sigma_i) \leq d\|\Sigma_i\|_2$. We also assume that $\beta < 1/2$.

**Notations.** We also define a few notations before moving on to the technical sections. We say that a Gaussian $\mathcal{N}(\mu, \Sigma)$ in high dimensions satisfying Condition 2 is contained within $S \subset \mathbb{R}^d$, if $B_{\sqrt{\frac{3}{2}\text{tr}(\Sigma)}}(\mu) \subseteq S$. For a low-dimensional Gaussian $\mathcal{N}(\mu, \Sigma')$ in $\ell < d$ dimensions, we say that a set $S \subset \mathbb{R}^d$ contains the Gaussian for fixed $0 < \beta < 1$ and $N \geq 1$, if $B_{\sqrt{\|\Sigma'\|_2}\sqrt{2\ell \ln(2N\ell/\beta)}}(\mu') \subset S$.

Also, given $D \in \mathcal{G}(d, k, s)$, we say that $B \subset \mathbb{R}^d$ is "pure", if for each Gaussian component $i$, $B$ either contains the Gaussian $\mathcal{N}(\mu_i, \Sigma_i)$, or $B_{\sqrt{\frac{3}{2}\text{tr}(\Sigma_i)}}(\mu_i) \cap B = \emptyset$. Similarly, given a set of points $T$ from $D$, we say that $S \subseteq T$ is "clean", if for every $i \in [k]$, $S$ has all points from component $i$ that lie in $T$, or it has none of them. We sometimes say that for a clean $S_1 \subseteq T$, $S_2$ is a clean subset of $S_1$ if $S_2 \subseteq S_1$ and $S_2$ is clean with respect to $T$.

## C.1 Deterministic Regularity Conditions

Here, we state a few results that would be useful in solving the problems in the next two subsections. The following condition bounds the number of points from each component of a Gaussian mixture.

**Condition C.1** (Sample Frequency). Suppose we have $n$ samples from a mixture of Gaussians $D = \{(\mu_1, \Sigma_1, w_1), \ldots, (\mu_k, \Sigma_k, w_k)\} \in \mathcal{G}(d, k, s)$ satisfying Condition 2. Then for each $i \in [k]$, the number of samples from component $i$ lies in $\left[\frac{nw_i}{2}, \frac{3nw_i}{2}\right]$. Furthermore, if $w_i \geq \frac{4\alpha}{9k}$, then the number of points from component $i$ lies in $\left[n(w_i - \frac{\alpha}{9k}), n(w_i + \frac{\alpha}{9k})\right]$.

The next condition bounds the distance of any point sampled from a Gaussian mixture from the mean of its respective component.

**Condition C.2** (Intra-Gaussian Distances From Mean). Suppose we have $n$ samples from a mixture of Gaussians $D = \{(\mu_1, \Sigma_1, w_1), \ldots, (\mu_k, \Sigma_k, w_k)\} \in \mathcal{G}(d, k, s)$ satisfying Condition 2. Then for each $i \in [k]$, if $x$ is one of the samples from component $i$, then $\frac{3}{4}\text{tr}(\Sigma_i) \leq \|x - \mu_i\|^2 \leq \frac{3}{2}\text{tr}(\Sigma_i)$.

This condition bounds the distance between any two points sampled from the same component of a Gaussian mixture.

**Condition C.3** (Intra-Gaussian Distances Between Points). Suppose we have $n$ samples from $D = \{(\mu_1, \Sigma_1, w_1), \ldots, (\mu_k, \Sigma_k, w_k)\} \in \mathcal{G}(d, k, s)$ satisfying Condition 2. Then for each $i \in [k]$, if $x, y$ are two of the samples from component $i$, then $\frac{3}{2}\text{tr}(\Sigma_i) \leq \|x - y\|^2 \leq 3\text{tr}(\Sigma_i)$.

This final condition quantifies the minimum distance between any two points from different components of a Gaussian mixture.

**Condition C.4** (Inter-Gaussian Distances). Suppose we have $n$ samples from a mixture of Gaussians $D = \{(\mu_1, \Sigma_1, w_1), \ldots, (\mu_k, \Sigma_k, w_k)\} \in \mathcal{G}(d, k, s)$ satisfying Condition 2, where $s \geq \Omega(\sqrt{\ln(n/\beta)})$. Then for any $i, j \in [k]$ with $i \neq j$, if $x$ is one of the samples from component $i$, and $y$ is one of the samples from component $j$, then $\|x - y\| \geq \frac{\sqrt{\max\{\text{tr}(\Sigma_i), \text{tr}(\Sigma_j)\}}}{4}$.

We state another condition that says that in low dimensions, too, the distance between the mean of a Gaussian from all its points is bounded, and it is true for all the components of the mixture.

**Condition C.5** (Intra-Gaussian Distances from Mean in Low Dimensions). Suppose we have $n$ samples from $D = \{(\mu_1, \Sigma_1, w_1), \ldots, (\mu_k, \Sigma_k, w_k)\} \in \mathcal{G}(\ell, k)$. For each $i \in [k]$, let $\sigma_i^2 = \|\Sigma_i\|_2$. Then for any $i \in [k]$ and a fixed $0 < \beta < 1$, if $x$ is a datapoint sampled from $\mathcal{N}(\mu_i, \Sigma_i)$, then $\|\mu_i - x\| \leq \sigma_i \sqrt{2\ell \ln(2n\ell/\beta)}$.

The final condition about low-dimensional Gaussian mixtures bounds the distance between two points from the same component.

**Condition C.6** (Intra-Gaussian Distances Between Points in Low Dimensions). Suppose we have $n$ samples from $D = \{(\mu_1, \Sigma_1, w_1), \ldots, (\mu_k, \Sigma_k, w_k)\} \in \mathcal{G}(\ell, k)$. For each $i \in [k]$, let $\sigma_i^2 = \|\Sigma_i\|_2$. Then for any $i \in [k]$ and a fixed $0 < \beta < 1$, if $x, y$ are a datapoints sampled from $\mathcal{N}(\mu_i, \Sigma_i)$, then $\|x - y\| \leq 2\sigma_i \sqrt{2\ell \ln(2n\ell/\beta)}$.

Now, we prove that the above conditions hold with high probability.

**Lemma C.7.** *Suppose we have $n$ samples from $D \in \mathcal{G}(d, k, s)$ satisfying Condition 2. If*

$$n \geq \max\left\{\frac{12}{w_{\min}} \ln(2k/\beta), \frac{405k^2}{2\alpha^2} \ln(2k/\beta)\right\},$$

*then Condition C.1 holds with probability at least $1 - \beta$.*

*Proof.* We simply use Lemma A.1 $k$ times with $p \geq w_{\min}$, and Lemma A.2 $k$ times, and take the union bound. $\qquad\square$

**Lemma C.8.** *Suppose we have $n$ samples from $D \in \mathcal{G}(d, k, s)$ satisfying Condition 2. Then Condition C.2 holds with probability at least $1 - \beta$.*

*Proof.* From the Hanson-Wright inequality (Lemma A.3), $\forall i, \forall x \sim \mathcal{N}(\mu_i, \Sigma_i)$, we have that

$$\text{tr}(\Sigma_i) - 2\|\Sigma_i\|_F \sqrt{\log(n/\beta)} \leq \|x - \mu_i\|_2^2 \leq \text{tr}(\Sigma_i) + 2\|\Sigma_i\|_F \sqrt{\log(n/\beta)} + 2\|\Sigma_i\|_2 \log(n/\beta).$$

Condition 2, combined with the above, gives us the required result. $\qquad\square$

**Lemma C.9.** *Suppose we have $n$ samples from $D \in \mathcal{G}(d, k, s)$ satisfying Condition 2. Then Condition C.3 holds with probability at least $1 - \beta$.*

*Proof.* For every $i$ and for any $x, y \sim \mathcal{N}(\mu_i, \Sigma_i)$, we have that $x - y \sim \mathcal{N}(0, 2\Sigma_i)$. Using Condition 2 and Lemma A.3 again, we have the result. $\qquad\square$

The following is a quantification of the median radius of a Gaussian in terms of the trace of its covariance.

**Lemma C.10.** *Suppose we have $n$ samples from $D = \{(\mu_1, \Sigma_1, w_1), \ldots, (\mu_k, \Sigma_k, w_k)\} \in \mathcal{G}(d, k, s)$ satisfying Condition 2. Then for each $i \in [k]$, the median radius $R_i$ of component $i$ lies in $\left(\sqrt{\frac{3}{4}\text{tr}(\Sigma_i)}, \sqrt{\frac{3}{2}\text{tr}(\Sigma_i)}\right)$.*

*Proof.* For each $i \in [k]$, let $a_i = \sqrt{\frac{3}{4}\text{tr}(\Sigma_i)}$ and $b_i = \sqrt{\frac{3}{2}\text{tr}(\Sigma_i)}$. We know from the proof of Lemma C.8 that for a given $i \in [k]$, $1 - \frac{\beta}{k}$ of the probability mass of $G_i := \mathcal{N}(\mu_i, \Sigma_i)$ lies in $B_{b_i}(\mu_i) \setminus B_{a_i}(\mu_i)$. Let $\beta_1 \leq \beta/k$ be the probability mass of $G_i$ in $B_{a_i}(\mu_i)$, and let $\beta_2 = \beta/k - \beta_1$. We know that $\beta_1 < 1/2$, therefore, $R_i > a_i$. Since the mass of $G_i$ outside $B_{b_i}(\mu_i)$ is $\beta_2 < 1/2$, it must be the case that $R_i < b_i$. Hence, the claim. $\qquad\square$

**Lemma C.11.** *Suppose we have $n$ samples from $D \in \mathcal{G}(d, k, s)$ satisfying Condition 2. If $s \geq \Omega(\sqrt{\ln(n/\beta)})$, then Condition C.4 holds with probability at least $1 - 2\beta$.*

*Proof.* For this, we will use Lemmata A.12 and C.10. We know from Lemma A.12 that with probability at least $1 - \beta/m$, for any given $i \in [k]$, and $x$ sampled from $G_i := \mathcal{N}(\mu_i, \Sigma_i)$ in the dataset sampled from $D$,

$$\|x - z\|^2 \geq (\max\{R_i - \ln(\beta/2m)\sigma_i, 0\})^2 + \|z - \mu_i\|^2 - 2\sqrt{2\ln(\beta/2m)}\sigma_i\|z - \mu_i\|.$$

For any $j \in [k]$ with $j \neq i$, let $z = \mu_j$. Suppose $R_i$ and $R_j$ are the median radii of $G_i$ and $G_j$, respectively. WLOG, let's assume that $R_i \geq R_j$. Then we have the following.

$$\|x - \mu_j\|^2 \geq (\max\{R_i - \ln(2m/\beta)\sigma_i, 0\})^2 + \|\mu_j - \mu_i\|^2 - 2\sqrt{2\ln(2m/\beta)}\sigma_i\|\mu_j - \mu_i\|$$

$$\geq (\max\{R_i - \ln(2m/\beta)\sigma_i, 0\})^2 + \frac{\|\mu_j - \mu_i\|^2}{2}$$

(Separation condition and Condition 2.)

Now, let $y \sim G_j$. Due to our separation condition, we know that,

$$2\sqrt{2\ln(2m/\beta)}\sigma_j \leq \frac{\|\mu_j - \mu_i\|}{2\sqrt{2}} \leq \frac{\|x - \mu_j\|}{2}.$$

We apply Lemma A.12 again with $z = x$.

$$\|x - y\|^2 \geq (\max\{R_j - \ln(2m/\beta)\sigma_j, 0\})^2 + \|x - \mu_j\|^2 - 2\sqrt{2\ln(2m/\beta)}\sigma_j\|x - \mu_j\|$$

$$\geq (\max\{R_j - \ln(2m/\beta)\sigma_j, 0\})^2 + \frac{\|x - \mu_j\|^2}{2}$$

(Separation condition.)

$$\geq (\max\{R_j - \ln(2m/\beta)\sigma_j, 0\})^2 + \frac{(\max\{R_i - \ln(2m/\beta)\sigma_i, 0\})^2}{4} + \frac{\|\mu_j - \mu_i\|^2}{8}$$

$$\geq \frac{\text{tr}(\Sigma_j)}{4} + \frac{\text{tr}(\Sigma_i)}{16} + \frac{\|\mu_j - \mu_i\|^2}{8} \qquad \text{(Condition 2.)}$$

$$\geq \frac{\text{tr}(\Sigma_i)}{16}$$

This proves the lemma. $\qquad \square$

**Lemma C.12.** *Suppose we have $n$ samples from $D = \{(\mu_1, \Sigma_1, w_1), \ldots, (\mu_k, \Sigma_k, w_k)\} \in \mathcal{G}(\ell, k)$. Then for a fixed $0 < \beta < 1$, Condition C.5 holds with probability at least $1 - \beta$.*

*Proof.* Fix $i \in [k]$, $x$ be an arbitrary sample from component $i$, and an eigenvector of $\Sigma_i$ $v$. In that direction, using Lemma A.4, we know that with probability at least $1 - \frac{\beta}{n\ell}$, $\|(x - \mu_i)vv^T\| \leq \sigma_i\sqrt{2\ln(2n\ell/\beta)}$. Applying the union bound in all $\ell$ directions of $\Sigma_i$, we have that with probability at least $1 - \frac{\beta}{n}$, $\|x - \mu_i\| \leq \sigma_i\sqrt{2\ell\ln(2n\ell/\beta)}$. Applying the union bound again over all $n$ points, we get the required result. $\qquad \square$

**Lemma C.13.** *Suppose we have $n$ samples from $D = \{(\mu_1, \Sigma_1, w_1), \ldots, (\mu_k, \Sigma_k, w_k)\} \in \mathcal{G}(\ell, k)$. Then for a fixed $0 < \beta < 1$, Condition C.6 holds with probability at least $1 - \beta$.*

*Proof.* The proof just involves application of triangle inequality for any pairs of points from the same component, along with Lemma C.12. $\qquad \square$

## C.2 $\widetilde{O}(1/w_{\min})$ **Public Samples**

In this subsection, we assume that we have $\widetilde{O}(1/w_{\min})$ public samples available. Here, the number of public samples is not enough to be able to do PCA accurately, so we, instead, try to use them to isolate the superclusters in the data, and perform private actions, like private PCA and private estimation algorithms, on the private data based on that information.

We start with a superclustering algorithm for public data. It first computes a constant-factor approximation to the diamater ($\approx r$) of the largest Gaussian in the public dataset by finding the largest minimum pairwise distance among all points. It then tries to find a supercluster around the point ($x$) that has the largest minimum distance to all other points. The idea is that $B_r(x)$ will have a lot of points, but if $B_{r+r}(x)$ doesn't have any more points, and $B_{r+2r}(x)$ doesn't have any more points either, then it must be the case that with high probability, no other Gaussian would have a point in $B_{2r}(x)$. This follows from the concentration properties of high-dimensional Gaussians, and the fact that $r$ approximates the diameter of the largest Gaussian. If we do encounter more points in either $B_{2r}(x)$ or $B_{3r}(x)$, we expand the ball by $O(r)$ and check for these conditions again. We show that the ball returned by the algorithm is of size at most $O(kr)$, and that it contains the Gaussian corresponding to the point $x$.

We now state the main result about Algorithm 3.

**Theorem C.14.** *There exists an algorithm, which if given a clean subset of*

$$m \geq O\left(\frac{\ln(k/\beta)}{w_{\min}}\right)$$

*points from $D \in \mathcal{G}(d, k)$, where $D$ satisfies Condition 2, it outputs $c \in \mathbb{R}^d$ and $R \in \mathbb{R}$, such that the following holds, given Conditions C.1 to C.4 hold.*

1. *$B_R(c)$ is pure.*

2. *Let $\Sigma'$ be the covariance of the largest Gaussian (covariance with maximum trace) contained in $B_R(c)$. Then $R \in O(k\sqrt{\text{tr}(\Sigma')})$.*

*Proof.* Let $\mathcal{N}(\mu, \Sigma)$ be the largest Gaussian that has points in $\widetilde{X}$. First, we claim that $r \in \Theta(\sqrt{\text{tr}(\Sigma)})$. We know from Lemma C.11, that the minimum distance between $x \sim \mathcal{N}(\mu_i, \Sigma_i)$ and $y \sim \mathcal{N}(\mu_j, \Sigma_j)$

---

**Algorithm 3:** Superclustering on Public Data $\text{SC}_{k,m}(\widetilde{X})$

---

**Input:** Samples $\widetilde{X}_1, \ldots, \widetilde{X}_{m'} \in \mathbb{R}^d$. Parameters $\beta, k, m > 0$.
**Output:** Centre $c \in \mathbb{R}^d$ and radius $R \in \mathbb{R}$.

Let $r \leftarrow 16 \cdot \max_{x \in \widetilde{X}} \min_{y \in \widetilde{X}} \|x - y\|$.

Let $c \leftarrow \arg\max_{x \in \widetilde{X}} \min_{y \in \widetilde{X}} \|x - y\|$.

Set $pure \leftarrow \text{FALSE}$.
Let $R \leftarrow r$.
**For** $i \leftarrow 1, \ldots, k$

    Let $m_i \leftarrow \left| B_R(c) \cap \widetilde{X} \right|$.

    **If** $\left| B_{R+r}(c) \cap \widetilde{X} \right| = m_i$

        **If** $pure = \text{TRUE}$

            **Return** $B_R(c)$

        **elif** $\left| B_{R+2r}(c) \cap \widetilde{X} \right| = m_i$

            **Return** $(c, R + r)$

        **Else**

            $pure = \text{FALSE}$

            $R \leftarrow R + 3r$

    **Else**

        **If** $\left| B_{R+2r}(c) \cap \widetilde{X} \right| = \left| B_{R+r}(c) \cap \widetilde{X} \right|$

            $pure = \text{TRUE}$

        **Else**

            $pure = \text{FALSE}$

        $R \leftarrow R + 2r$

**Return** $(c, R)$.

---

for $i \neq j$ is lower bounded by $\frac{\sqrt{\max\{\text{tr}(\Sigma_i), \text{tr}(\Sigma_j)\}}}{4}$. We also know from Lemma C.9 that for any $i \in [k]$, and any $x, z \sim \mathcal{N}(\mu_i, \Sigma_i)$, $\|x - z\| \geq \sqrt{\frac{3}{2} \text{tr}(\Sigma_i)}$. Let $y \in \widetilde{X}$, such that $y$ is sampled from component $j \neq i$ and is closest to $x$, that is, $y$ is the closest point to $x$ that has not been sampled from component $i$ to which $x$ belongs. This means that distance between $x$ and its nearest point is at least $\min\left\{ \sqrt{\frac{3}{2}\text{tr}(\Sigma_i)}, \frac{\sqrt{\max\{\text{tr}(\Sigma_i), \text{tr}(\Sigma_j)\}}}{4} \right\} \geq \min\left\{ \sqrt{\frac{3}{2}\text{tr}(\Sigma_i)}, \frac{\sqrt{\text{tr}(\Sigma_i)}}{4} \right\}$. We also know from Lemma C.9 that for any $x, z \sim \mathcal{N}(\mu_i, \Sigma_i)$, $\|x - z\| \leq \sqrt{3\text{tr}(\Sigma_i)}$. This means that the point in $\widetilde{X}$ that is closest to $x$ is at most $\sqrt{3\text{tr}(\Sigma_i)}$ far from $x$. This shows that the distance between $x$ and the point in $\widetilde{X}$ that is closest to $x$ lies in $\left[ \frac{\sqrt{\text{tr}(\Sigma_i)}}{4}, \sqrt{3\text{tr}(\Sigma_i)} \right]$. This is true for any point in $\widetilde{X}$. Therefore, the largest minimum distance has to lie in the interval $\left[ \frac{\sqrt{\text{tr}(\Sigma)}}{4}, \sqrt{3\text{tr}(\Sigma)} \right]$. This shows that $r \in \Theta(\sqrt{\text{tr}(\Sigma)})$.

We use this to show that $B_R(c)$ is pure. We prove the following claims for that.

**Claim C.15.** *If at the end of any iteration, the algorithm doesn't exit the loop, but calls any of the return steps within the loop instead, then the returned ball is pure.*

*Proof.* Suppose it happens in iteration $i$. Consider the case where the return statement was called in the **elif** block. If $B_R(c)$ were intersecting with a component, then $B_{R+r}(c)$ would contain it (Lemmata C.8 and C.9) because $r$ is large enough. Now, we need to show that $B_{R+r}(c)$ doesn't intersect with any new component. By construction, it must mean that $\left| B_R(c) \cap \widetilde{X} \right| = m_i$, but

$\left| B_{R+r}\left(c\right) \cap \widetilde{X}\right| = m_i$ and $\left| B_{R+2r}\left(c\right) \cap \widetilde{X}\right| = m_i$. If $B_{R+r}\left(c\right)$ intersects with another component that it doesn't fully contain, then it must be the case that the component would have at least one more point in $B_{R+2r}\left(c\right) \setminus B_{R+r}\left(c\right)$ and would be contained in that region (by Lemma C.8) because $r$ is large enough. Since that doesn't happen, there cannot be any component that is partially contained in $B_{R+r}\left(c\right)$. Therefore, $B_{R+r}\left(c\right)$ is pure.

If return were called because $pure$ were true, then it must have been the case that in iteration $i-1$, before the final update to $R$, $pure$ must have been set to TRUE because $B_{R+r}\left(c\right)$ found new points, but $B_{R+2r}\left(c\right)$ couldn't. In all other case, $pure$ is set to FALSE. Since in iteration $i$, $B_{R+r}\left(c\right)$ couldn't find new points either, by the same reasoning as above, $B_R\left(c\right)$ must be pure. □

**Claim C.16.** *In iteration $i$, either the algorithm exits with a pure ball containing at least $i-1$ components, or at the end of iteration $i$, $B_R\left(c\right)$ contains at least $i$ components.*

*Proof.* We show this by induction on $i$, as defined in Algorithm 3.
**Base Case:** At the beginning of the first iteration, $R = r$. But we know from Lemmata C.9 and C.8 that $r$ is big enough that the mean of the Gaussian from which $c$ has been sampled, along with all other points from the same Gaussian would be contained in $B_r\left(c\right)$ because the mean of the Gaussian is at most $\sqrt{\frac{3}{2}\mathrm{tr}(\Sigma)}$ away from $c$ and all other points from the same component are at most $\sqrt{3\mathrm{tr}(\Sigma)}$ away from $c$. Now, if the algorithm does not exit after the end of the iteration, our base case holds because we have at least one component in the ball at end of the iteration. If the algorithm exited, it must be the case that $B_{2r}\left(c\right)$ is pure (from the above claim) and has at least one component.
**Inductive Step:** Suppose for $i \geq 1$, assume that at the end of iteration $i$, $B_R\left(c\right)$ contains at least $i$ Gaussian components, or that it exits with at least $i-1$ components. If the algorithm exits in iteration $i$, we are done. So, we assume that it doesn't. In iteration $i+1$, if the algorithm exits and returns a ball, then it is pure (by the claim above and the inductive hypothesis), and has at least $i$ components. So, we're done. Otherwise, either $B_{R+r}\left(c\right)$ or $B_{R+2r}\left(c\right)$ finds new points. If $B_{R+r}\left(c\right)$ finds new points, it is fully containing a component not previously contained within $B_R\left(c\right)$, or it is intersecting with new components. In the former case, a new component is added by the end of the iteration, and our claim holds. This holds in the latter case, too, by Lemmata C.8 and C.9 because $B_{R+2r}\left(c\right)$ would contain that new component. So, we have contained at least $i+1$ components at the end of iteration $i+1$. If $B_{R+r}\left(c\right)$ has no new points, but $B_{R+2r}\left(c\right)$ does, then these must be points from a new component. Therefore, $B_{R+3r}\left(c\right)$ would completely contain the new component by Lemmata C.8 and C.9. So, we have $i+1$ components again by the end of the iteration. □

From the above argument, we have that either the algorithm exited in the loop, and returned a pure ball, or it had $k$ components at the end of iteration $k$, in which case, the returned ball is again pure. This gives us the first part of the theorem.

Now, we prove the second part. Let $c \in \widetilde{X}$ be as defined in Algorithm 3. Let $\Sigma$ be the covariance of the largest Gaussian $G$ in $\widetilde{X}$. Suppose $c$ was sampled from component $G' := \mathcal{N}(\mu', \Sigma')$. If $\mathrm{tr}(\Sigma) = \mathrm{tr}(\Sigma')$, then by the proof above, $r \in \Theta(\sqrt{\mathrm{tr}(\Sigma)})$. Suppose $\mathrm{tr}(\Sigma') < \mathrm{tr}(\Sigma)$. Then in the worst case, the nearest point to $c$ is $\sqrt{3\mathrm{tr}(\Sigma')}$ away from $c$. It is possible in this case that all points from the largest Gaussian had nearest points $\frac{\sqrt{\mathrm{tr}(\Sigma)}}{4}$ away from them, but $\sqrt{3\mathrm{tr}(\Sigma')} \geq \frac{\sqrt{\mathrm{tr}(\Sigma)}}{4}$. In this case $r$ would still be $16\sqrt{3}$-factor approximation of $\sqrt{\mathrm{tr}(\Sigma')}$. We know that the returned ball $B_R\left(c\right)$ contains the component of $c$. Also, the loop of the algorithm runs at most $k$ times, where in each iteration, $R$ can only increase additively by $3r$. Therefore, the final radius can be at most $(3k+1)r \in O(k\sqrt{\mathrm{tr}(\Sigma')})$. This proves the second part of the theorem. □

Now, we restate a folklore private algorithm for obtaining a projection matrix of the top-$k$ subspace of a dataset via PCA, along with its main result. Here, we just talk about its utility, but focus on the privacy later when we instantiate it for different forms of differential privacy. Given a dataset, it simply truncates all points to within a given range, computes the empirical covariance matrix, then adds random noise to each entry of the matrix that is sampled from a Gaussian distribution scaled according to the appropriate privacy parameters depending on the notion of privacy. In the algorithm, $f_{PCA}(\mathrm{PrivParams})$ denotes an appropriate function of the privacy parameters, which we would set according to the type of differential privacy guarantee we require.

---
**Algorithm 4:** DP PCA $\mathrm{PrivPCA}_{\mathrm{PrivParams},\ell}(X, r)$

> **Input:** Private samples $X = (X_1, \dots, X_{n'}) \in \mathbb{R}^{n' \times d}$. Radius $r \in \mathbb{R}$. Parameters
> $\qquad$ $\mathrm{PrivParams} \subset \mathbb{R}, \ell > 0$.
> **Output:** Projection matrix $\Pi \in \mathbb{R}^{d \times d}$.
>
> Let $f_{PCA}$ be an appropriate function of the elements of $\mathrm{PrivParams}$ to ensure DP.
> Set $\sigma_P \leftarrow 2r^2 f_{PCA}(\mathrm{PrivParams})$.
> Truncate all points of $X$ to within $B_r(0)$ to get dataset $Y$.
> Let $E \in \mathbb{R}^{d \times d}$, such that for all $i, j \in [d]$ with $i \le j$, $E_{i,j} \sim \mathcal{N}(0, \sigma_P^2)$ and $E_{j,i} \leftarrow E_{i,j}$.
> Let $\Pi$ be the projection matrix of the top-$\ell$ subspace of $Z \leftarrow Y^T Y + E$.
>
> **Return** $\Pi$.
---

**Theorem C.17** (Theorem 9 from [DTTZ14])**.** *Let $X, Y, Z, \ell, c, r, \sigma_P$ be quantities as defined in Algorithm 4, $C_\ell$ be the best rank-$\ell$ approximation to $Y^T Y$, and $\widehat{C}_\ell$ be the rank-$\ell$ approximation to $Z$. Then with probability at least $1 - \beta$,*

$$\|Y^T Y - \widehat{C}_\ell\|_2 \le \|Y^T Y - C_\ell\|_2 + O\left(\sigma_P \sqrt{d} + \sigma_P \sqrt{\ln(1/\beta)}\right).$$

We get the following corollary using Theorem C.17 and Lemmata A.13 and A.14.

**Corollary C.18.** *Let $X, Y, Z, \ell, c, r, \sigma_P, \Pi$ be quantities as defined in Algorithm 4, such that $X$ is a clean subset of $n$ samples from a mixture of $k$ Gaussians $D$ satisfying Condition 2, and $X$ contains points from $k'$ components. Suppose $w_{\min}$ is the minimum mixing weight of a components with respect to $D$. Let $\sigma_{\max}^2$ be the largest directional variance of any component that has points in $X$, and $\Sigma$ be the covariance of the Gaussian with largest trace that has points in $X$. If $r \in O(k\sqrt{\mathrm{tr}(\Sigma)})$, all points in $X$ lie within $B_r(0)$, and*

$$n \ge O\left(\frac{d}{w_{\min}} + d^{1.5} k^2 f_{PCA}(\mathrm{PrivParams}) + \frac{\ln(k/\beta)}{w_{\min}}\right),$$

*then with probability at least $1 - O(\beta)$, for each component $(\mu', \Sigma', w')$ that has points in $X$,*

$$\|\mu' - \mu'\Pi\| \le O\left(\frac{\sigma_{\max}}{\sqrt{w'}}\right).$$

*Proof.* By construction of our algorithm and our assumption on $r$, we know that $O\left(\sigma_P \sqrt{d} + \sigma_P \sqrt{\ln(1/\beta)}\right) \in f_{PCA}(\mathrm{PrivParams}) \cdot O\left(d^{1.5} k^2 + dk^2\sqrt{\ln(1/\beta)}\right) \in f_{PCA}(\mathrm{PrivParams}) \cdot O\left(d^{1.5} k^2\right)$ (since $D$ satisfies Condition 2). Using Lemma A.7 and our bound on $n$, we know that for each component $(\mu', \Sigma', w')$, the empirical mean of all the points from that Gaussian in $X$ would be at most $\frac{\sqrt{\|\Sigma'\|_2}}{\sqrt{w'}}$ away from $\mu'$. By the same reasoning, the empirical mean of the same projected points would be at most $\frac{\sqrt{\|\Sigma'\|_2}}{\sqrt{w'}}$ away from $\mu'\Pi$ because the projection of a Gaussian is still a Gaussian. Lemmata A.13 and A.14, and Theorem C.17 together guarantee that the distance between the empirical mean of those points and that of the projected points of that component would be at most $O\left(\frac{\sigma_{\max}}{\sqrt{w'}}\right)$. Applying the triangle inequality gives us the result. $\qquad\square$

Now, we mention a result about the sensitivity of $Y^T Y$ in Algorithm 4, which would be used in later subsections.

**Lemma C.19.** *In Algorithm 4, the $\ell_2$ sensitivity of $Y^T Y$ is $2r^2$.*

*Proof.* Since all points in $X$ are truncated to within $B_r(0)$, for all $i$, $\|X_i\| \le r$. Therefore, for a neighbouring dataset $X'$, and corresponding truncated dataset $Y'$, $\|Y^T Y - Y'^T Y'\|_F \le 2r^2$. $\qquad\square$

Next, we describe an algorithm for partitioning the data after the PCA step has been performed. For this, we assume that the data is in low-dimensions ($\ell$-dimensional), where Condition 2 may not hold, but there is at least one cluster that is well-separated from the other points, or there is just one cluster in the whole dataset. We provide a private partitioner for private and public data for this regime, which in the first case, returns a partition of the datasets that contains their clean subsets, and in the second case, it returns the same cluster itself. For this, we define the following queries (akin to those in [KSSU19]) that would be used for the public and the private data (respectively) in the algorithm.

$$Q_{\text{Pub}}(X, c, r, t) := (|X \cap B_r(c)| \geq t) \wedge$$
$$(|X \cap (B_{11r}(c) \setminus B_r(c))| = 0) \wedge$$
$$(|X \cap (\mathbb{R}^d \setminus B_{11r}(c))| \geq t) \tag{3}$$

$$Q_{\text{Priv}}(X, c, r, t, \text{PrivParams}) := (\text{PCount}_{\text{PrivParams}}(X \cap B_r(c)) \geq t) \wedge$$
$$\left( \text{PCount}_{\text{PrivParams}}(X \cap (B_{5r}(c) \setminus B_r(c))) < \frac{t}{320} \right) \wedge$$
$$\left( \text{PCount}_{\text{PrivParams}}(X \cap (\mathbb{R}^d \setminus B_{5r}(c))) \geq t \right) \tag{4}$$

Just like in [KSSU19], we define the notion of a *terrific ball*. We say a ball $B_r(c)$ is $(\gamma, t)$-*terrific* with respect to some dataset $X$ for $\gamma > 1, t > 0$, if (1) $B_r(c)$ contains at least $t$ points from $X$, (2) $B_{\gamma r}(c) \setminus B_r(c)$ contains at most $\frac{t}{80}$ points from $X$, and (3) $\mathbb{R}^d \setminus B_{\gamma r}(c)$ contains at least $t$ points from $X$. We sometimes omit the parameter $t$ when the context is clear.

---

**Algorithm 5:** DP Partitioner DPLowDimPartitioner$_{n,m,w_{\min},\text{PrivParams}}(\widetilde{Y}, \widetilde{Z}, r_{\max}, r_{\min})$

---

**Input:** Private Samples $\widetilde{Z}_1, \ldots, \widetilde{Z}_{n'} \in \mathbb{R}^d$. Public Samples $\widetilde{Y}_1, \ldots, \widetilde{Y}_{m'} \in \mathbb{R}^d$. Parameters
$\quad n \geq n', m \geq m', w_{\min} > 0$. Privacy Parameters PrivParams.
**Output:** A tuple of centre $c \in \mathbb{R}^d$, radius $R \in \mathbb{R}$, or $\perp$.

**For** $i \leftarrow 0, \ldots, \log(r_{\max}/r_{\min})$
$\quad r_i \leftarrow \frac{r_{\max}}{2^i}$
$\quad$ **For** $j \leftarrow 1, \ldots, m'$
$\quad\quad c_j \leftarrow \widetilde{Y}_j$
$\quad\quad$ **If** $Q_{\text{Pub}}(\widetilde{Y}, c_j, r_i, \frac{m w_{\min}}{2}) = \text{TRUE}$
$\quad\quad\quad$ **If** $Q_{\text{Priv}}(\widetilde{Z}, c_j, 2r_i, \frac{n w_{\min}}{4}, \text{PrivParams}) = \text{TRUE}$
$\quad\quad\quad\quad$ **Return** $(c = c_j, R = 2r_i)$.
$\quad\quad\quad$ **Else**
$\quad\quad\quad\quad$ **Return** $\perp$.
**Return** $\perp$.

---

Now, we prove the main theorem about the utility of Algorithm 5. In the following, $f_{\text{PCount}}(\text{PrivParams})$ denotes an appropriate function of the privacy parameters based on the accuracy guarantees of PCount (Lemma A.23) depending on the notion of DP being used. We prove the privacy guarantees in later sections.

**Theorem C.20.** *Let $Y$ be a clean subset of a set of public samples from $D \in \mathcal{G}(d, k)$, $Z$ be a clean subset of private samples from $D$, $\Pi$ be a projection matrix to $\ell$ dimensions, $\widetilde{Y} = Y\Pi, \widetilde{Z} = Z\Pi$ be the input to Algorithm 5, $r_{\max}, r_{\min} > 0$, such that for some $j \in [m']$, $\widetilde{Y} \cap B_{r_{\max}}\left(\widetilde{Y}_i\right) = \widetilde{Y}$ and $\widetilde{Z} \cap B_{r_{\max}}\left(\widetilde{Y}_i\right) = \widetilde{Z}$ and $B_{r_{\max}}\left(\widetilde{Y}_i\right)$ is pure with respect to the components of $D$ projected on to the subspace of $\Pi$ having points in $\widetilde{Y}$ (and $\widetilde{Z}$). Suppose $\sigma_{\max}^2$ is the largest directional variance among the Gaussians that have points in $Y$. Let that Gaussian be $\mathcal{N}(\mu, \Sigma)$. Suppose,*

$$n \geq O\left( \frac{d \ln(k/\beta)}{w_{\min}} + \frac{\ln(1/\beta)}{w_{\min} \cdot f_{\text{PCount}}(\text{PrivParams})} \right)$$

*and*

$$m \geq O\left( \frac{\ln(k/\beta)}{w_{\min}} \right).$$

*Then we have the following with probability at least $1 - 4\beta$.*

1. *Suppose there are at least two Gaussians that have points in $Y$, and the same Gaussians have points in $Z$. For any other Gaussian $\mathcal{N}(\mu', \Sigma')$ that has points in $Y$ and $Z$, suppose for $N = m+n$, $\|\mu\Pi - \mu'\Pi\| \geq \Omega(\sigma_{\max}\sqrt{\ell \ln(N\ell/\beta)})$. Let $r_{\min} \leq \sigma_{\max}\sqrt{2\ell \ln(2N\ell/\beta)}$. Then the ball returned by Algorithm 5 is pure with respect to the components of $D$ projected by $\Pi$, which have points in $\widetilde{Y}$, such that $\mathbb{R}^d \setminus B_R(c)$ only contains components from $D$ projected by $\Pi$ that have points in $\widetilde{Y}$ (alternatively, $\widetilde{Z}$), and so does $B_R(c)$.*

2. *Suppose there is only one Gaussian that has points in $Y$ and $Z$. Then the algorithm returns $\perp$.*

*Proof.* We first prove the first part of the theorem. In this case, we know that there exists a ball of some radius $r > 0$ and some centre $c \in \widetilde{Y}$, such that $B_r(c)$ has the required properties. We prove the existence of such a ball first. Let $c \in \widetilde{Y}$ be a point, such that the corresponding Gaussian to $c$ in $d$ dimensions is $\mathcal{N}(\mu, \Sigma)$ (with $\|\Sigma\| = \sigma_{\max}^2$). For any Gaussian $\mathcal{N}(\mu', \Sigma')$ that has points in $Y$, in the subspace of $\Pi$, $\|\mu\Pi - \mu'\Pi\| \geq 103\sigma_{\max}\sqrt{\ell \ln(N\ell/\beta)}$. Now, we know that for any Gaussian $\mathcal{N}(\mu'', \Sigma'')$, in this subspace, for any $y \sim \mathcal{N}(\mu'', \Sigma'')$ that lies in $Y$ and any $z \sim \mathcal{N}(\mu'', \Sigma'')$ that lies in $Z$, $\|y\Pi - \mu''\Pi\|, \|z\Pi - \mu''\Pi\| \leq \sqrt{\|\Pi\Sigma''\Pi\|_2}\sqrt{2\ln(2N\ell/\beta)}$ from Lemma C.12. The triangle inequality implies that for any $y, z \sim \mathcal{N}(\mu, \Sigma)$ with $y \in Y$ and $z \in Z$, and $a, b \sim \mathcal{N}(\mu', \Sigma')$ with $a \in Y$ and $b \in Z$, $\|y\Pi - a\Pi\|, \|z\Pi - b\Pi\| \geq 100\sigma_{\max}^2\sqrt{\ell \ln(2N\ell/\beta)}$. Given that $B_{2\sigma_{\max}\sqrt{2\ln(2N\ell/\beta)}}(c)$ contains all points in $Y\Pi$ and $Z\Pi$, whose Gaussian in high dimensions is $\mathcal{N}(\mu, \Sigma)$, and contains the Gaussian in low dimensions, this ball satisfies the required properties.

Now, let's say the algorithm returns a ball $B = B_{2r_i}(c_j)$. We know that in this case, because the radius $r_i$ decreases when $i$ increases, $r_i \geq 2\sigma_{\max}\sqrt{2\ell \ln(2N\ell/\beta)}$. Because $r_i$ is bigger than the largest possible diameter of any Gaussian component in the $\ell$-dimensional subspace, $B_{r_i}(c_j)$ must be, at least, containing the component of $c_j$ in that subspace. We know that the shell $S = B_{11r_i}(c_j) \setminus B_{r_i}(c_j)$ has no points from any component in $\widetilde{Y}$ (by construction). For any component inside $B_{r_i}(c_j)$ (either partially or completely), the distance between the mean of any such component from the mean of any other component that is completely outside $B_{r_i}(c_j)$ has to be at least $10r_i - 2 \times \frac{r_i}{2} = 9r_i$ (because $r_i$ is at least twice the maximum distance from the mean of a Gaussian to any of its points in that subspace). If there a component, whose mean lies in the shell $T = B_{\frac{21r_i}{2}}(c_j) \setminus B_{\frac{3r_i}{2}}(c_j)$, it would be completely contained inside $S$, and we would see points of $\widetilde{Y}$ in $S$, which would be a contradiction. Therefore, the means of all components that lie completely outside $B_{r_i}(c_j)$ must be within $\mathbb{R}^d \setminus B_{\frac{21r_i}{2}}(c_j)$, which means that they would be entirely contained within $\mathbb{R}^d \setminus B_{10r_i}(c_j)$ (because of Lemma C.12). Therefore, $B$ cannot have any component that lies completely outside $B_{r_i}(c_j)$. By similar reasoning, the mean of any component that partially intersects with $B_{r_i}(c_j)$ has to be within $B_{\frac{3r_i}{2}}(c_j)$. Therefore, by Lemma C.12 again, that component would be contained within $B_{2r_i}(c_j)$. Therefore, the ball $B$ is pure. By the construction of the query $Q_{\text{Priv}}$, our sample complexity (of $n$), and the guarantees of PCount, with probability at least $1 - 3\beta$, we know that the noise in each call to PCount cannot be more than $\frac{nw_{\min}}{1280}$ (Lemma A.23). We also know that the number of points from each component is at least $\frac{nw_{\min}}{2}$ (from Lemma C.7). Since $B_{2r_i}(c_j)$ is pure, it must have at least $\frac{nw_{\min}}{2}$ points in it, so the noisy answer via PCount would be at least $\frac{nw_{\min}}{4}$. Similarly, since there is no Gaussian component even partially contained within $B_{10r_i}(c_j) \setminus B_{2r_i}(c_j)$, the noisy answer will be less than $\frac{nw_{\min}}{1280}$. Finally, since there is at least one component in $\mathbb{R}^d \setminus B_{10r_i}(c_j)$, the noisy answer of PCount would be over $\frac{nw_{\min}}{4}$. So, $Q_{\text{Priv}}$ would return TRUE. This proves the first part of the theorem.

Now, we prove the second part of the theorem. It is sufficient to show that there exists no 5-terrific ball in the private dataset $\widetilde{Z}$ when all the data is coming from a single Gaussian. First, we claim that if there is a 5-terrific ball in a dataset $X$, then there exists a 3-terrific ball, too. The argument is simple. Suppose $B_r(c)$ is a 5-terrific ball, then it is easy to see that $B_{r'}(c)$ for $r' = \frac{5r}{3}$ is a 3-terrific ball. This is because $B_{r'}(c)$ would contain more points that $B_r(c)$, hence, $B_{3r'}(c) \setminus B_{r'}(c)$ would contain fewer points than $B_{5r}(c) \setminus B_r(c)$. Since $\mathbb{R}^d \setminus B_{5r}(c) = \mathbb{R}^d \setminus B_{3r'}(c)$, the final constraint would hold, as well. Hence, the claim. So now, it is sufficient to show that there exists no 3-terrific ball with respect to $\widetilde{Z}$.

We prove this via contradiction. We know that due to our sample complexity (bound on $n$), the noise in each call to PCount is at most $\frac{nw_{\min}}{1280}$. Therefore, for a query $Q_{\mathrm{Priv}}(\widetilde{Z}, c', r', \frac{nw_{\min}}{4}, \mathrm{PrivParams})$ to return a ball $B_{r'}(c')$, $B_{r'}(c')$ and $\mathbb{R}^d \setminus B_{5r'}(c')$ must contain at least $\frac{nw_{\min}}{8}$ points from $\widetilde{Z}$, and $B_{5r'}(c') \setminus B_{r'}(c')$ must contain at most $\frac{nw_{\min}}{640}$ points from $\widetilde{Z}$. This implies the existence of a 3-terrific ball of radius $r$ with respect to $\widetilde{Z}$. This means that there exists a unit vector $v$ such that on projecting all the data on to $v$, we would have an three adjacent intervals along the vector $I_1, I_2, I_3$ of length $2r$ each along the direction of $v$, such that $I_1$ and $I_3$ together contain at most $\frac{nw_{\min}}{640}$ points, $I_2$ contains at least $\frac{nw_{\min}}{8}$ points, and the rest of the line contains at least $\frac{nw_{\min}}{8}$ points. Suppose the mixing weight of the Gaussian in question is $w_i$. Lemma C.7 shows that the number of points in $\widetilde{Z}$ is $Cnw_i$, where $\frac{1}{2} \leq C \leq \frac{3}{2}$. A straightforward application of Chernoff bound (Lemma A.1) and the union bound over all $2^{O(d)}$ unit vectors in a cover of the unit sphere in $\mathbb{R}^d$, along with our sample complexity, implies that the probability mass contained in $I_2$ and the part of line excluding $I_1$, $I_2$, and $I_3$ (which we call, $I_4$) is at least $\frac{w_{\min}}{8Cw_i}$ each, and the mass contained in $I_1 \cup I_3$ is strictly less than $\frac{w_{\min}}{160Cw_i}$. We show that this is impossible when the underlying distribution along $v$ is a (one-dimensional) Gaussian.

Suppose the variance along $v$ of the said Gaussian is $\sigma^2$. WLOG, we will assume that its mean is $0$. Note that for the said condition about the probability mass to be true, $I_2$ must contain the mean. This is because of the symmetry about the mean and unimodality of one-dimensional Gaussians – if $I_2$ does not contain the mean, then either $I_1$ or $I_3$ would be closer to the mean, and would contain more mass than $I_2$ because all three of these intervals are of the same length. Next, we show that the total mass contained within $I_1$ and $I_3$ is minimised when the mean is at the centre of $I_2$. After that, it would be sufficient to show that in this configuration, the probability mass contained within $I_1$ and $I_3$ together is at least $\frac{w_{\min}}{160Cw_i}$ when the mass contained in $I_2$ and $I_4$ each is at least $\frac{w_{\min}}{8Cw_i}$. Let $0 \leq r_1 \leq r$ and $r_2 = 2r - r_1$, such that one end point of $I_2$ is $r_1$ away from the mean, and the other end point is $r_2$ away from the mean. Then the probability mass contained within $I_1 \cup I_3$ is

$$\frac{1}{\sqrt{2\pi}\sigma} \int\limits_{r_1}^{r_1+2r} e^{-\frac{t^2}{2\sigma^2}} dt + \frac{1}{\sqrt{2\pi}\sigma} \int\limits_{r_2}^{r_2+2r} e^{-\frac{t^2}{2\sigma^2}} dt = \frac{1}{\sqrt{2\pi}} \int\limits_{r_1/\sigma}^{(r_1+2r)/\sigma} e^{-\frac{t^2}{2}} dt + \frac{1}{\sqrt{2\pi}} \int\limits_{(2r-r_1)/\sigma}^{(4r-r_1)/\sigma} e^{-\frac{t^2}{2}} dt.$$

Define $f(r_1) = \frac{1}{\sqrt{2\pi}} \int\limits_{r_1/\sigma}^{(r_1+2r)/\sigma} e^{-\frac{t^2}{2}} dt + \frac{1}{\sqrt{2\pi}} \int\limits_{(2r-r_1)/\sigma}^{(4r-r_1)/\sigma} e^{-\frac{t^2}{2}} dt$. Then we have the following.

$$\frac{df(r_1)}{dr_1} = \frac{1}{\sqrt{2\pi}} \left[ \frac{1}{\sigma} \cdot e^{-\frac{(r_1+2r)^2}{2\sigma^2}} - \frac{1}{\sigma} \cdot e^{-\frac{r_1^2}{2\sigma^2}} - \frac{1}{\sigma} \cdot e^{-\frac{(4r-r_1)^2}{2\sigma^2}} + \frac{1}{\sigma} \cdot e^{-\frac{(2r-r_1)^2}{2\sigma^2}} \right]$$

Within the interval $[0, r]$, setting the above to $0$, we have $r_1 = r$. It can be checked that for all $r_1 \in [0, r]$, $\frac{d^2 f(r_1)}{dr_1^2} > 0$. This means that within $[0, r]$, $f$ is minimised at $r_1 = r$. Therefore, for the rest of the proof, we would concentrate on lower bounding the quantity $q(r) = \frac{2}{\sqrt{2\pi}} \int\limits_{r/\sigma}^{3r/\sigma} e^{-\frac{t^2}{2}} dt$. We define $p(r) = \frac{2}{\sqrt{2\pi}} \int\limits_{0}^{r/\sigma} e^{-\frac{t^2}{2}} dt$ to be the mass of $I_2$ in this regime, and $\lambda(r) = \frac{2}{\sqrt{2\pi}} \int\limits_{3r/\sigma}^{\infty} e^{-\frac{t^2}{2}} dt$ to be the mass of $I_4$. We consider three cases.

**Case 1:** $r^2 = \sigma^2 + \tau, 0 \leq \tau \leq \sigma^2$. Noting that $e^{-x+0.25} \geq e^{-x^2}$ for $x \in [0, 1]$ and $\frac{1}{\sqrt{2}} \leq \frac{r}{\sqrt{2}\sigma} \leq 1$, we have the following.

$$\frac{w_{\min}}{8Cw_i} \leq p(r) = \frac{2}{\sqrt{\pi}} \int\limits_{0}^{r/\sqrt{2}\sigma} e^{-s^2} ds \leq \frac{2e^{\frac{1}{4}}}{\sqrt{\pi}} \int\limits_{0}^{r/\sqrt{2}\sigma} e^{-s} ds = \frac{2e^{\frac{1}{4}}}{\sqrt{\pi}} \cdot \left( 1 - e^{-\frac{r}{\sqrt{2}\sigma}} \right)$$

Now, we compute $q(r)$.

$$q(r) = \frac{2}{\sqrt{\pi}} \int\limits_{r/\sqrt{2}\sigma}^{3r/\sqrt{2}\sigma} e^{-s^2} ds$$

$$= \frac{2}{\sqrt{\pi}} \left[ \int_{r/\sqrt{2}\sigma}^{1} e^{-s^2} ds + \int_{1}^{3/\sqrt{2}} e^{-s^2} ds + \int_{3/\sqrt{2}}^{3r/\sqrt{2}\sigma} e^{-s^2} ds \right]$$

$$\geq \frac{2}{\sqrt{\pi}} \left[ \int_{r/\sqrt{2}\sigma}^{1} e^{-s} ds + \int_{1}^{3/\sqrt{2}} e^{-s^2} ds + \int_{3/\sqrt{2}}^{3r/\sqrt{2}\sigma} e^{-\frac{3rs}{\sqrt{2}\sigma}} ds \right]$$

$$= \frac{2}{\sqrt{\pi}} \left[ e^{-\frac{r}{\sqrt{2}\sigma}} - \frac{1}{e} + \int_{1}^{3/\sqrt{2}} e^{-s^2} ds + \frac{\sqrt{2}\sigma}{3r} \cdot e^{-\frac{9r}{2\sigma}} - \frac{\sqrt{2}\sigma}{3r} \cdot e^{-\frac{9r^2}{2\sigma^2}} \right]$$

$$\geq \frac{2}{\sqrt{pi}} \left[ \int_{1}^{3/\sqrt{2}} e^{-s^2} ds - \frac{1}{e} + e^{-\frac{r}{\sqrt{2}\sigma}} + \frac{1}{3} \cdot e^{-\frac{9r}{2\sigma}} - \frac{1}{3} \cdot e^{-\frac{9r^2}{2\sigma^2}} \right]$$

$$> \frac{2e^{\frac{1}{4}}}{20\sqrt{\pi}} \cdot \left( 1 - e^{-\frac{r}{\sqrt{2}\sigma}} \right)$$

$$\geq \frac{w_{\min}}{160Cw_i}$$

We have a contradiction. Thus, in this case, the probability mass in $I_1 \cup I_3$ has to be at least $\frac{w_{\min}}{160Cw_i}$.

**Case 2:** $r^2 = \sigma^2 + \tau, \tau \geq \sigma^2$. We have the following upper bound on $\lambda(r)$ using Lemma A.4.

$$\lambda(r) = \frac{2}{\sqrt{2\pi}} \int_{3r/\sigma}^{\infty} e^{-\frac{t^2}{2}} dt \leq 2e^{-\frac{9r^2}{2\sigma^2}}$$

We know that $\lambda(r) \geq \frac{w_{\min}}{8Cw_i}$. This implies that $\frac{r^2}{\sigma^2} \leq \frac{2}{9} \ln\left( \frac{16Cw_i}{w_{\min}} \right)$. This is also equivalent to $\frac{w_{\min}}{16Cw_i} \leq \frac{1}{e^9}$. Lemma A.6 gives us the following lower bound on $1 - p(r)$.

$$1 - p(r) = \frac{2}{\sqrt{2\pi}} \int_{r/\sigma}^{\infty} e^{-\frac{t^2}{2}} dt \geq 0.06 e^{-\frac{3r^2}{2\sigma^2} + \frac{3}{2}} = (0.06)e^{\frac{3}{2}} e^{-\frac{3r^2}{2\sigma^2}}$$

This gives us the following lower bound on $q(r)$.

$$q(r) = \frac{2}{\sqrt{2\pi}} \int_{r/\sigma}^{\infty} e^{-\frac{t^2}{2}} dt - \frac{2}{\sqrt{2\pi}} \int_{3r/\sigma}^{\infty} e^{-\frac{t^2}{2}} dt$$

$$\geq (0.06)e^{\frac{3}{2}} e^{-\frac{3r^2}{2}} - 2e^{-\frac{9r^2}{2\sigma^2}}$$

$$\geq (0.03)e^{\frac{3}{2}} e^{-\frac{3r^2}{2}} \qquad \text{(For } r^2 \geq 2\sigma^2.\text{)}$$

$$\geq (0.03)e^{\frac{3}{2}} e^{-\frac{3}{2} \cdot \frac{2}{9} \ln\left( \frac{16Cw_i}{w_{\min}} \right)}$$

$$= (0.03)e^{\frac{3}{2}} \cdot \left( \frac{w_{\min}}{16Cw_i} \right)^{\frac{1}{3}}$$

$$\geq \frac{w_{\min}}{160Cw_i} \qquad \text{(In the regime } \frac{w_{\min}}{16Cw_i} \leq \frac{1}{e^9}.\text{)}$$

This gives us a contradiction for the second case, as well.

**Case 3:** $r^2 = \sigma^2 - \tau, 0 \leq \tau < \sigma^2$. This final case has two sub-cases to analyse: $9r^2 \leq 2\sigma^2$ and $9r^2 > 2\sigma^2$. In the first instance, we have the following bounds on $p(r)$.

$$\frac{w_{\min}}{8Cw_i} \leq p(r) = \frac{2}{\sqrt{2\pi}} \int_{0}^{r/\sigma} e^{-\frac{t^2}{2}} dt \leq \frac{2r}{\sqrt{2\pi}\sigma}$$

Next, we lower bound $p(3r)$.

$$p(3r) = \frac{2}{\sqrt{2\pi}} \int_0^{3r/\sigma} e^{-\frac{t^2}{2}} dt \geq \frac{2}{\sqrt{2\pi}} \cdot \frac{3r}{\sigma} \cdot e^{-\frac{9r^2}{2\sigma^2}} \geq \frac{6r}{e\sqrt{2\pi}\sigma}$$

This gives us the following lower bound on $q(r)$.

$$\begin{aligned}
q(r) &= \frac{2}{\sqrt{2\pi}} \int_0^{3r/\sigma} e^{-\frac{t^2}{2}} dt - \frac{2}{\sqrt{2\pi}} \int_0^{r/\sigma} e^{-\frac{t^2}{2}} dt \\
&\geq \frac{6r}{e\sqrt{2\pi}\sigma} - \frac{2r}{\sqrt{2\pi}\sigma} \\
&> \frac{r}{5\sqrt{2\pi}\sigma} \\
&> \frac{w_{\min}}{160Cw_i}
\end{aligned}$$

This shows that this situation is impossible. So, we move on to the final instance, that is, when $\frac{2\sigma^2}{9} < r^2 < \sigma^2$. Just as we did in Case 1, we bound $p(r)$. We get the following.

$$\frac{w_{\min}}{8Cw_i} \leq p(r) \leq \frac{2e^{\frac{1}{4}}}{\sqrt{\pi}} \cdot \left(1 - e^{-\frac{r}{\sqrt{2}\sigma}}\right)$$

We lower bound $p(3r)$ again.

$$\begin{aligned}
p(3r) &= \frac{2}{\sqrt{\pi}} \int_0^1 e^{-s^2} ds + \frac{2}{\sqrt{\pi}} \int_1^{3r/\sqrt{2}\sigma} e^{-s^2} ds \\
&\geq \frac{2}{\sqrt{\pi}} \int_0^1 e^{-s^2} ds + \frac{2}{\sqrt{\pi}} \int_0^{3r/\sqrt{2}\sigma} e^{-\frac{3rs}{\sqrt{2}\sigma}} ds \\
&= \frac{2}{\sqrt{\pi}} \int_0^1 e^{-s^2} ds + \frac{2}{\sqrt{\pi}} \cdot \frac{\sqrt{2}\sigma}{3r} \left[e^{-\frac{3r}{\sqrt{2}\sigma}} - e^{-\frac{9r^2}{2\sigma^2}}\right] \\
&\geq \frac{2}{\sqrt{\pi}} \int_0^1 e^{-s^2} ds + \frac{2\sqrt{2}}{3\sqrt{\pi}} \left[e^{-\frac{3r}{\sqrt{2}\sigma}} - e^{-\frac{9r^2}{2\sigma^2}}\right]
\end{aligned}$$

Now, we lower bound $q(r)$.

$$\begin{aligned}
q(r) &= \frac{2}{\sqrt{2\pi}} \int_0^{3r/\sigma} e^{-\frac{t^2}{2}} dt - \frac{2}{\sqrt{2\pi}} \int_0^{r/\sigma} e^{-\frac{t^2}{2}} dt \\
&\geq \frac{2}{\sqrt{\pi}} \int_0^1 e^{-s^2} ds + \frac{2\sqrt{2}}{3\sqrt{\pi}} \left[e^{-\frac{3r}{\sqrt{2}\sigma}} - e^{-\frac{9r^2}{2\sigma^2}}\right] - \frac{2e^{\frac{1}{4}}}{\sqrt{\pi}} \cdot \left(1 - e^{-\frac{r}{\sqrt{2}\sigma}}\right) \\
&\geq \frac{1}{20} \cdot \frac{2e^{\frac{1}{4}}}{\sqrt{\pi}} \cdot \left(1 - e^{-\frac{r}{\sqrt{2}\sigma}}\right) \qquad\qquad \text{(For } \frac{2\sigma^2}{9} \leq r^2 \leq \sigma^2 \text{.)} \\
&\geq \frac{w_{\min}}{160Cw_i}
\end{aligned}$$

This gives us the final contradiction, hence, completing the proof for Case 3.

Therefore, there cannot be such intervals on the line of $v$. This means that there cannot be any such direction, where such intervals exist, which means that there cannot be a 3-terrific ball with respect to $\widetilde{Z}$, implying that the algorithm could not have found a 5-terrific ball with respect to $\widetilde{Z}$. By the construction of the algorithm, the output would be $\perp$. This completes the proof of the theorem. $\qquad \square$

We now state the main private algorithm for clustering points from a Gaussian mixture (Algorithm 6) that utilises the public data available to it, along with a theorem describing its utility guarantees. In the theorem, $f'_{PCA}$ and $f'_{\text{PCount}}$ denote appropriate functions of the privacy parameters in terms of the previously mentioned functions $f_{PCA}$ and $f_{\text{PCount}}$, respectively. Note that the output of the algorithm itself is not private, but the intermediate steps involved are. Since its output will not be released to public, and will just be utilised by our main private estimation algorithm, there will be no violation of privacy in the end. This algorithm is a general private framework, which could be instantiated for approximate DP and zCDP by choosing the right parameters and privacy primitives, for example, it would use approximate DP PCA algorithm for the former, and zCDP PCA algorithm for the latter. It first uses Algorithm 3 on the public data to find a supercluster that tightly bounds a group of clusters. Since the private data is sampled from the same distribution as the public data, it uses that supercluster to isolate the points in the private dataset corresponding to the components contained within the supercluster. Next, it uses private PCA to project the data on to the top-$k$ subspace, and then works within that subspace to either determine if the data is coming from a single Gaussian, or to partition the dataset into clean subsets in case it is coming from at least two Gaussians. In the former case, it adds the set of private points in the original $d$-dimensional space to the output (which contains sets of points). In the latter case, it just proceeds to work on the partitions independently.

**Theorem C.21.** *There exists an algorithm (Algorithm 6) that takes $n$ private samples $X$, and $m$ public samples from $D \in \mathcal{G}(d, k, s)$ satisfying Condition 2, and outputs a partition $C$ of $X$, such that given Conditions C.1 to C.6 hold, and*

$$s \geq \Omega(\sqrt{k \ln((n+m)k/\beta)}),$$

$$n \geq O\left( \frac{d \ln(k/\beta)}{w_{\min}} + d^{1.5} k^{2.5} f'_{PCA}(\text{PrivParams}) + \frac{\sqrt{k} \ln(k/\beta)}{w_{\min} \cdot f'_{\text{PCount}}(\text{PrivParams})} \right),$$

*and*

$$m \geq O\left( \frac{\ln(k/\beta)}{w_{\min}} \right),$$

*then with probability at least $1 - O(\beta)$, $|C| = k$ and each $S \in C$ is clean and non-empty.*

*Proof.* It is enough to show that $\forall i \geq 0$, at the end of the $i$-th iteration, (1) a new Gaussian component is isolated in $C$ and $count$ equals the number of isolated components in $C$, (2) or the private dataset $X$ is further partitioned into non-empty, clean subsets, and the public dataset $\widetilde{X}$ is further partitioned into clean subsets, such that $|Q_{\text{Pub}}| = |Q_{\text{Priv}}|$ and for each $j \in [|Q_{\text{Pub}}|]$, $Q_{\text{Pub}}[j]$ and $Q_{\text{Priv}}[j]$ have points from the same Gaussian components. We can prove this via induction on $i$.

**Base Case:** The end of the 0-th iteration essentially means the actual start of the loop. In this case, we know that $Q_{\text{Pub}}$ and $Q_{\text{Priv}}$ are clean subsets of $\widetilde{X}$ and $X$, respectively. Therefore, the claim trivially holds in this case.

**Inductive Step:** We assume for all iterations, up to and including some $i \geq 0$, the claim holds. Then we show that it holds for iteration $i + 1$, as well. By the inductive hypothesis, we know that $Z$ and $Y$ are clean subsets, and contain points from the same components. When the superclustering algorithm (Algorithm 3) is called, it finds a pure ball that will either contain the whole dataset $Y$ (hence, $Z$ because the ball is pure), or it partitions $Y$ into clean subsets (Theorem C.14). Since the components from which the points in $Y$ and $Z$ come from are the same, $B_{R_i}(c_i)$ would contain points from the same components from $Z$ as in $Y$ (because the deterministic regularity conditions hold). So, adding $Z \setminus Z^i$ to $Q_{\text{Priv}}$ and $Y \setminus Y^i$ to $Q_{\text{Pub}}$ ensures that the partition of $Z$ added to the back of $Q_{\text{Priv}}$ is clean and contains the points from the same components as those that have points in $Y \setminus Y^i$ (which itself gets added to the back of $Q_{\text{Pub}}$), which preserves the ordering of the set of components in the two queues.

If there is just one component that has points in $Y^i$ (hence, in $Z^i$), then after projecting the data points on to the subspace of $\Pi_i$, and feeding the projected datasets to Algorithm 5, we will get $\perp$ (by the guarantees from Theorem C.20) with probability at least $1 - \frac{\beta}{2k}$. Because $Z^i$ is a clean subset, $C$ now contains one more clean subset of $X$, and we have isolated a new component, and updated the value of $count$ to reflect the change.

---

**Algorithm 6:** DP GMM Hard Clustering DPHardClustering$_{\text{PrivParams},\beta,k,w_{\min}}(\widetilde{X}, X)$

---

**Input:** Private samples $X = (X_1, \ldots, X_n) \in \mathbb{R}^{n \times d}$. Public samples
$\quad\quad \widetilde{X} = (\widetilde{X}_1, \ldots, \widetilde{X}_m) \in \mathbb{R}^{m \times d}$. Parameters $\text{PrivParams} \subset \mathbb{R}, \beta, k, w_{\min} > 0$.
**Output:** Set $C \subset \mathcal{P}(X)$.

Let $Y \leftarrow \widetilde{X}$ and $Z \leftarrow X$.
Let $C \leftarrow \emptyset$.
Let $Q_{\text{Priv}}$ and $Q_{\text{Pub}}$ be queues of sets of points.
Add set $Z$ to $Q_{\text{Priv}}$, and set $Y$ to $Q_{\text{Pub}}$.
$\text{PrivParams}'$ be the set of privacy parameters modified as per composition based on
$\quad$ PrivParams.
Set $count \leftarrow 0$ and $i \leftarrow 1$.

**While** $count < k$ *and* $i \leq 2k$
$\quad$ Pop $Q_{\text{Priv}}$ to get $Z$, and $Q_{\text{Pub}}$ to get $Y$.

$\quad$ `// Run superclustering algorithm on the public dataset.`
$\quad$ Set $(c_i, R_i) \leftarrow \text{SC}_{k,m}(Y)$.

$\quad$ `// Partition both datasets on the basis of the supercluster.`
$\quad$ $Y^i \leftarrow Y \cap B_{R_i}(c_i)$ and $Z^i \leftarrow Z \cap B_{R_i}(c_i)$.
$\quad$ Add $Z \setminus Z^i$ to $Q_{\text{Priv}}$, and $Y \setminus Y^i$ to $Q_{\text{Pub}}$.

$\quad$ `// Private PCA: project points of both datasets onto returned`
$\quad\quad$ `subspace.`
$\quad$ Let $M^i \in \mathbb{R}^{|Z^i| \times d}$, such that each row $M_j^i \leftarrow c_i$.
$\quad$ $\Pi_i \leftarrow \text{PrivPCA}_{\text{PrivParams}',k}(Z^i - M^i, R_i)$.
$\quad$ $Y' \leftarrow (Y^i - M^i)\Pi_i$ and $Z' \leftarrow (Z^i - M^i)\Pi_i$.

$\quad$ `// If there is only one Gaussian, add it to` $C$`, otherwise further`
$\quad\quad$ `partition the datasets.`
$\quad$ $B \leftarrow \text{DPLowDimPartitioner}_{n,m,w_{\min},\text{PrivParams}'}(Z', Y', R_i, \frac{R_i}{\sqrt{d}})$.
$\quad$ **If** $B = \perp$
$\quad\quad$ $C \leftarrow C \cup \{Z^i\}$.
$\quad\quad$ $count \leftarrow count + 1$.
$\quad$ **Else**
$\quad\quad$ $B$ is an ordered pair $(c_i', r_i')$.
$\quad\quad$ $S' \leftarrow B_{r_i'}(c') \cap Y'$ and $T' \leftarrow B_{r_i'}(c') \cap Z'$.
$\quad\quad$ Let $S$ be points in $Y^i$ corresponding to $S'$, and $T$ be points in $Z^i$ corresponding to $T'$.
$\quad\quad$ Add $S$ to $Q_{\text{Pub}}$ and $T$ to $Q_{\text{Priv}}$.
$\quad\quad$ Add $Y^i \setminus S$ to $Q_{\text{Pub}}$ and $Z^i \setminus T$ to $Q_{\text{Priv}}$.

$\quad$ $i \leftarrow i + 1$
**Return** $C$.

---

Suppose there are at least two components that have points in $Y^i$ (hence, in $Z^i$). Then we know from Theorem C.14, that if $\Sigma$ is the covariance having the largest trace among all components that have points in $Y^i$ (hence, in $Z^i$), then $R_i \in O(k\sqrt{\text{tr}(\Sigma)})$. Let the largest directional variance among the Gaussians that have points in $Z^i$ (hence, in $Y^i$) be $\sigma_{\max}^2$, its mean be $\mu$, and its mixing weight be $w$. From the guarantees in Corollary C.18, we know that $\|\mu - \mu\Pi_i\| \leq O\left(\frac{\sigma_{\max}}{\sqrt{w}}\right)$ with probability at least $1 - \frac{\beta}{2k}$. Because of the separation condition, applying the triangle inequality, we know that the distance between $\mu\Pi_i$ and the projected mean of any other Gaussian having points in $Z^i\Pi$ is at least $\Omega(\sigma_{\max}\sqrt{k \ln((n+m)k)/\beta})$. Then from the guarantees of Theorem C.20, we have that a ball is returned, and it is pure with respect to the components of the mixture projected by $\Pi_i$ that have points in $Z^i\Pi$. Therefore, subsets $S'$ and $T'$ are clean, and so are $S$ and $T$. This implies that $Y^i \setminus S$ and $Z^i \setminus T$ are clean, too. By adding them in order to $Q_{\text{Pub}}$ and $Q_{\text{Priv}}$ respectively, we partition

the datasets $\widetilde{X}$ and $X$ further into clean subsets, and preserve the ordering of the components in the respective queues. This proves the claim.

Note that before the PCA step, we recentre the points in both $Y^i$ and $Z^i$ for the ease of analysis later when we provide guarantees for privacy. It does not affect correctness of any step of the algorithm because it is just recentering of data.

Now, we just have to show that the number of iterations we allow is enough with high probability. Because we have $k$ components in the mixture, we can only partition into clean subsets at most $k-1$ times. For each component, the loop will run at most one time. So, $2k-1$ iterations are enough to capture all the components, and $|C| = k$. Taking the union bound over all possible $2k-1$ iterations, we have the required result. $\qquad\square$

We finally state the private algorithm (Algorithm 7) for estimating the parameters of mixtures of Gaussians, along with a theorem highlighting its utility guarantees. Just like the DP clustering algorithm above, it is a private framework that could instantiated for approximate DP and zCDP settings by using the corresponding private algorithms. It calls Algorithm 6 to cluster the points according to their respective components, then calls private estimators to estimate the parameters of the respective components. It finally uses PCount to estimate their mixing weights based on the private data.

---

**Algorithm 7:** DP GMM Hard Case Estimator $\text{DPHardEstimator}_{\text{PrivParams},\alpha,\beta,k,w_{\min}}(\widetilde{X}, X)$

> **Input:** Private samples $X = (X_1, \ldots, X_n) \in \mathbb{R}^{n \times d}$. Public samples
> $\qquad \widetilde{X} = (\widetilde{X}_1, \ldots, \widetilde{X}_m) \in \mathbb{R}^{m \times d}$. Parameters $\text{PrivParams} \subset \mathbb{R}, \beta, k, w_{\min} > 0$.
> **Output:** Set $\widehat{D} = \{(\widehat{\mu}_1, \widehat{\Sigma}_1, \widehat{w}_1), \ldots, (\widehat{\mu}_k, \widehat{\Sigma}_k, \widehat{w}_k)\}$.
>
> `// Clustering.`
> Set $C \leftarrow \text{DPHardClustering}_{\text{PrivParams},\beta,k,w_{\min}}(X, \widetilde{X})$.
>
> `// Parameter estimation.`
> $\widehat{D} \leftarrow \emptyset$.
> **For** $i \leftarrow 1, \ldots, k$
> $\qquad$ Set $(\widehat{\mu}_i, \widehat{\Sigma}_i) \leftarrow \text{DPGaussianEstimator}_{\text{PrivParams},\frac{\beta}{k}}(C_i)$.
> $\qquad$ Set $\widehat{w}_i \leftarrow \max\left\{ \frac{1}{n} \cdot \text{PCount}_{\text{PrivParams}}(|C_i|), \frac{\alpha}{2k} \right\}$.
> $\qquad$ $\widehat{D} \leftarrow \widehat{D} \cup \{(\widehat{\mu}_i, \widehat{\Sigma}_i, \widehat{w}_i)\}$.
>
> **Return** $\widehat{D}$.

---

**Theorem C.22.** *For all $\alpha, \beta > 0$ and sets of privacy parameters $\text{PrivParams}$, there exists an algorithm (Algorithm 7) that takes $n$ private samples and $m$ public samples from $D \in \mathcal{G}(d, k, s)$, such that if $D = \{(\mu_1, \Sigma_1, w_1), \ldots, (\mu_k, \Sigma_k, w_k)\}$ satisfies Condition 2, and*

$$s \in \Omega(\sqrt{k \ln((n+m)k/\beta)})$$

$$n \in O\left( \frac{n_{GE}\ln(k/\beta)}{w_{\min}} + \frac{d\ln(k/\beta)}{w_{\min}} + d^{1.5}k^{2.5}f'_{PCA}(\text{PrivParams}) + \frac{\sqrt{k}\ln(k/\beta)}{w_{\min} \cdot f'_{\text{PCount}}(\text{PrivParams})} \right)$$

$$m \in O\left( \frac{\ln(k/\beta)}{w_{\min}} \right),$$

*where $n_{GE}$ is the sample complexity of privately learning a Gaussian using Lemma A.24 according to the type of DP required, then it $(\alpha, \beta)$-learns $D$.*

*Proof.* By our sample complexity bound and Theorem C.21, with probability at least $1 - \frac{\beta}{3}$, for each $i \in [k]$, $C_i$ is clean with respect to $X$. Therefore, each $C_i$ can be used to learn an independent component.

Next, by the guarantees of Lemma A.24, for each $i \in [k]$, $\text{d}_{\text{TV}}(\mathcal{N}(\widehat{\mu}_i, \widehat{\Sigma}_i), \mathcal{N}(\mu_i, \Sigma_i)) \leq \alpha$ with probability at least $1 - \frac{\beta}{3k}$.

Finally, using Lemmata C.7 and A.23, along with our sample complexity, we know that the error due to each call to PCount is at most $\frac{\alpha}{2k}$ with probability at least $1 - \frac{\beta}{3k}$, and that the error due to sampling is at most $\frac{\alpha}{2k}$. We apply the triangle inequality to get the final error of $\frac{\alpha}{k}$ for each mixing weight.

Applying the union bound over all failure events (including the failures of Conditions C.1 to C.6), we have the desired result. $\qquad\square$

### C.2.1 $(\varepsilon, \delta)$-DP Algorithm

Here, we describe our results under approximate DP constraints, and instantiate Algorithms 4, 6, and 7 for this version of DP. We will not restate the entire algorithms, but just describe how PrivParams and the sample complexity in these different algorithms change. We now state the main theorem of our section.

**Theorem C.23.** *For all $\alpha, \beta, \varepsilon, \delta > 0$, there exists an $(O(\varepsilon), O(\delta))$-DP algorithm $\mathcal{M}$ that takes $n$ private samples and $m$ public samples from $D \in \mathcal{G}(d, k, s)$, and is private with respect to the private samples, such that if $D = \{(\mu_1, \Sigma_1, w_1), \ldots, (\mu_k, \Sigma_k, w_k)\}$ satisfies Condition 2, and*

$$s \in \Omega(\sqrt{k \ln((n+m)k/\beta)})$$

$$n \in O\left( \frac{d^2 \ln\left(\frac{k}{\beta}\right)}{w_{\min}\alpha^2} + \frac{(d^2 \log(k/\delta) + d \log^{1.5}(k/\delta)) \cdot \mathrm{polylog}\left(d, \frac{k}{\beta}, \frac{1}{\alpha}, \frac{\sqrt{k}}{\varepsilon}, \ln\left(\frac{k}{\delta}\right)\right)}{w_{\min}\alpha\varepsilon} \right.$$

$$\left. + \frac{d^{1.5} k^{2.5} \log\left(\frac{k}{\delta}\right)}{\varepsilon} \right)$$

$$m \in O\left( \frac{\log(k/\beta)}{w_{\min}} \right),$$

*then $\mathcal{M}$ $(\alpha, \beta)$-learns $D$.*

*Proof.* We mainly focus on the privacy guarantees here because the accuracy would follow from Theorem C.22 after setting the parameters appropriately. Note that the privacy parameters in this case in the set PrivParams are $\varepsilon, \delta$. As per the advanced composition guarantees of DP (Lemma A.19), we set the privacy parameters in PrivParams$'$ in Algorithm 6 to be $\varepsilon' = \frac{\varepsilon}{\sqrt{12k \log(1/\delta)}}$ and $\delta' = \frac{\delta}{4k}$. Define $\varepsilon_0 = \sqrt{k}\varepsilon'$.

Now, we show that the intermediate steps in the call to Algorithm 6 yield $(\varepsilon, \delta)$-DP. We don't release the output of that step itself, but use it in subsequent steps. The clusters are formed on the basis of intersection of the privately formed balls with the private dataset, and are then used subsequently in other private algorithms. So, the final algorithm would be private, as well. Thus, it is enough to prove the privacy guarantees of those intermediate steps that yield the said balls.

In Algorithm 6, we start working on the private dataset $X$ inside the loop, specifically, when we use private PCA. In each iteration $i$, we ensure that each row of the input to Algorithm 4 ($Z^i - M^i$) has norm at most $R_i$ by virtue of selecting points that lie in a ball of radius $R_i$, and by recentering the points appropriately. Then by Lemma C.19, the sensitivity of $Y^T Y$ in Algorithm 4 is at most $2R_i^2$. Therefore, adding Gaussian noise calibrated to $\varepsilon', \delta'$ to $Y^T Y$, that is, setting $f_{PCA}(\varepsilon', \delta') = \frac{\sqrt{2\ln(2/\delta')}}{\varepsilon'}$, implying that $\sigma_P = \frac{2R_i^2 \sqrt{2\ln(2/\delta')}}{\varepsilon'}$ is enough to ensure $(\varepsilon', \delta')$-DP in this step (Lemmata A.22 and A.18). The next step in Algorithm 6 that works with the private data is at the call to Algorithm 5. In this algorithm, only one call to PCount is performed, and this is the only time it works with the private dataset. Therefore, by the guarantees of Lemma A.23, we know that this step is $\varepsilon'$-DP. The next steps are either partitioning the private data or adding the isolated private data to $C$. Since neither of those sets are being released to public, there is no privacy loss here. Therefore, each iteration is $(2\varepsilon', \delta')$-DP (by Lemma A.19). Applying composition over all $2k$ iterations, we have that all operations together in the entire run of the loop in the algorithm are $(2\varepsilon, \delta)$-DP.

In Theorem C.21, setting $f'_{PCA}(\mathrm{PrivParams}) = f_{PCA}(\varepsilon_0, \delta_0) \in O\left( \frac{\log(k/\delta)}{\varepsilon} \right)$ give us the right sample complexity for accuracy for all calls to Algorithm 4 (by Lemma A.19), since we already

multiply the $\sqrt{k}$ factor in the numerator. Similarly, setting $f'_{\text{PCount}}(\text{PrivParams}) = f_{\text{PCount}}(\varepsilon_0) = \frac{\varepsilon}{\sqrt{12\log(1/\delta)}}$ in Theorem C.21 gives us the right sample complexity for all calls to Algorithm 5.

Now, each call to the approximate-DP Gaussian learner (Lemma A.24) is on a disjoint part of the private dataset $X$. Therefore, all the $k$ calls together are $(\varepsilon, \delta)$-DP because changing one point in $X$ can change one point in only one of the clusters. By the same reasoning, all calls to PCount together are $(\varepsilon, \delta)$-DP.

As far as the accuracy goes, the first two terms in the sample complexity ensures that enough points go to each call to the Gaussian estimator from Lemma A.24 and to each call to PCount because of Lemma C.7. Therefore, our GMM estimator $(\alpha, \beta)$-learns the mixture. $\square$

### C.2.2 $\rho$-zCDP Algorithm

Here, we state our results under zCDP, and instantiate Algorithms 4, 6, and 7 for this version of DP.

**Theorem C.24.** *For all $\alpha, \beta, \rho > 0$, there exists an $O(\rho)$-zCDP algorithm $\mathcal{M}$ that takes $n$ private samples and $m$ public samples from $D \in \mathcal{G}(d, k, s)$, and is private with respect to the private samples, such that if $D = \{(\mu_1, \Sigma_1, w_1), \ldots, (\mu_k, \Sigma_k, w_k)\}$ satisfies Condition 2, for each $i \in [k]$, $\|\mu_i\| \leq R$ and $\mathbb{I} \preceq \Sigma_i \preceq K\mathbb{I}$, and*

$$s \in \Omega(\sqrt{k \ln((n+m)k/\beta)})$$

$$n \in O\left( \frac{d^2 \ln\left(\frac{k}{\beta}\right)}{w_{\min}\alpha^2} + \frac{d^2 \cdot \text{polylog}\left(\frac{dk}{\alpha\beta\rho}\right) + d \log\left(\frac{dk \log(R)}{\alpha\beta\rho}\right)}{w_{\min}\alpha\sqrt{\rho}} \right.$$

$$\left. + \frac{d^{1.5}\sqrt{k\log(K)} \cdot \text{polylog}\left(\frac{dk\log(K)}{\rho\beta}\right) + \sqrt{dk\log\left(\frac{Rdk}{\beta}\right)} + d^{1.5}k^{2.5}}{\sqrt{\rho}} \right)$$

$$m \in O\left( \frac{\log(k/\beta)}{w_{\min}} \right),$$

*then $\mathcal{M}$ $(\alpha, \beta)$-learns $D$.*

*Proof.* Again, we focus on the privacy guarantees because the accuracy would follow from Theorem C.22 after setting the parameters correctly. Note that the privacy parameter in this case in the set PrivParams is $\rho$. As per the advanced composition guarantees of DP (Lemma A.19), we set the privacy parameter in $\text{PrivParams}'$ in Algorithm 6 to be $\rho' = \frac{\rho}{2k}$.

Now, we show that the intermediate steps in the call to Algorithm 6 yield $O(\rho)$-zCDP. As before, it is enough to prove the privacy guarantees of those intermediate steps, since the output of the algorithm itself wouldn't be released, but would be used by Algorithm 7, instead.

In Algorithm 6, we start working on the private dataset $X$ inside the loop, specifically, when we use private PCA. In each iteration $i$, by the same argument as in the proof of Theorem C.23, the sensitivity of $Y^T Y$ in the call to Algorithm 4 is $2R_i^2$. Therefore, adding Gaussian noise calibrated to $\rho'$ to $Y^T Y$, that is, setting $f_{PCA}(\rho') = \frac{1}{\sqrt{2\rho'}}$, implying that $\sigma_P = \frac{2R_i^2}{\sqrt{2\rho'}}$ is enough to ensure $\rho'$-zCDP in this step (Lemmata A.22 and A.18). The next step in Algorithm 6 that works with the private data is at the call to Algorithm 5. In this algorithm, only one call to PCount is performed, and this is the only time it works with the private dataset. Therefore, by the guarantees of Lemma A.23, we know that this step is $\rho'$-zCDP. Therefore, each iteration is $2\rho'$-zCDP (by Lemma A.19). Applying composition over all $2k$ iterations, we have that all operations together in the entire run of the loop in the algorithm are $2\rho$-zCDP.

In Theorem C.21, setting $f'_{PCA}(\text{PrivParams}) = f_{PCA}(\rho) \in O\left(\frac{1}{\sqrt{\rho}}\right)$ give us the right sample complexity for accuracy for all calls to Algorithm 4 (by Lemma A.19), since we already multiply the $\sqrt{k}$ factor in the numerator. Similarly, setting $f'_{\text{PCount}}(\text{PrivParams}) = f_{\text{PCount}}(\sqrt{2\rho}) = \sqrt{2\rho}$ in Theorem C.21 gives us the right sample complexity for all calls to Algorithm 5.

Now, each call to the zCDP Gaussian learner (Lemma A.24) is on a disjoint part of the private dataset $X$. Therefore, all the $k$ calls together are $\rho$-zCDP because changing one point in $X$ can change one point in only one of the clusters. By the same reasoning, all calls to PCount together are $\rho$-zCDP.

For the accuracy guarantees, the first two terms in the sample complexity ensures that enough points go to each call to the Gaussian estimator from Lemma A.24 and to each call to PCount because of Lemma C.7. Therefore, our GMM estimator $(\alpha, \beta)$-learns the mixture. $\qquad\square$

## C.3  $\widetilde{O}(d/w_{\min})$ Public Samples

In this subsection, we assume that we have $\widetilde{O}(d/w_{\min})$ public data samples available. In this case, the number of public samples is enough to do PCA accurately. So, the approach would be much simpler than before, since we don't need to find a supercluster anymore. Also, because we have enough public samples, we could simply use the public dataset itself to partition the data when in low dimensions. Therefore, the entire clustering operations to partition the private dataset could be done using the public dataset itself, without having to touch the private dataset at all. As in the previous subsection, we will first give a general private algorithm for estimating the GMM, but will instantiate it separately for approximate DP and zCDP.

We start with the low-dimensional partitioner first, which just uses the public dataset, and the public query, as defined in 3.

---

**Algorithm 8:** Public Low-Dimensional Partitioner $\text{LowDimPartitioner}_{m,w_{\min}}(\widetilde{Y}, r_{\max}, r_{\min})$

**Input:** Public Samples $\widetilde{Y}_1, \ldots, \widetilde{Y}_{m'} \in \mathbb{R}^d$. Parameters $m \geq m', w_{\min} > 0$.
**Output:** A tuple of centre $c \in \mathbb{R}^d$, radius $R \in \mathbb{R}$, or $\perp$.

**For** $i \leftarrow 0, \ldots, \log(r_{\max}/r_{\min})$
$\quad r_i \leftarrow \frac{r_{\max}}{2^i}$
$\quad$**For** $j \leftarrow 1, \ldots, m'$
$\quad\quad c_j \leftarrow \widetilde{Y}_j$
$\quad\quad$**If** $Q_{\text{Pub}}(\widetilde{Y}, c_j, r_i, \frac{mw_{\min}}{2}) = \text{TRUE}$
$\quad\quad\quad$**Return** $(c = c_j, R = 2r_i)$.

**Return** $\perp$.

---

**Theorem C.25.** *Let $Y$ be a clean subset of a set of public samples from $D \in \mathcal{G}(d,k)$, $\Pi$ be a projection matrix to $\ell$ dimensions, $\widetilde{Y} = Y\Pi$ be the input to Algorithm 8, $r_{\max}, r_{\min} > 0$, such that for some $j \in [m']$, $\widetilde{Y} \cap B_{r_{\max}}\left(\widetilde{Y}_i\right) = \widetilde{Y}$ and $B_{r_{\max}}\left(\widetilde{Y}_i\right)$ is pure with respect to the components of $D$ projected on to the subspace of $\Pi$ having points in $\widetilde{Y}$. Suppose $\sigma_{\max}^2$ is the largest directional variance among the Gaussians that have points in $Y$. Let that Gaussian be $\mathcal{N}(\mu, \Sigma)$. Suppose,*
$$m \geq O\left(\frac{d \ln(k/\beta)}{w_{\min}}\right).$$
*Then we have the following with probability at least $1 - \beta$.*

1. *Suppose there are at least two Gaussians that have points in $Y$. For any other Gaussian $\mathcal{N}(\mu', \Sigma')$ that has points in $Y$, suppose for $N > m$, $\|\mu\Pi - \mu'\Pi\| \geq \Omega(\sigma_{\max}\sqrt{\ell \ln(N\ell/\beta)})$. Let $r_{\min} \leq \sigma_{\max}\sqrt{2\ell \ln(2N\ell/\beta)}$. Then the ball returned by Algorithm 8 is pure with respect to the components of $D$ projected by $\Pi$, which have points in $\widetilde{Y}$, such that $\mathbb{R}^d \setminus B_R(c)$ only contains components from $D$ projected by $\Pi$ that have points in $\widetilde{Y}$, and so does $B_R(c)$.*

2. *Suppose there is only one Gaussian that has points in $Y$. Then the algorithm returns $\perp$.*

*Proof.* The proof for the first part is the same as that for the first part of Theorem C.20. So, we don't discuss that any further.

For the next part, note that we could relax the query $Q_{\text{Pub}}$ by modifying the second constraint to say $|X \cap (B_{11r}(c) \setminus B_r(c))| < \frac{t}{320}$. If the answer to the original query is true, then the answer to the

modified query would be true, as well. In other words, if the answer to the modified query is false, then the answer to the original query is false, too. Therefore, it is enough to show that with probability at least $1 - \beta$, there cannot be an 11-terrific ball in $\widetilde{Y}$. As argued in the proof of Theorem C.20, it is enough to show that there exists no 3-terrific ball in $\widetilde{Y}$. We already proved this in the proof of Theorem C.20. Hence, we have the claim. $\qquad\square$

Next, we provide a clustering algorithm to partition the private dataset that just operates on the public dataset (Algorithm 9). The algorithm is the same as Algorithm 6, but uses the public data solely to partition the private data. It uses tools like PCA and Algorithm 8 on public data.

---

**Algorithm 9:** GMM Easy Clustering $\text{EasyClustering}_{\beta,k,w_{\min}}(\widetilde{X}, X)$

**Input:** Private samples $X = (X_1, \ldots, X_n) \in \mathbb{R}^{n \times d}$. Public samples
$\qquad \widetilde{X} = (\widetilde{X}_1, \ldots, \widetilde{X}_m) \in \mathbb{R}^{m \times d}$. Parameters $\beta, k, w_{\min} > 0$.
**Output:** Set $C \subset \mathcal{P}(X)$.

Let $Y \leftarrow \widetilde{X}$ and $Z \leftarrow X$.
Let $C \leftarrow \emptyset$.
Let $Q_{\text{Priv}}$ and $Q_{\text{Pub}}$ be queues of sets of points.
Add set $Z$ to $Q_{\text{Priv}}$, and set $Y$ to $Q_{\text{Pub}}$.
Set $count \leftarrow 0$ and $i \leftarrow 1$.

**While** $count < k$ *and* $i \leq 2k$
$\qquad$ Pop $Q_{\text{Priv}}$ to get $Z$, and $Q_{\text{Pub}}$ to get $Y$.

$\qquad$ // PCA: project points of both datasets onto returned subspace.
$\qquad$ Let $\Pi_i$ be the top-$k$ subspace of $Y^T Y$.
$\qquad$ $Y' \leftarrow Y\Pi_i$ and $Z' \leftarrow Z\Pi_i$.

$\qquad$ // If there is only one Gaussian, add it to $C$, otherwise further
$\qquad\quad$ partition the datasets.
$\qquad$ Let $R_i \leftarrow 4 \max\limits_{y_1, y_2 \in Y} \{\|y_1 - y_2\|\}$ and $r_i \leftarrow \frac{\sqrt{2k \ln(2(n+m)k/\beta)}}{4\sqrt{d}} \min\limits_{y_1, y_2 \in Y} \{\|y_1 - y_2\|\}$.
$\qquad$ $B \leftarrow \text{LowDimPartitioner}_{m, w_{\min}}(Y', R_i, r_i)$.
$\qquad$ **If** $B = \perp$
$\qquad\qquad$ $C \leftarrow C \cup \{Z^i\}$.
$\qquad\qquad$ $count \leftarrow count + 1$.
$\qquad$ **Else**
$\qquad\qquad$ $B$ is an ordered pair $(c_i', r_i')$.
$\qquad\qquad$ $S' \leftarrow B_{r_i'}(c') \cap Y'$ and $T' \leftarrow B_{r_i'}(c') \cap Z'$.
$\qquad\qquad$ Let $S$ be points in $Y^i$ corresponding to $S'$, and $T$ be points in $Z^i$ corresponding to $T'$.
$\qquad\qquad$ Add $S$ to $Q_{\text{Pub}}$ and $T$ to $Q_{\text{Priv}}$.
$\qquad\qquad$ Add $Y^i \setminus S$ to $Q_{\text{Pub}}$ and $Z^i \setminus T$ to $Q_{\text{Priv}}$.
$\qquad$ $i \leftarrow i + 1$
**Return** $C$.

---

**Theorem C.26.** *There exists an algorithm (Algorithm 9) that takes $n$ private samples $X$, and $m$ public samples from $D \in \mathcal{G}(d, k, s)$ satisfying Condition 2, and outputs a partition $C$ of $X$, such that given Conditions C.1 to C.6 hold, and*

$$s \geq \Omega(\sqrt{k \ln((n+m)k/\beta)}),$$

*and*

$$m \geq O\left(\frac{d \ln(k/\beta)}{w_{\min}}\right),$$

*then with probability at least $1 - O(\beta)$, $|C| = k$ and each $S \in C$ is clean and non-empty.*

*Proof.* As in the proof of Theorem C.21, it is enough to show that $\forall i \geq 0$, at the end of the $i$-th iteration, (1) a new Gaussian component is isolated in $C$ and $count$ equals the number of isolated

components in $C$, (2) or the private dataset $X$ is further partitioned into non-empty, clean subsets, and the public dataset $\widetilde{X}$ is further partitioned into clean subsets, such that $|Q_{\text{Pub}}| = |Q_{\text{Priv}}|$ and for each $j \in [|Q_{\text{Pub}}|]$, $Q_{\text{Pub}}[j]$ and $Q_{\text{Priv}}[j]$ have points from the same Gaussian components. We can prove this via induction on $i$.

**Base Case:** The proof is exactly the same as in the proof of Theorem C.21.

**Inductive Step:** We assume for all iterations, up to and including some $i \geq 0$, the claim holds. Then we show that it holds for iteration $i + 1$, as well. By the inductive hypothesis, we know that $Z$ and $Y$ are clean subsets, and contain points from the same components.

We first reason about the correctness of the PCA step. Using Lemma A.7 and our bound on $m$, we know that for each component $(\mu', \Sigma', w')$, the empirical mean of all the points from that Gaussian in $\widetilde{X}$ would be at most $\frac{\sqrt{\|\Sigma'\|_2}}{\sqrt{w'}}$ away from $\mu'$. By the same reasoning, the empirical mean of the same projected points would be at most $\frac{\sqrt{\|\Sigma'\|_2}}{\sqrt{w'}}$ away from $\mu'\Pi$ because the projection of a Gaussian is still a Gaussian. Lemmata A.13 (by setting $B = 0$) and A.14 together guarantee that the distance between the empirical mean of those points and that of the projected points of that component would be at most $O\left(\frac{\sigma_{\max}}{\sqrt{w'}}\right)$. The triangle inequality implies that $\|\mu' - \mu'\Pi\| \leq O\left(\frac{\sigma_{\max}}{\sqrt{w'}}\right)$. This happens with probability at least $1 - \frac{\beta}{2k}$.

Now, if there is just one component that has points in $Y$ (hence, in $Z$), then on feeding $Y'$ to Algorithm 5, we will get $\perp$ (by the guarantees from Theorem C.20) with probability at least $1 - \frac{\beta}{2k}$. Because $Z$ is a clean subset, $C$ now contains one more clean subset of $X$, and we have isolated a new component, and updated the value of $count$ to reflect the change.

Suppose there are at least two components that have points in $Y$ (hence, in $Z$). Let the largest directional variance among the Gaussians that have points in $Y$ be $\sigma_{\max}^2$, its mean be $\mu$, and its mixing weight be $w$. From the guarantees of the PCA step, we know that $\|\mu - \mu\Pi_i\| \leq O\left(\frac{\sigma_{\max}}{\sqrt{w}}\right)$. Because of the separation condition, applying the triangle inequality, we know that the distance between $\mu\Pi_i$ and the projected mean of any other Gaussian having points in $Z\Pi$ is at least $\Omega(\sigma_{\max}\sqrt{k\ln((n+m)k)/\beta})$. Also, we know by the construction of the algorithm that for any $y \in Y$, $Y \subseteq B_{R_i}(y)$, and from Lemmata C.8, C.9 and C.11 that $B_{R_i}(y)$ would contain all the components of $D$ that have points in $Y$. Also, by Lemmata C.9, and C.11, $r_i \leq \sigma_{\max}\sqrt{2k\ln(2(n+m)k/\beta)}$. Then from the guarantees of Theorem C.25, we have that a ball is returned, and it is pure with respect to the components of the mixture projected by $\Pi_i$ that have points in $Y'$. Therefore, subsets $S'$ and $T'$ are clean, and so are $S$ and $T$. This implies that $Y \setminus S$ and $Z \setminus T$ are clean, too. By adding them in order to $Q_{\text{Pub}}$ and $Q_{\text{Priv}}$ respectively, we partition the datasets $\widetilde{X}$ and $X$ further into clean subsets, and preserve the ordering of the components in the respective queues. This proves the claim.

As argued before in the proof Theorem C.21, allowing up to $2k$ iterations is enough. Therefore, we get a clean partition of $X$. $\qquad\square$

Finally, we provide the DP algorithm for learning GMM's in this regime of public data. The algorithm is the same as Algorithm 7, except that it calls Algorithm 9, instead of Algorithm 6.

**Theorem C.27.** *For all $\alpha, \beta > 0$ and sets of privacy parameters* PrivParams*, there exists an algorithm (Algorithm 10) that takes $n$ private samples and $m$ public samples from $D \in \mathcal{G}(d, k, s)$, such that if $D = \{(\mu_1, \Sigma_1, w_1), \ldots, (\mu_k, \Sigma_k, w_k)\}$ satisfies Condition 2, and*

$$s \in \Omega(\sqrt{k\ln((n+m)k/\beta)})$$
$$n \in O\left(\frac{n_{GE}\ln(k/\beta)}{w_{\min}}\right)$$
$$m \in O\left(\frac{d\ln(k/\beta)}{w_{\min}}\right),$$

*where $n_{GE}$ is the sample complexity of privately learning a Gaussian using Lemma A.24 and Theorem B.5 according to the type of DP required, then it $(\alpha, \beta)$-learns $D$.*

---

**Algorithm 10:** DP GMM Easy Case Estimator $\text{DPEasyEstimator}_{\text{PrivParams},\alpha,\beta,k,w_{\min}}(\widetilde{X}, X)$

---

**Input:** Private samples $X = (X_1, \ldots, X_n) \in \mathbb{R}^{n \times d}$. Public samples
$\quad\quad \widetilde{X} = (\widetilde{X}_1, \ldots, \widetilde{X}_m) \in \mathbb{R}^{m \times d}$. Parameters $\text{PrivParams} \subset \mathbb{R}, \beta, k, w_{\min} > 0$.
**Output:** Set $\widehat{D} = \{(\widehat{\mu}_1, \widehat{\Sigma}_1, \widehat{w}_1), \ldots, (\widehat{\mu}_k, \widehat{\Sigma}_k, \widehat{w}_k)\}$.

// Clustering.
Set $C \leftarrow \text{EasyClustering}_{\beta,k,w_{\min}}(X, \widetilde{X})$.

// Parameter estimation.
$\widehat{D} \leftarrow \emptyset$.
**For** $i \leftarrow 1, \ldots, k$
$\quad$ Set $(\widehat{\mu}_i, \widehat{\Sigma}_i) \leftarrow \text{DPGaussianEstimator}_{\text{PrivParams}, \frac{\beta}{k}}(C_i)$.
$\quad$ Set $\widehat{w}_i \leftarrow \max\left\{\frac{1}{n} \cdot \text{PCount}_{\text{PrivParams}}(|C_i|), \frac{\alpha}{2k}\right\}$.
$\quad$ $\widehat{D} \leftarrow \widehat{D} \cup \{(\widehat{\mu}_i, \widehat{\Sigma}_i, \widehat{w}_i)\}$.

**Return** $\widehat{D}$.

---

*Proof.* By Theorem C.26, we know that with probability at least $1 - \frac{\beta}{3}$, $|C| = k$, and for each $i \in [k]$, $C_i$ is a clean subset of $X$.

Next, by the guarantees of Lemma A.24 and Theorem B.5, for each $i \in [k]$, with probability at least $1 - \frac{\beta}{3k}$, $\text{d}_{\text{TV}}(\mathcal{N}(\widehat{\mu}_i, \widehat{\Sigma}_i), \mathcal{N}(\mu_i, \Sigma_i)) \leq \alpha$.

Finally, by the same argument as in the proof of Theorem C.22, the mixing weights are estimated accurately, as well.

Therefore, taking the union bound over all failure events, we have the required result. $\qquad\square$

### C.3.1 $(\varepsilon, \delta)$-DP Algorithm

Here, we instantiate Algorithm 10 for the case of approximate DP.

**Theorem C.28.** *For all $\alpha, \beta, \varepsilon, \delta > 0$, there exists an $(O(\varepsilon), O(\delta))$-DP algorithm $\mathcal{M}$ that takes $n$ private samples and $m$ public samples from $D \in \mathcal{G}(d, k, s)$, and is private with respect to the private samples, such that if $D = \{(\mu_1, \Sigma_1, w_1), \ldots, (\mu_k, \Sigma_k, w_k)\}$ satisfies Condition 2, and*

$$s \in \Omega(\sqrt{k \ln((n+m)k/\beta)})$$

$$n \in O\left(\frac{d^2 \ln\left(\frac{k}{\beta}\right)}{w_{\min}\alpha^2} + \frac{(d^2 \log(k/\delta) + d \log^{1.5}(k/\delta)) \cdot \text{polylog}\left(d, \frac{k}{\beta}, \frac{1}{\alpha}, \frac{\sqrt{k}}{\varepsilon}, \ln\left(\frac{k}{\delta}\right)\right)}{w_{\min}\alpha\varepsilon}\right)$$

$$m \in O\left(\frac{d \log(k/\beta)}{w_{\min}}\right),$$

*then $\mathcal{M}$ $(\alpha, \beta)$-learns $D$.*

*Proof.* We again focus on the privacy guarantees here because the accuracy would follow from Theorem C.27 after setting the parameters appropriately. Note that the privacy parameters in this case in the set $\text{PrivParams}$ are $\varepsilon, \delta$. We just have to show the privacy guarantees for the calls to the Gaussian estimator and PCount because all other steps involve computations on the public dataset.

By the same argument as in the proof for Theorem C.23, all $k$ calls together to the approximate DP learner from Lemma A.24 are $(\varepsilon, \delta)$-DP, and all calls together to PCount are $(\varepsilon, \delta)$-DP. Therefore, we get $(2\varepsilon, 2\delta)$-DP guarantee in the end.

For the accuracy goal, the sample complexity ensures that enough points go to each call to the Gaussian estimator from Lemma A.24 and to each call to PCount because of Lemma C.7. Therefore, our GMM estimator $(\alpha, \beta)$-learns the mixture. $\qquad\square$

Finally, we instantiate Algorithm 10 for the case of zCDP. Note that we don't need any bounds on the range parameters in this case because we are using the Gaussian learner from Section B.

**Theorem C.29.** *For all $\alpha, \beta, \rho > 0$, there exists an $O(\rho)$-zCDP algorithm $\mathcal{M}$ that takes $n$ private samples and $m$ public samples from $D \in \mathcal{G}(d, k, s)$, and is private with respect to the private samples, such that if $D = \{(\mu_1, \Sigma_1, w_1), \ldots, (\mu_k, \Sigma_k, w_k)\}$ satisfies Condition 2, and*

$$s \in \Omega(\sqrt{k \ln((n+m)k/\beta)})$$

$$n \in O\left(\frac{d^2 \ln\left(\frac{k}{\beta}\right)}{w_{\min}\alpha^2} + \frac{d^2 \cdot \text{polylog}\left(\frac{dk}{\alpha\beta\rho}\right)}{w_{\min}\alpha\sqrt{\rho}}\right)$$

$$m \in O\left(\frac{d \log(k/\beta)}{w_{\min}}\right),$$

*then $\mathcal{M}$ $(\alpha, \beta)$-learns $D$.*

*Proof.* We mainly prove the privacy guarantees because the accuracy would follow from Theorem C.27 after setting the parameters correctly. Note that the privacy parameter in this case in the set PrivParams is $\rho$.

Using the same argument as in the proof of Theorem C.23, we have $\rho$-zCDP for all $k$ calls to the learner in Section B (by Theorem B.5), and $\rho$-zCDP for all calls together to PCount. Therefore, we have $2\rho$-zCDP in the end.

By the same argument as before, the sample complexity is enough to send enough points to each call to the learner in Section B and to PCount. Therefore, our GMM estimator $(\alpha, \beta)$-learns the mixture. $\square$

# D A Proof-of-Concept Numerical Result

For Gaussian mean estimation, our approach is relatively simple to implement on top of an existing private algorithm. We offer some proof-of-concept simulations that demonstrate the effectiveness of public data in private statistical estimation.[5] In Figure 1, we show plots that evaluate 1-public-sample private mean estimation (the algorithm described in Section 2.1 and in more detail in Appendix B.1.1).

We examine the effect of 1 public sample on the performance of CoinPress [BDKU20] with its best parameter setting ($t = 2$), in a case where the initial a priori bounds on the mean are weak. Concretely, we draw $n$ samples from a $d = 50$ dimensional Gaussian $\mathcal{N}(\mu_k, I_d)$, where $\mu_k = k \cdot [1, ..., 1]^T$ and correspondingly set our a priori bound $R = k\sqrt{d}$ for CoinPress. We show results for $k \in [10, 100, 1000]$, representing varying levels of strength in our a prior bounds on the mean.

We follow the evaluation protocol from [BDKU20]: we target zCDP with $\rho = 0.5$, and at each sample size $n$ we run the estimator 100 times and report the 10% trimmed mean of error from the ground truth (we additionally report the 10% trimmed standard deviation as error bars). We also follow their practice of treating target failure probabilities $\beta_i$ of various steps of the algorithm as hyperparameters that can be tuned for the best empirical results. The new step we introduce (using the public sample to set $R$ based on $d, \beta$) uses $\beta = 0.01$, which is the same value used for all the $\beta_i$'s in CoinPress.

The numerical result demonstrates the promise of utilizing public data for private data analysis, and confirms the takeaway that very little public data can help greatly when a priori knowledge of the private data is weak. As is visible from the plots, the error of our public-private algorithm tracks the non-private algorithm closer when a priori bounds on the mean are weak ($k = 1000$ case).

Note that these results are only meant to be a proof-of-concept simulation to demonstrate the promise of public data – thorough tuning and evaluation of these algorithms (which is necessary to bring these algorithms to practice) is an important direction for future work.

---

[5]Our code is available at `https://github.com/alexbie98/1pub-priv-mean-est`. Experiments run within two minutes on a 2020 M1 MacBook Air.

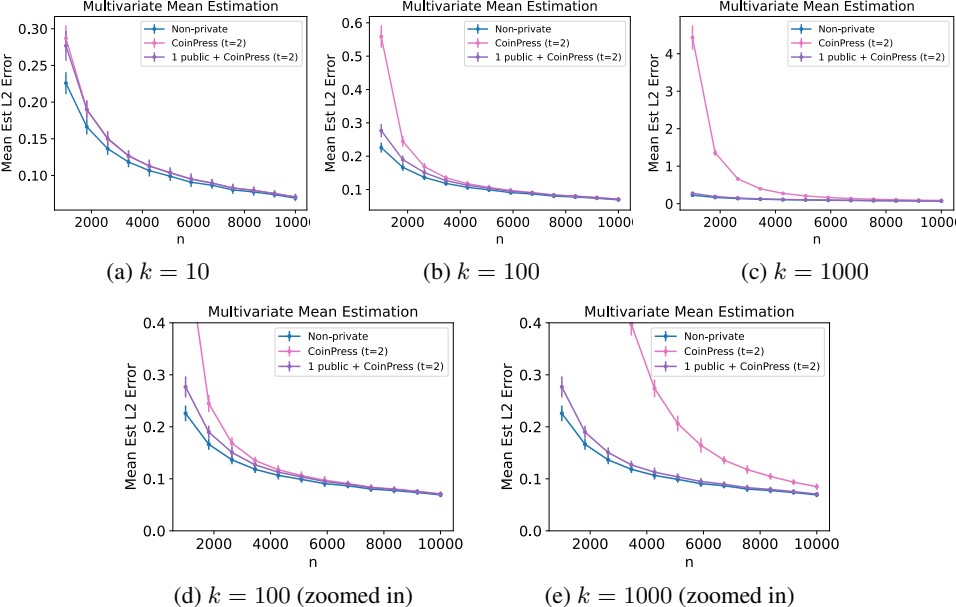

Figure 1: Comparing the error of CoinPress [BDKU20] under its best setting, against CoinPress with 1 public sample for mean estimation of $\mathcal{N}(k \cdot [1, ..., 1]^T, I_d)$ for $d = 50$, targeting zCDP at $\rho = 0.5$. Larger $k$ corresponds to weaker a priori bounds on the mean for CoinPress. For large $k$, a single public sample significantly improves results.