# OpenReview forum: "Private Estimation with Public Data"
_NeurIPS.cc/2022/Conference — NeurIPS 2022 Accept_

### Official Review · Reviewer_cFnJ · 2022-07-07

**Rating:** 7
**Confidence:** 3
**Soundness:** 4 excellent
**Presentation:** 4 excellent
**Contribution:** 3 good

**Summary:**

The paper studies estimation a setting in which there are two sets of samples available: A "public" dataset, and a "private" dataset whose elements must be protected with differential privacy. This is studied for two classes of distributions: 1) d-dimensional Gaussians, and 2) mixtures of d-dimensional Gaussians. In case 2) the public dataset must come from the same distribution as the private one, but for 1) it suffices that it is a Gaussian that is "not too far" from the same distribution. In both settings it is shown that a small amount of public data (much less than what is needed for estimation) can be used to gear the utility of the private sample, lowering the sampling complexity.
In terms of techniques the algorithm for 1) uses the public data to do a parameter range estimation and then apply a private estimator that needs these parameters, a fairly natural (and generally applicable) idea. For 2) the public data is used to improve various steps in an existing algorithm [KSSU19], which is more involved.

**Questions:**

- Given the prior work, what do you mean more precisely by "we initiate the study" in the abstract?
- In Theorem 1.2, does $\gamma$ need to be known to the algorithm, or can the sample size somehow be adapted to how well the public distribution approximates the private one?

**Limitations:**

There is an adequate discussion of limitations. Potential negative societal impacts are unlikely, so not discussed.

**Strengths And Weaknesses:**

Strengths:
- Addresses a timely and significant question (how to leverage public data) in two classical settings
- Adds to the theory of an approach that has shown great promise empirically, but where there is still only a limited theoretical understanding
- The fact that public samples from an "almost entirely dissimilar" distribution can help is intriguing and worth investigating in other settings (could lead to follow-up work)
- As far as I can tell, the technical contribution is solid (I am not familiar with all the past work on which this builds)
- The writing is clear and of high quality

Weaknesses:
- In setting 1) it was already known how to get results of a similar flavor without public data in the setting of *approximate* DP, so the advance lies only in obtaining pure DP, which is mainly of theoretical interest
- In setting 2) the public data must come from the same distribution as the private data, which is arguably a strong assumption. (I would guess it can be relaxed along the lines of case 1), but this is not worked out in the paper.)

---

> ### Author Response · Authors · 2022-08-02
> **Response to Reviewer cFnj**
>
> We thank the reviewer for their thoughtful feedback and questions. We are glad to hear that the reviewer finds the problem we study – understanding how to leverage public data in private data analysis – timely and significant, and our technical contribution solid. We agree with the reviewer that adding theory to this empirically promising approach is an important direction, and hope that our work helps refine our understanding of the role of public data in private data analysis.
>
> In the following, we address the specific concerns and questions raised by the reviewer, in order of appearance:
>
> **On the practical implication of our Gaussian estimation results:**
>
> > *In setting 1) it was already known how to get results of a similar flavor without public data in the setting of approximate DP, so the advance lies only in obtaining pure DP, which is mainly of theoretical interest*
>
> For estimating Gaussians: our advance is indeed theoretical, but we argue that there are reasons why our results might be interesting to a practitioner as well.
> Several recent works have focused on this problem of removing range bounds, developing significant technical machinery to do so via approximate DP. However, just because relaxing to approximate DP offers a solution, does not mean approximate DP is the right privacy guarantee we should adopt for every given scenario. Ideally, the decision on what privacy guarantee to go with in practice should depend on the situation itself, rather than the available tools. For example, we note that **the largest scale practical deployment to date of differential privacy, the 2020 US Census, decided to use zCDP** [1]. They make the following assessment of the two definitions:
>
> *zCDP provides privacy protection that weakens as $\delta$ approaches $0$, but never results in catastrophic failure, which some approximate differential privacy mechanisms do permit.*
>
> Along these lines, our work shows that in the situation where public data is available, there are alternative solutions that enjoy the benefits of stronger privacy definitions. Some examples of these benefits: both zCDP and pure DP are stronger guarantees that protect against the chance of ``catastrophic failure'' as specified by the delta parameter. Furthermore, even in the case where the analyst is comfortable with approx DP guarantees, zCDP offers composition benefits over approximate DP. Since gaussian estimation is a fundamental task, it is easy to see how it may be a part of a larger analysis. In that case, the much tighter composition guarantees of zCDP vs approximate DP (see the plot under “Composition under variants of differential privacy” in [2]) provide another practical use for our results.
>
> **On the same-distribution assumptions for our Gaussian mixture estimation results:**
>
> >*In setting 2) the public data must come from the same distribution as the private data, which is arguably a strong assumption. (I would guess it can be relaxed along the lines of case 1), but this is not worked out in the paper.)*
>
> We agree that the assumption is strong. We do believe it is possible to relax the assumption to a certain degree, and that similar algorithms and analyses would work with some modifications. We think relaxing these assumptions in the analysis would be valuable, and leave it to future work, seeing as our current manuscript introduces novel algorithmic ideas and is already long and technical.
>
> **Replies to specific questions:**
>
> >*Given the prior work, what do you mean more precisely by "we initiate the study" in the abstract?*
>
> Although public data has been used in private data analysis, we are unaware of work where this is done specifically for private statistical estimation tasks.
>
> >*In Theorem 1.2, does $\gamma$ need to be known to the algorithm, or can the sample size somehow be adapted to how well the public distribution approximates the private one?*
>
> We do require an upper bound on $\gamma$ to be known by the algorithm. However, we can be inaccurate by large polynomial factors of $1/(1- \gamma)$ in the coarse estimation step, and not pay significantly (only polylogarithmically in the above expression) in our private sample complexity. We agree that removing the need for the bound to be pre-specified would be an interesting direction.
>
> **Final comments:**
> If the reviewer has additional concerns and questions that they feel have not been sufficiently addressed in our reply, we would be happy to address them in the discussion phase.
>
> **References:**
> [1] John M. Abowd, Robert Ashmead, Ryan Cumings-Menon, Simson Garfinkel, Micah Heineck, Christine Heiss, Robert Johns, et al. “The 2020 Census Disclosure Avoidance System TopDown Algorithm”. Harvard Data Science Review.
> [2] Joseph P. Near and Chiké Abuah. Programming Differential Privacy. https://programming-dp.com/notebooks/ch8.html.

---

### Official Review · Reviewer_2n6x · 2022-07-09

**Rating:** 6
**Confidence:** 4
**Soundness:** 4 excellent
**Presentation:** 3 good
**Contribution:** 3 good

**Summary:**

This work studies differentially private estimation of Gaussian distributions and mixtures of Gaussians with additional public non-private data. For Gaussian estimation, the authors demonstrate that with a small amount of public data, the number of private samples required no longer depends on the bound on $\ell_2$ norm of the mean and condition number of the covariance matrix, and instead depends logarithmically on the discrepancy between public and private data distributions. For Gaussian mixtures, similar results are shown when the public and private distributions are the same.

**Questions:**

Questions regarding the results have already been stated in the Weakness section.

Suggestion on writing:
- The algorithm description for Gaussian mixture only consists of high-level ideas. Perhaps the technical contribution could stand out more if the authors could highlight a few important intermediate results or claims?
- Add a brief description about the problem formulation of estimating high-dimensional Gaussian and Gaussian mixtures.

**Limitations:**

Limitations and negative impacts are adequately addressed.

**Strengths And Weaknesses:**

Strengths:
- Understanding the performance of DP algorithms with public data is an important problem both in theory and in practice. The authors proved meaningful results in the context of estimating Gaussian distributions, which is a fundamental statistical estimation problem.
- For Gaussian mixtures, the improvement on $k$ is significant.
- The results are comprehensive. The paper studies both standard Gaussian estimation and mixture of Gaussian. Algorithms are designed for various scenarios.
- Overall, the writing is clear and well structured. The authors did a good job by motivating the algorithm ideas with a simple Gaussian mean estimation problem.

Weakness:
- For Gaussian estimation: considering that the dependence on dimensionality remains the same, removing the logarithmic dependence on range parameters might seem a weak improvement. The improvement can also be achieved by using approximate DP, which is often used in practice, so the practical implication of the result seems limited.
- I wonder if at least some of the results can be implied by the range-independent results for approximate DP. For $(\epsilon, \delta)$-DP, in the worst case roughly $\delta$ fraction of the dataset could be released as-is, which might be viewed equivalently as "public dataset". It would be great if the authors could discuss the connections.

---
Update: raised my score to 6 because the authors addressed the concern about the relation to $(\epsilon, \delta)$-DP.

---

> ### Author Response · Authors · 2022-08-02
> **Response to Reviewer 2n6x**
>
> We thank the reviewer for their thoughtful feedback and questions. We are glad the reviewer finds the problem we study – understanding the role of public data in private estimation – important to both theory and practice, and furthermore finds our results meaningful and comprehensive. We are also glad the reviewer finds our motivating Gaussian mean estimation example illuminating. In the following, we address specific concerns and questions raised by the reviewer, in order of appearance.
>
> **On the importance and practical implication of our Gaussian estimation results:**
> > *For Gaussian estimation: considering that the dependence on dimensionality remains the same, removing the logarithmic dependence on range parameters might seem a weak improvement. The improvement can also be achieved by using approximate DP, which is often used in practice, so the practical implication of the result seems limited.*
>
> For Gaussian estimation, we argue that even logarithmic dependence on the unknown distribution parameters poses a significant problem. Having algorithm parameters and guarantees that depend on a priori knowledge of the solution is always not ideal, and we argue that in the case of privacy, it is especially undesirable. Since setting these parameters a priori is difficult (perhaps easier for the mean, but certainly difficult for spectral bounds on the covariance), an analyst is unfortunately incentivized to look at the private data in order to set these parameters when faced with low utility. This practice undermines and can even completely invalidate privacy guarantees (e.g. see this case study [1] on hyperparameter tuning).
> It is precisely this desire to eliminate solution-dependent algorithm parameters which has spurred a string of recent works ([2,3,4]), each developing significant technical machinery to address the problem via relaxing to approximate DP.
> However, just because relaxing to approximate DP offers a solution, it does not mean that approximate DP is the right privacy guarantee we should adopt for every given scenario. Ideally, the decision on what privacy guarantee to go with, in practice, should depend on the situation itself, rather than the available tools. For example, we note that **the largest scale practical deployment to date of differential privacy, the 2020 US Census, decided to use zCDP** [5]. They make the following assessment of the two definitions:
>
> *zCDP provides privacy protection that weakens as $\delta$ approaches $0$, but never results in catastrophic failure, which some approximate differential privacy mechanisms do permit.*
>
> Along these lines, our work shows that in the situation where public data is available, there are alternative solutions that enjoy the benefits of stronger privacy definitions. Some examples of these benefits: both zCDP and pure DP are stronger guarantees that protect against the chance of ``catastrophic failure'' as specified by the $\delta$ parameter.  Furthermore, even in the case where the analyst is comfortable with approximate DP guarantees, zCDP offers composition benefits over approximate DP. Since Gaussian estimation is a fundamental task, it is easy to see how it may be a part of a larger analysis. In that case, the much tighter composition guarantees of zCDP vs approximate DP (see the plot under “Composition under variants of differential privacy” in [6]) provide another practical use for our results.

---

> > ### Author Response · Authors · 2022-08-02
> > **Response to Reviewer 2n6x (Continued)**
> >
> > **On the connection between approx DP and pure DP with public data:**
> >
> > > *I wonder if at least some of the results can be implied by the range-independent results for approximate DP. For $(\varepsilon,\delta)$-DP, in the worst case roughly $\delta$ fraction of the dataset could be released as-is, which might be viewed equivalently as "public dataset". It would be great if the authors could discuss the connections.*
> >
> > The reviewer points out an interesting possible connection between approximate DP and public-private pure DP. After thinking it over, we make the following remark:
> >
> > *Algorithms satisfying $(\varepsilon,\delta)$-DP do not necessarily satisfy public-private $\varepsilon’$-DP for any $\varepsilon’\geq 0$, and for any designation of the rows as ``private data''.*
> >
> > To see why, consider the following mechanism that, with probability $\delta$, outputs the entire dataset. This is $(0,\delta)$-DP. However, no matter what non-empty subset of dataset rows we designate as our ”private dataset”, individuals in that “private dataset” have a non-zero probability of being released (which would, otherwise, have been $0$ had they not participated). Therefore, they do not enjoy pure DP guarantees for any $\varepsilon'\geq 0$.
> >
> > In other words, $\delta$ could be interpreted as the probability of catastrophic failure -- but that failure could take any form (releasing one private row or the entire private dataset to the public, or any other leaks about the private data itself). If we simply choose to release $\delta$ fraction of our private dataset (and label it "public"), the catastrophic failure probability $\delta$ would essentially become $1$ because our algorithm is deterministically releasing private data with complete certainty, and this would annihilate any non-trivial privacy guarantees that we wanted to achieve. As such, there is no immediate way to translate $(\varepsilon,\delta)$-DP results to our setting.
> >
> > **Suggestions on writing:**
> > We thank the reviewer for suggestions on improving our writing! We plan to take these suggestions into account for the final version of the manuscript. We will expand our section on Gaussian mixtures to highlight more precisely how we use public data. We will also add the problem formulations of estimating Gaussians and Gaussian mixtures before describing our results.
> >
> > **Final comments:**
> > If the reviewer has additional concerns and questions that they feel have not been sufficiently addressed in our response, we would be happy to address them in the discussion phase.
> >
> > **References:**
> > [1] Nicolas Papernot, Thomas Steinke. “Hyperparameter Tuning with Renyi Differential Privacy”. ICLR 2022.
> > [2] Ishaq Aden-Ali, Hassan Ashtiani, Gautam Kamath. “On the Sample Complexity of Privately Learning Unbounded High-Dimensional Gaussians”. ALT 2021.
> > [3] Hassan Ashtiani, Christopher Liaw. “Private and polynomial time algorithms for learning Gaussians and beyond”. COLT 2022.
> > [4] Gautam Kamath, Argyris Mouzakis, Vikrant Singhal, Thomas Steinke, Jonathan Ullman. “A Private and Computationally-Efficient Estimator for Unbounded Gaussians”. COLT 2022.
> > [5] John M. Abowd, Robert Ashmead, Ryan Cumings-Menon, Simson Garfinkel, Micah Heineck, Christine Heiss, Robert Johns, et al. “The 2020 Census Disclosure Avoidance System TopDown Algorithm”. Harvard Data Science Review.
> > [6] Joseph P. Near and Chiké Abuah. Programming Differential Privacy. https://programming-dp.com/notebooks/ch8.html.

---

> > ### Comment · Reviewer_2n6x · 2022-08-07
> > **Response to author comment**
> >
> > Thank the authors for the thoughtful response. The example you gave helps to clarify my question about the relation to $(\epsilon, \delta)$-DP, which is my main concern. Therefore, I'm willing to raise my score.

---

### Official Review · Reviewer_Hm4a · 2022-07-12

**Rating:** 4
**Confidence:** 3
**Soundness:** 2 fair
**Presentation:** 2 fair
**Contribution:** 2 fair

**Summary:**

This paper studies differential private estimation by taking advantage of some public data. It has been shown that some improvements on the sample complexity can be made when given a certain amount of public data.

**Questions:**

(1) This paper simply assumes the public comes from a Gaussian distribution. This assumption is too strong in realism. For example, the data in the economy and biology domain can usually be heavy-tailed. Is that possible to extend the current assumption to be a general one?

(2) In Algorithm 1, using only d+1 samples $\tilde{X}_i$ to obtain the covariance $\widehat{\Sigma}$ leads to inconsistent estimation, especially when the dimension d is large. Would this make the output inefficient in a certain case?

(3) Similarly, in Lemma 2.5, how could you make sure the matrix $\Sigma_Y=\frac{1}{L}\widehat{\Sigma}^{-1/2}\Sigma\widehat{\Sigma}^{-1/2}$ is positive definite?

(4) In Lemma 2.5, is the probability at least 1-$\beta$ going to 1 with the increasing of the dimension d? In the current presentation, it seems $\beta$ is an arbitrary constant greater than 0.

**Limitations:**

Yes

**Strengths And Weaknesses:**

Strengths: The targeted problem is interesting. The idea of taking advantage of public data is good. This paper is easy to follow.

Weakness:
(1) The organization of this paper should be improved. There are more than thirty pages appendix. The appendix is too long.

(2) This paper simply assumes the public comes from a Gaussian distribution. This assumption is too strong in realism. For example, the data in the economy and biology domain can usually be heavy-tailed. Is that possible to extend the current assumption to be a general one?

(3) In Algorithm 1, using only d+1 samples $\tilde{X}_i$ to obtain the covariance $\widehat{\Sigma}$ leads to inconsistent estimation, especially when the dimension d is large. Would this make the output inefficient in a certain case?

(4) Similarly, in Lemma 2.5, how could you make sure the matrix $\Sigma_Y=\frac{1}{L}\widehat{\Sigma}^{-1/2}\Sigma\widehat{\Sigma}^{-1/2}$ is positive definite?

(5) In Lemma 2.5, is the probability at least 1-$\beta$ going to 1 with the increasing of the dimension d? In the current presentation, it seems $\beta$ is an arbitrary constant greater than 0.

(6) No numerical performance is conducted to demonstrate the effectiveness of the proposed method.

---

> ### Author Response · Authors · 2022-08-02
> **Response to Reviewer Hm4a**
>
> We would like to thank the reviewer for their thoughtful feedback and questions. We are glad to hear that the reviewer finds the problem we study interesting, and shares our belief that taking advantage of public data is a promising approach toward addressing shortcomings in private data analysis. We hope that our study offers insights and inspires future work on understanding the role of public data in private data analysis.
>
> In the following, we address specific questions and concerns raised by the reviewer, in order of appearance.
>
> **Organization and appendix length:**
>
> > *The organization of this paper should be improved. There are more than thirty pages appendix. The appendix is too long.*
>
> Parts of our argument are inherently technical, and thus unfortunately, require many pages to present a complete, precise, and correct argument. As is standard for theory papers in NeurIPS/ICML, we elected to give a broad overview of the approach in the body (highlighting key technical components when possible), and eschewing the details to the appendix/supplement. If the reviewer has any specific suggestions on how to improve presentation in the body, we would be happy to incorporate them.
>
> **On Gaussianity assumptions:**
>
> > *This paper simply assumes the public comes from a Gaussian distribution. This assumption is too strong in realism. For example, the data in the economy and biology domain can usually be heavy-tailed. Is that possible to extend the current assumption to be a general one?*
>
> First, we would also like to point out to the reviewer that our results also address Gaussian mixtures – a less restrictive modeling assumption which is employed in practice (e.g. even for complex speech data, GMMs are a component in deep-learning-based speech recognition systems [1]).
> We agree that our study of Gaussians and Gaussian mixtures does not cover all realistic use cases. Still, we believe it is a fundamental problem that is of relevance to the NeurIPS community (see papers [2, 3], and [4] – note that [4] won a NeurIPS 2018 best paper award). Our study of this relatively ‘simple’ setting already uncovers some conceptual takeaways: a small amount of public data can remove necessary boundedness requirements even when the public data's distribution differs significantly from the private data's distribution; and the private sample complexity may be improved in terms of other parameters (e.g. the number of mixture components in our results about Gaussian mixtures) in other cases using trace amounts of public data.
> With regards to extensions to other cases: it is common to study the Gaussian case first, and then the heavy-tailed case afterwards. See for example, in the case of private estimation without public data, [5] and then [6]. We suspect that similar techniques to ours could also be applied to the heavy-tailed setting, after modifying the appropriate concentration and spectral inequalities. However, as the reviewer comments, the paper is already long and technical, so these details could be worked out in further work.

---

> > ### Author Response · Authors · 2022-08-02
> > **Response to Reviewer Hm4a (Continued)**
> >
> > **Clarifications to technical concerns:**
> > > *In Algorithm 1, using only $d+1$ samples $\widetilde X_i$ to obtain the covariance $\widehat Σ$ leads to inconsistent estimation, especially when the dimension $d$ is large. Would this make the output inefficient in a certain case?*
> >
> > As the reviewer points out, using $d+1$ public samples in Algorithm 1 results in weak accuracy guarantees on our initial ‘coarse estimate’ computed from public data only. Our goal here is to employ a minimal amount of public data (far less than the $\Omega(d^2/\alpha^2)$ samples required non-privately for accurate estimation). We are not focused on the statistical consistency, or other asymptotic behaviours of this initial coarse public-data estimate since our focus is on how well we can perform private estimation with a very small amount of public data. The goal of this coarse estimate is just to precondition the distribution in order to make it friendly for the next estimation step using the private data, which is assumed to be available in much larger quantities.
> > In the second step of our Gaussian estimator, we refine the coarse estimate with judicious use of private data. As $d$ increases, the error in the coarse estimate translates to a $\log(d/\beta)$ factor increase in the second and third terms of the private sample complexity, compared to the bounded case. This increase is overshadowed by $d^2$ factors that would exist, even if our coarse estimate were correct up to a constant.
> > The full estimator using both public and private data achieves consistency as well as finite sample guarantees with respect to the number of private samples. We do not claim statistical efficiency, which is not commonly studied in the CS literature on DP estimation.
> >
> > > *Similarly, in Lemma 2.5, how could you make sure the matrix $\Sigma_Y = \frac 1 L \hat{\Sigma}^{-1/2}\Sigma\hat{\Sigma}^{-1/2}$ is positive definite?*
> >
> > We know that with probability 1 over samples from a $d$-dimensional Gaussian, $\hat{\Sigma}$ is positive definite (PD). Matrix inverses and square-roots preserve PD-ness. The result of pre-multiplying and post-multiplying a PD matrix with a PD matrix remains PD.
> >
> > > *In Lemma 2.5, is the probability at least $1-\beta$ going to $1$ with the increasing of the dimension $d$? In the current presentation, it seems $\beta$ is an arbitrary constant greater than $0$.*
> >
> > $\beta$ is a parameter of the problem: the algorithm designer can choose it to be any value $>0$ they wish (even possibly dependent on $d$, if need be). When one targets a lower failure probability $\beta$ for the algorithm, Algorithm 1 (public data preconditioning) is adjusted to be more conservative, to account for more low probability events where the public data covariance does not transform the data to the desired range.
> >
> > **On experiments:**
> > > *No numerical performance is conducted to demonstrate the effectiveness of the proposed method.*
> >
> > We follow a long line of work on understanding the theoretical sample complexity of private estimation (e.g. [2, 3, 7]). Following these theoretical works and inspired by their techniques, subsequent works have explored practical tools for private estimation [8]. We position our work into the former line, and believe that building practical tools for these problems is a very interesting direction for follow-up work.
> >
> > **Final comments:**
> > We hope our responses to the reviewer’s questions, including the ones about the technical details, have helped address the concerns over the soundness and the significance of our work. If these concerns persist, or if the reviewer has any additional comments or questions, we would be happy to continue the conversation during the discussion phase to reach a resolution.

---

> > > ### Author Response · Authors · 2022-08-02
> > > **Response to Reviewer Hm4a (Continued)**
> > >
> > > **References:**
> > > [1] Yongqiang Wang, Abdelrahman Mohamed, Due Le, Chunxi Liu, Alex Xiao, Jay Mahadeokar, Hongzhao Huang et al. "Transformer-based acoustic modeling for hybrid speech recognition." ICASSP 2020.
> > > [2] Gautam Kamath, Or Sheffet, Vikrant Singhal, Jonathan Ullman. “Differentially Private Algorithms for Learning Mixtures of Separated Gaussians.” NeurIPS 2019.
> > > [3] Ishaq Aden-Ali, Hassan Ashtiani, Christopher Liaw. “Privately Learning Mixtures of Axis-Aligned Gaussians.” NeurIPS 2021.
> > > [4] Hassan Ashtiani, Shai Ben-David, Nicholas Harvey, Christopher Liaw, Abbas Mehrabian, Yaniv Plan. “Nearly tight sample complexity bounds for learning mixtures of Gaussians via sample compression schemes.” NeurIPS 2018.
> > > [5] Gautam Kamath, Jerry Li, Vikrant Singhal, Jonathan Ullman. “Privately Learning High-Dimensional Distributions”. COLT 2019.
> > > [6] Gautam Kamath, Vikrant Singhal, Jonathan Ullman. “Private Mean Estimation of Heavy-Tailed Distributions”. COLT 2020.
> > > [7] Vikrant Singhal, Thomas Steinke. “Privately Learning Subspaces”. NeurIPS 2021.
> > > [8] Sourav Biswas, Yihe Dong, Gautam Kamath, Jonathan Ullman. “CoinPress: Practical Private Mean and Covariance Estimation”. NeurIPS 2020.

---

> > > > ### Author Response · Authors · 2022-08-03
> > > > **Response to Reviewer Hm4a (Continued)**
> > > >
> > > > **A proof-of-concept numerical result:**
> > > > Although we position our work as theoretical, since our approach is relatively simple to implement on top of an existing private algorithm, we offer some proof-of-concept simulations that demonstrate the effectiveness of public data in private statistical estimation. Below, we show some plots that evaluate **$1$ public sample private mean estimation** (the algorithm described in Section 2.1 and in more detail in Appendix B.1.1).
> > > >
> > > > We examine the effect of 1 public sample on the performance of CoinPress [8] with its best parameter setting ($t=2$), in a case where the initial a priori bounds on the mean are weak. Concretely, we draw $n$ samples from a $d=50$ dimensional Gaussian, $\mathcal N (\mu , I_d)$, where $\mu = 1000 \cdot [1,1,..,1]^T$, and set our a priori bound to be $R = 1000\sqrt d$ for CoinPress.
> > > >
> > > > Please find the image of the two plots via the following link: https://i.imgur.com/duYteRQ.png
> > > >
> > > > We follow the evaluation protocol from [8]: we target zCDP with $\rho = 0.5$, and at each sample size $n$ we run the estimator 100 times and report the 10% trimmed mean of error from the ground truth. (The second plot is the first one zoomed in.)
> > > >
> > > > The numerical result demonstrates the promise of utilizing public data for private data analysis, and confirms the takeaway that very little public data can help greatly when a priori knowledge of the private data is weak.  As is visible from these plots, the error of our public-private algorithm nearly matches the error of the non-private algorithm.
> > > >
> > > > Note that these results are very preliminary – thorough tuning and evaluation of these algorithms (which is necessary to bring these algorithms to practice) is an important direction for future work.

---

### Meta-Review · Area_Chair_zgVp · 2022-08-29

**Recommendation:** Accept
**Confidence:** Certain

**Metareview:**

This paper studies private estimation with a small amount of public data. The idea is that the small public dataset may allow for significantly stronger positive results (e.g., in terms of sample complexity of private data). The authors study two fundamental settings in this direction -- estimating a Gaussian and a Gaussian mixture -- and provide interesting and technically non-trivial positive results. The consensus from the reviews and subsequent discussion is that this work is both conceptually and technically interesting.

**Award:**

No

---

### Decision · Program_Chairs · 2022-09-14

Accept